# PM2.5 leads to adverse pregnancy outcomes by inducing trophoblast oxidative stress and mitochondrial apoptosis via KLF9/CYP1A1 transcriptional axis

Shuxian Li[1†], Lingbing Li[2†], Changqing Zhang[1], Huaxuan Fu[3], Shuping Yu[4], Meijuan Zhou[1], Junjun Guo[1], Zhenya Fang[1], Anna Li[1], Man Zhao[1], Meihua Zhang[1]*, Xietong Wang[1,5]*

[1]Key Laboratory of Birth Regulation and Control Technology of National Health Commission of China, Maternal and Child Health Care Hospital of Shandong Province Affiliated to Qingdao University, Jinan, China; [2]The Second Hospital, Cheeloo College of Medicine, Shandong University, Jinan, China; [3]Jinan Environmental Monitoring Center of Shandong Province, Jinan, China; [4]School of Public Health, Weifang Medical University, Weifang, China; [5]Department of Obstetrics and Gynecology, Shandong Provincial Hospital Affiliated to Shandong First Medical University, Jinan, China

*For correspondence:
meihua2013@163.com (MZ);
xietong789656@163.com (XW)

†These authors contributed equally to this work

Competing interest: The authors declare that no competing interests exist.

**Abstract** Epidemiological studies have demonstrated that fine particulate matter (PM2.5) is associated with adverse obstetric and postnatal metabolic health outcomes, but the mechanism remains unclear. This study aimed to investigate the toxicological pathways by which PM2.5 damaged placental trophoblasts in vivo and in vitro. We confirmed that PM2.5 induced adverse gestational outcomes such as increased fetal mortality rates, decreased fetal numbers and weight, damaged placental structure, and increased apoptosis of trophoblasts. Additionally, PM2.5 induced dysfunction of the trophoblast cell line HTR8/SVneo, including in its proliferation, apoptosis, invasion, migration and angiogenesis. Moreover, we comprehensively analyzed the transcriptional landscape of HTR8/SVneo cells exposed to PM2.5 through RNA-Seq and observed that PM2.5 triggered overexpression of pathways involved in oxidative stress and mitochondrial apoptosis to damage HTR8/SVneo cell biological functions through CYP1A1. Mechanistically, PM2.5 stimulated KLF9, a transcription factor identified as binding to *CYP1A1* promoter region, which further modulated the CYP1A1-driven down-stream phenotypes. Together, this study demonstrated that the KLF9/CYP1A1 axis played a crucial role in the toxic progression of PM2.5 induced adverse pregnancy outcomes, suggesting adverse effects of environmental pollution on pregnant females and putative targeted therapeutic strategies.

## Editor's evaluation

This study offers a valuable finding using a mouse model exposed to PM2.5 samples collected from highly polluted city air. The solid evidence provided strongly supports the assertions of the authors that PM2.5 triggers a KLF9/CYP1A1 signaling pathway, resulting in placental dysfunction, oxidative stress, mitochondrial issues, and adverse gestational outcomes. This research holds substantial relevance for medical biologists engaged in studying environmental factors impacting maternal and fetal health.

## Introduction

Particulate matter (PM) is one of the most widespread and harmful air pollutants, consisting of a complex mixture of airborne particles and various chemicals, such as organic chemicals, transition metal oxides, acids, dust, and sulfates (*Daellenbach et al., 2020*). The toxicity of PM is closely related to the properties of the particles, such as the composition, quantity, mass, shape, size, and surface area concentration (*Gao et al., 2020*). PM2.5 is defined as the aerodynamic diameter of less than 2.5 μm. PM2.5 has been proven to penetrate into the alveoli through inhalation, depositing on the alveolar wall and eventually enter the systemic blood circulation, leading to negative effects on primary organs, including the lung, cardiovascular, immune system, nervous system and genital system (*Chu et al., 2019*; *Yue et al., 2019*; *Qiu et al., 2018*). Moreover, the respirable particles have been classified as a known human carcinogen by the International Agency for Research on Cancer, with a growing number of studies uncovering the adverse effects in human and animals (*Loomis et al., 2013*; *Jeong et al., 2021*). Globally, China is considered to be one of the most severely polluting countries by PM2.5 as a result of the rapid economic development it has experienced. Long-term exposure to PM2.5 is estimated to cause around one million premature deaths annually (*Xue et al., 2019*). In response, the Chinese Government enacted a series of strict laws in 2013 to combat environment pollution (*Huang et al., 2018*). As a result, the annual average PM2.5 level in China dropped to 45.5 μg/m$^3$ in 2017, compared to 67.4 μg/m$^3$ in 2013 (*Zhang et al., 2019*). However, according to the air quality monitoring system of the United Nations Environment Program (UNEP) in cooperation with the Swiss technology company IQAir (https://www.iqair.com/), the 2021 average PM2.5 concentration in China was still 6.5 multiples of the World Health Organization's annual air quality guideline value. PM2.5 pollution continues to be significant due to economic development and geographical conditions especially in the northern plains of China (*Jin et al., 2022*).

Due to the toxic action of PM2.5, mounting evidence attests that exposure to PM2.5 during pregnancy leads to a variety of adverse pregnancy outcomes, such as placental dysfunction, ovarian dysfunction, impaired fetal development, lower birth weights, preterm births, reduced gestational age, congenital disability, and stillbirth (*Zhou et al., 2020*; *Bekkar et al., 2020*; *Leung et al., 2022*; *Wang et al., 2017*). Exposure to PM2.5 during gestation can also have adversely effects on the health of the offspring, such as cardiovascular disease, respiratory damage, and neurodevelopmental impairment, as it has been reported that the etiology of many adult diseases may originate in the fetal period (*Hart et al., 2021*; *Lavigne et al., 2016*). However, most of the present research on PM2.5 in adverse obstetric outcomes is focused on the epidemiological aspects, and thus the underlying mechanisms are not yet clear. The placenta is the most important organ from the mother to deliver oxygen and nutrients to the fetus to support its regular growth. Additionally, the placenta is uniquely able to adapt to environmental changes through active metabolism to guarantee the needs of the mother and fetus (*Blake et al., 2020*). When exceeding the regulatory range of the placenta, PM2.5 can adversely affect the development of the placenta and the fetal growth (*Iyengar and Rapp, 2001*). In in vivo human placenta perfusion studies, the PM2.5 could enter the circulating bloodstream through the alveolar wall and deposited in the placenta, thus negatively affecting or even destroying placental function and creating a potential risk to the fetus (*Martens et al., 2017*; *Michikawa et al., 2022*). Previous studies have found that urban PM2.5 affects the function of the placenta, such as placental thrombosis, chorioamnionitis, and inadequate trophoblast formation and invasion, which adversely impacts fetal growth by interfering with mother-fetus interactions (*Martens et al., 2017*; *Ghosh et al., 2021*; *Guo et al., 2021*). PM2.5 exposure during pregnancy also induces abnormal alterations in the placental genome DNA methylation profile, which is primarily concentrated in genes associated with reproductive development, immune regulation and material metabolism (*Nawrot et al., 2018*). Moreover, Nääv Å et al. demonstrated that PM2.5 caused placental trophoblast cytotoxicity, impaired hormone regulation, inflammation and oxidative stress (*Nääv et al., 2020*). However, as a significant target of PM2.5 stimulation, the underlying molecular mechanisms associated with placental trophoblastic PM2.5 exposure have not been completely elucidated.

Metabolism of xenobiotics by cytochrome P450 signaling pathway is mainly responsible for the intracellular degradation of exogenous substances such as polycyclic aromatic hydrocarbons (PAHs), halogen-containing organic compounds and heterocyclic amines (*Lu et al., 2021*). As a cytochrome P450 (CYP) enzyme, CYP1A1 plays an essensial role in the metabolism of endogenous substrates (such as hormones 17β estradiol [*Kisselev et al., 2005*], melatonin [*Goh et al., 2021*],

inflammatory mediators arachidonic acid [*Sroczyńska et al., 2022*] and eicosapentaenoic acid [*Schwarz et al., 2004*]) and exogenous substrates (such as PAHs [*Chen et al., 2022b*], doxin [*Molcan et al., 2017*], and halogenated aromatic hydrocarbons [*Kasai et al., 2008*]). Many PAH compounds, such as benzo[a]pyrene, are not toxic in themselves and only become cytotoxic and carcinogenic after activation by CYP1A1. For example, CYP1A1 converts benzo[a]pyrene into the carcinogenic 7,8-dihydroxy-9,10-epoxybenzo[a]pyrene by multiple oxidation steps which causes severe oxidative damage to cells by generating excess reactive oxygen species (ROS) (*Zajda et al., 2019*; *Gastelum et al., 2020*). It has been proven that PM2.5 contributes to the increase of CYP1A1 in human neutrophils, bronchial epithelial cells, and alveolar macrophage due to the adsorption of adsorbs aromatic hydrocarbons on their surface, which in-turn inducing oxidative stress, inflammatory responses, and apoptosis (*Chen et al., 2019*; *Van Winkle et al., 2015*; *Abbas et al., 2009*). Yue C et al. also found that PM2.5 induced malformations in zebrafish embryonic heart development by activating CYP1A1 (*Yue et al., 2017*). However, very little research has examined the role of CYP1A1 in PM2.5 induced placental dysplasia and adverse pregnancies.

Our previous epidemiological study identified that prenatal exposure to increased PM2.5 could shorten the gestation period and increased the risk of preterm birth in Jinan, the capital of Shandong Province, China (*Wang et al., 2022*). Jinan is a large city in northern China, with an area of $7.6 \times 10^3$ km$^2$ and a population of more than 9 million, and its geographical structure is a typical basin texture surrounded by mountains on 3 of its 4 sides, hindering the dispersion of environmental pollutants. In this study, we collected PM2.5 from the atmosphere in the urban area of Jinan to investigate its effects on the placenta and on pregnancy outcomes. First, scanning electron microscope (SEM) and energy dispersive spectroscopy (EDS) were used to analyze the morphology of PM2.5 particles and the elemental composition. We then investigated the effects of PM2.5 on pregnancy outcomes, placental function of mice and trophoblast cell line HTR8/SVneo. Subsequently, we performed a comprehensive analysis of the transcriptional landscape of PM2.5-exposed trophoblast cells by RNA-Seq. We found that the activation of the cytochrome P450 pathway was most prominent after PM2.5 exposure and verified that PM2.5 could trigger oxidative stress resulting in mitochondrial apoptosis, with *CYP1A1* being the key gene involved in this process. Based on bioinformatics analysis, we further found for the first time that PM2.5 activated the transcription factor KLF9, which transcriptionally promoted the expression of CYP1A1, thereby modulating oxidative stress damage and mitochondrial apoptosis. Overall, this research elucidated the mechanism of PM2.5-mediated trophoblast apoptosis and adverse pregnancy outcomes, suggesting that the KLF9/CYP1A1-related pathway could be a promising target for clinical prevention and treatment.

## Results
### Characterization of PM2.5
Microscopic morphology and structure analysis of PM2.5 particles was performed using SEM coupled with EDS. As shown in *Figure 1B*, we explicitly observed that the PM2.5 particles had randomly aggregated and scattered in the field of view, with most of them being less than 2.5 µm in length. The particles were primarily flocculent in shape, with only a few flakes and rods. Elemental analysis is conclusive evidence that directly indicated the source of pollution. Therefore, we randomly selected three locations of the collected PM2.5 to detect and quantify the elemental composition by EDS. As shown in *Figure 1C*, the elements O, N, and C were predominantly present, while other elements such as Cl, Si, and the heavy metals S, Al, and Ni were also detected. The proportion of the elements displayed the following trend: O>N > C>Cl > Si>S > Na>K > Al>Ca > Ni.

### Gestational PM2.5 exposure results in adverse pregnancy outcomes and impaired placental development
The PM2.5-exposed mouse model was established by intratracheal instillation of PM2.5 at 1.5 d, 7.5 d, and 12.5 d of pregnancy (*Figure 2A*). The amount of PM2.5 particulate in the tracheal droplets was calculated based on 2020 PM2.5 exposure of pregnant women in Jinan, Shandong province (*Wang et al., 2022*), combined with physiological indicators of pregnant mice (*Vermillion et al., 2018*). Mice were euthanized and dissected on 15.5 d, and the weight of the fetus, the number of fetuses, and the weight of the placenta were assessed. As shown in *Figure 2B and C*, PM2.5 treatment resulted

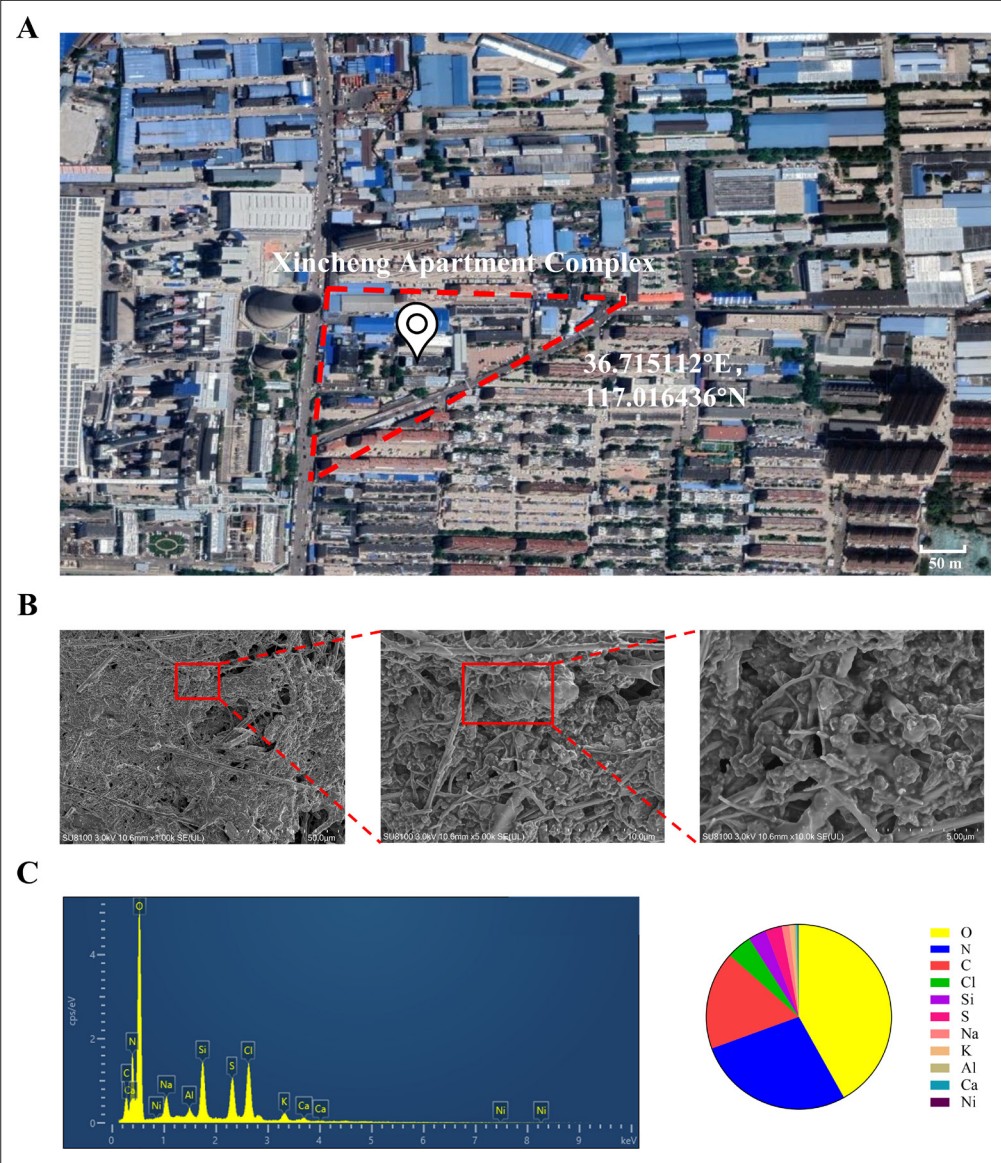

**Figure 1.** The geographic location of the PM2.5 collection site and analysis of PM2.5 morphological and elemental composition. (**A**) Map and coordinates of the PM2.5 sampling site. The image was obtained from Google Maps. (**B**) SEM images of PM2.5 in an enlarged field. Scale bar, 50 µm (left image), 10 µm (middle image) and 5 µm (right image). (**C**) Analysis of PM2.5 elemental composition via SEM coupled with EDS.

in a significant reduction in fetal numbers, embryo resorption, and the increase of stillbirths (n=8). The weight of the placenta of the pregnant mice was also decreased. These results suggested that PM2.5 had a deleterious effect on the placenta and fetal development in vivo. Next, we examined the tissue and cellular structure of the placenta by HE staining. As shown in *Figure 2D*, the placental cells in the normal group were well differentiated and densely arranged, with uniform and regular intercellular gaps; the capillaries were evenly distributed and a large number of red blood cells were visible in the vessels. Notably, compared with the control group, the cells of the labyrinth layer in the PM2.5-exposed group were loosely distributed, and disorganized, with more vacuoles and larger intercellular gaps. The distribution of the placental capillaries was reduced and the number of red blood cells was significantly lower. In addition, we used TUNEL staining to detect the occurrence of apoptosis in the placental cells and fluorescent staining with CK-7 to label trophoblast cells. As shown in *Figure 2E*, PM2.5 exposure resulted in increased levels of apoptosis in the mouse placental labyrinth cells compared with the normal group, with apoptosis being primarily observed in the

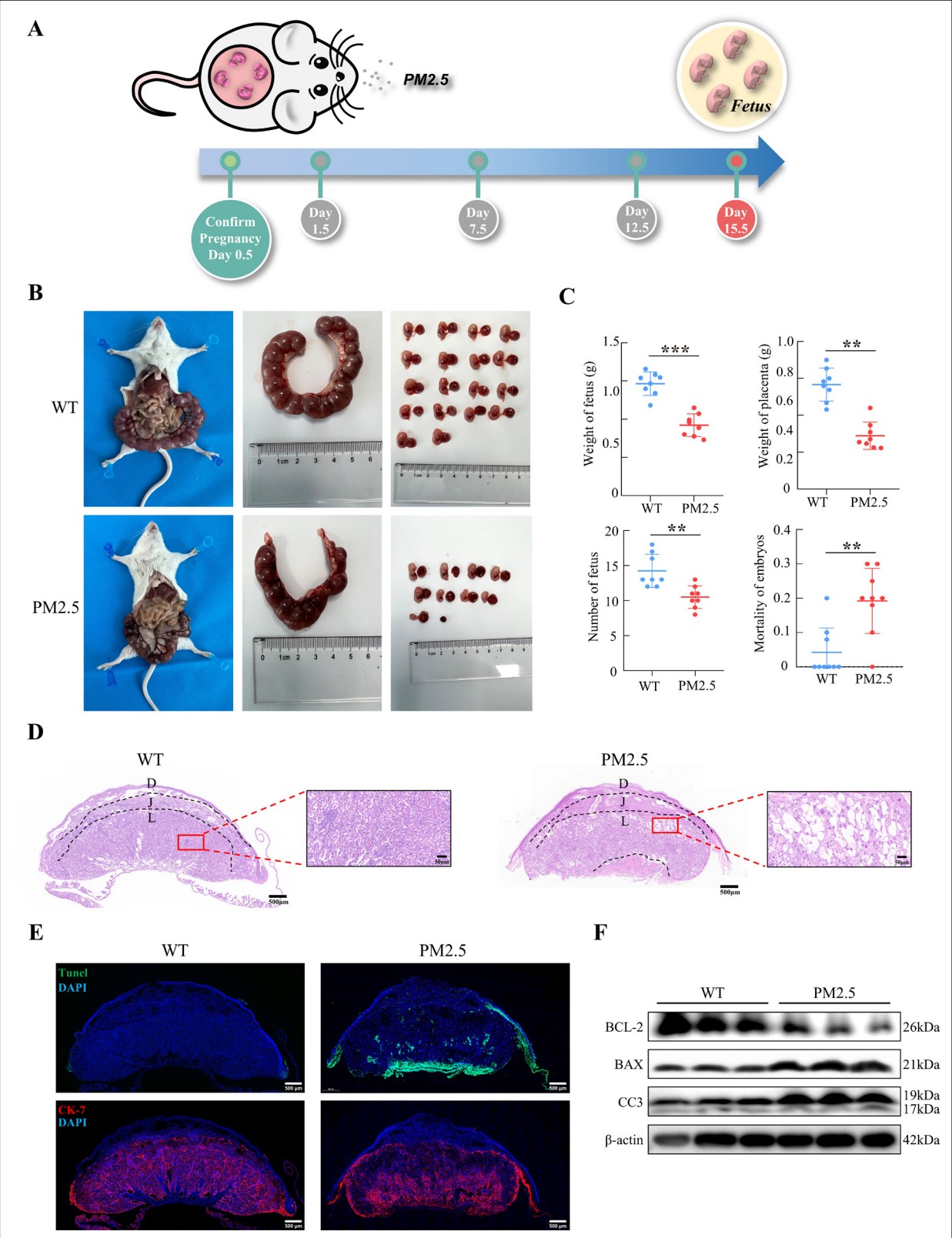

**Figure 2.** Gestational PM2.5 exposure resulted in adverse pregnancy outcomes and impaired placental development in mice. (**A**) Schematic diagram of the in vivo exposure model constructed by intratracheal instillation of PM2.5 in mice at 1.5 d, 7.5 d, and 12.5 d of pregnancy. (**B**) Representative images of uterine, fetal, and placental morphology in wild-type and PM2.5 exposed mice at 15.5 d of gestation. (**C**) The effect of PM2.5 on the fetus and placenta, including the weight of the fetus, the weight of the placenta, the number of the fetuses, and the mortality of the embryos (n=8, the dots in the

*Figure 2 continued on next page*

*Figure 2 continued*

top two graphs indicate the average placental or fetal mouse weight per pregnant mouse) (**, p<0.01; ***, p<0.001). (**D**) Representative HE staining of placental tissues from the wild-type and PM2.5-exposed mice. Compared with the wild-type placental tissues, the labyrinth layer in the PM2.5-exposed group was loosely distributed and disorganized, with a large number of vacuolated structures and large intercellular gaps (D: Decidual; J: Junctional zone; L: Labyrinth). Scale bar, 500 μm (upper images) and 50 μm (lower images) (**E**) Apoptosis of placental cells was detected by TUNEL staining. TUNEL: apoptotic cells; DAPI: nuclei; CK-7: trophoblast cells. Scale bar, 500 μm. (**F**) The expression of apoptosis-related proteins in the placental tissue of mice was detected by western blotting. PM2.5 decreased the expression of the anti-apoptotic protein BCL-2 and increased the expression of the pro-apoptotic proteins BAX and Cleaved-caspase 3 (CC3).

The online version of this article includes the following source data for figure 2:

**Source data 1.** Labelled gel images.

**Source data 2.** Raw unlabelled gel images.

trophoblasts of the mouse placental labyrinth. Apoptosis was also evident below the labyrinth layer in the PM2.5-exposed group. This site was dominated by fetal mouse umbilical cords, and increased apoptosis in this area could be caused by placenta prolonged placement during placental removal or external damage. The expression of apoptosis-related proteins including cleaved-caspase 3, BCL-2 and BAX were measured using western blotting (n=3), which also confirmed the increased levels of apoptosis in the placenta of PM2.5-exposed mice (*Figure 2F*). In summary, PM2.5 exposure resulted in poor pregnancy outcomes in mice, and this was most likely related to cellular damage and apoptosis in the placenta.

## PM2.5 impairs multiple biological functions of placental trophoblast cells

The placental trophoblast cell line HTR8/SVneo was incubated with different concentrations of PM2.5 (50, 100, and 200 μg/ml) for 24 hr based on previous studies (*Duan et al., 2020*; *Guo et al., 2022*). Transmission electron microscopy clearly indicated that PM2.5 particles significantly changed the morphology of the cells (*Figure 3A*). Compared with the control group, the PM2.5-exposed cells were swollen and deformed, with a large number of vacuoles in the cytoplasm, and with mitochondria exhibiting visibly damaged intracellular ultrastructure. The mitochondria in the cells of the PM2.5-exposed group were round or oval, with some becoming solid. The mitochondrial volume was reduced, the matrix was thickened, the cristaes were inconspicuous, and the interior of the mitochondria were precipitated with dense, indefinite material. Flow cytometry was used to assess whether cells had taken up PM2.5 based on the changes of the SSC-A (Side Scatter Area) values (*Zhao and Ibuki, 2015*). As shown in *Figure 3B*, the FSC-A remained largely the same as the control group after PM2.5 treatment, but the SSC-A increased significantly in a concentration-dependent manner. This result further proved that PM2.5 could be taken up by HTR8/SVneo cells. CCK-8 assays showed that PM2.5 significantly inhibited cell proliferation with a concentration-dependent way (*Figure 3C*). EDU staining results also showed that PM2.5 significantly inhibited cell proliferation (*Figure 3D and E*). When the PM2.5 concentration reached 100 μg/mL, cell proliferation decreased by nearly 50%. Annexin V-FITC/PI staining and flow cytometry analysis were used to detect cell apoptosis and it was found that PM2.5 significantly increased cell apoptosis (*Figure 3F and G*). The wound healing assay was carried out to assess the migration of HTR8/SVneo cells. The results indicated that the cells in the PM2.5-exposed group migrated more slowly compared to the corresponding migration rate in the control group (*Figure 3H and I*). In support of this, the Transwell invasion assays showed that the invasive activity of PM2.5-exposed HTR8/SVneo cells was significantly reduced, with very few cells visible across the bottom membrane at PM2.5 concentrations up to 200 μg/mL (*Figure 3J and K*). Finally, we detected the angiogenic ability of HTR8/SVneo cells, a process that mimics the formation of blood vessels after trophoblast invades the endometrium (*Belyakova et al., 2019*). As expected, PM2.5 had a detrimental effect on angiogenesis in trophoblast cells, and we found that at PM2.5 concentrations of 200 μg/mL, cells were largely unable to form lumen structures (*Figure 3L and M*). In conclusion, these results confirmed that PM2.5 could be phagocytosed by placental trophoblasts and it impaired trophoblast biological functions including invasion, migration, tube formation, while increasing apoptosis.

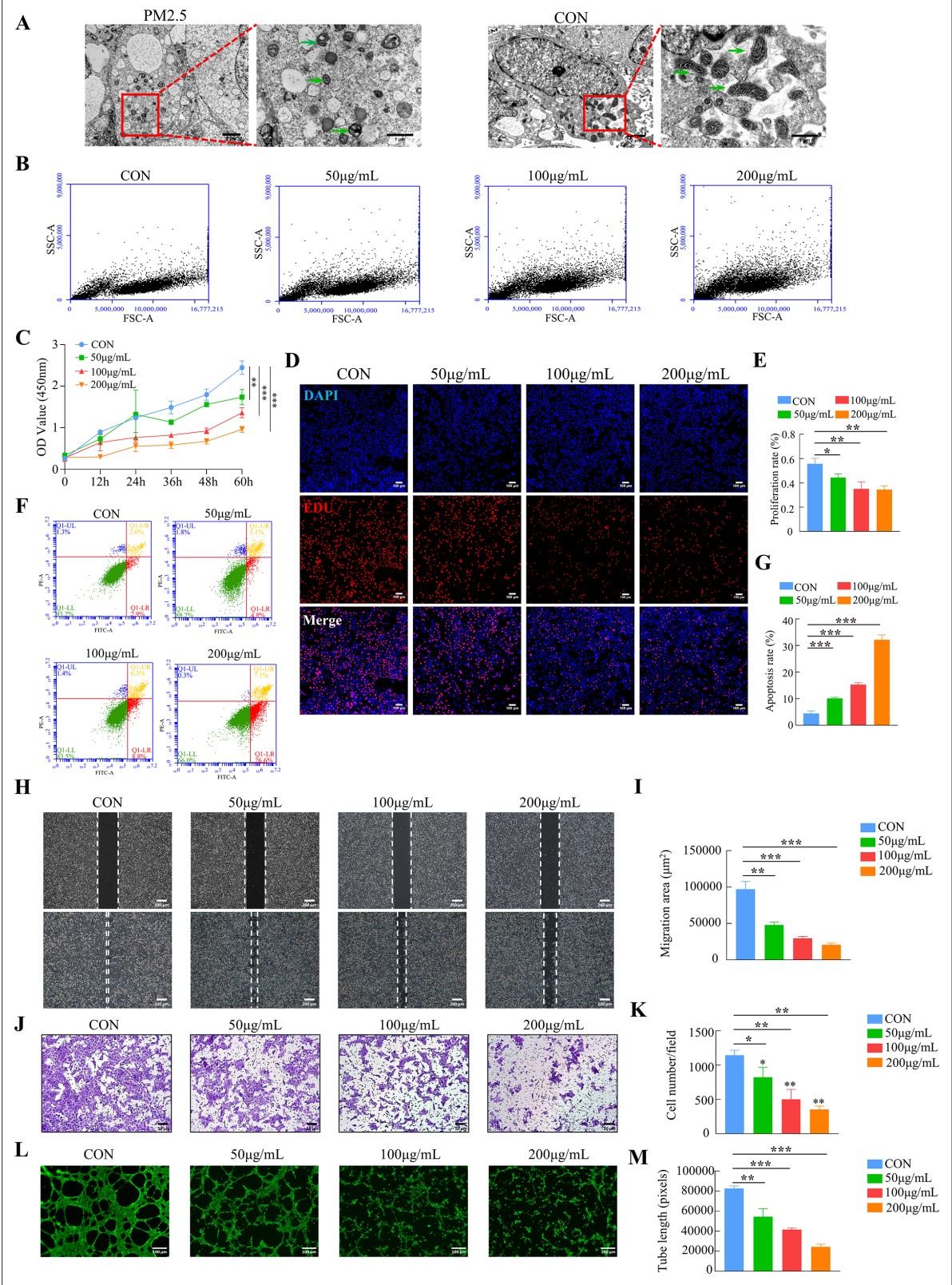

**Figure 3.** PM2.5 impaired the biological functions of HTR8/SVneo. (**A**) Representative TEM images of morphological changes of HTR8/SVneo caused by PM2.5. The green arrows indicated mitochondria in the cell. Scale bar, 2 μm (left images) and 1 μm (right images) (**B**) Detection of PM2.5 particles internalized by HTR8/SVneo cells using flow cytometry. The horizontal coordinates of the FSC-A represented the size of the particles, and the vertical coordinates of the SSC-A represented intracellular particle complexity. (FSC-A: Forward Scatter-Area; SSC-A: Side Scatter-Area) (**C**) Proliferation

*Figure 3 continued on next page*

*Figure 3 continued*

curves were determined by CCK-8 assays. The plots represented the proliferative ability of HTR8/SVneo cells. (**D**) Representative images showing the suppressed proliferation of HTR8/SVneo cells by PM2.5. The nuclei were stained with DAPI (blue), the cells in the proliferation stage were stained with EDU (red). Scale bar, 100 μm. (**E**) Quantitative analysis of the proliferation rate of HTR8/SVneo cells. (**F**) Annexin V-FITC/PI staining and flow cytometry analysis were used to determine the percentage of apoptotic cells following treatment with different concentrations of PM2.5. (**G**) Histogram analysis indicated the apoptotic rate of the cells in each group. (**H**) Representative images of the wound healing assay of HTR8/SVneo cells at 0 and 24 hr time points following treatment with different concentrations of PM2.5. Scale bar, 200 μm. (**I**) The histogram indicated the migration area (μm²) in each group. (**J**) The representative images of Transwell invasion assay at 24 h in the different groups. Scale bar, 50μm. (**K**) The histogram indicated the proportion of cells in each group. (**L**) Representative images showing cell tube formation following treatment with different concentrations of PM2.5 for 4 hr. Scale bar, 100μm. (**M**) The histogram indicated the comparison of the tube length of HTR8/SVneo cells (*, p<0.05; **, p<0.01; ***, p<0.001).

## Identification of PM2.5-exposed trophoblast gene expression profiles using high-throughput sequencing

To elucidate the molecular mechanism by which PM2.5 affected trophoblast biological functions, we performed RNA-Seq analysis. After analyzing a total of 17,795 genes identified in the sequencing data, the dispersion of the expression distribution of these genes was determined (*Figure 4—figure supplement 1A*), and the density plot in *Figure 4—figure supplement 1B* demonstrated the trends and regions of gene abundance with expression in the samples. In addition, 3D principal component analysis showed that the clusters of samples with high similarity were consistent with the actual exposure grouping, suggesting that PM2.5 significantly changed the gene expression of the HTR8 cells (*Figure 4—figure supplement 1C*). By applying the cut-off criteria of Q value<0.05 and |log2[Fold Change]|≥1, we identified 32 coding genes that exhibited differential expression between the control and PM2.5 treatment groups, comprising 24 up-regulated genes and 8 down-regulated genes. These findings were visually represented through volcano plots and heat maps (*Figure 4A and B*). The KEGG pathway enrichment analysis was performed to predict the potential regulatory mechanisms of PM2.5-induced trophoblast damage (*Figure 4C and D*). Our findings were further supported by GSEA, which revealed significant upregulation of four enriched pathways - cytochrome P450 pathway, chemical carcinogenesis pathway, ovarian steroidogenesis pathway and Steroid biosynthesis pathway - in the PM2.5-exposed group (*Figure 4E*). The results from both analyses suggest that cytochrome P450 is the most significantly enriched pathway. Activation of this pathway mediates oxidative stress and modifies mitochondrial function, while exacerbating the production and accumulation of toxic metabolic intermediates, typically reactive oxygen species (ROS) (*Yuan et al., 2019*; *Martínez et al., 2018*; *Leung et al., 2013*). We hypothesized that the cytochrome P450 pathway may play a key role in PM2.5-induced placental pathogenesis.

## PM2.5 induces trophoblast damage by triggering oxidative stress

To verify whether oxidative stress was involved in the adverse effect of PM2.5, we first detected the levels of GSH, SOD, and MDA in the placenta of mice after PM2.5 exposure. As shown in *Figure 5A*, PM2.5 exposure resulted in a significant decrease in GSH and SOD level and a considerable increase in MDA levels in the mouse placental tissues (n=8). We further detected the expression of the antioxidant proteins heme oxygenase-1 (HO-1), NADPH quinone oxidoreductase 1 (NQO-1), glutamate-cysteine ligase (GCLC), and superoxide dismutase 1 (SOD-1) in mice placenta by western blotting. The results showed that PM2.5 exposure resulted in reduced expression of these antioxidant proteins in the placenta (n=3) (*Figure 5B*). These results confirmed that PM2.5 exposure caused oxidative stress in the PM2.5- exposed mice's placenta. Additionally, the expression of the antioxidant proteins HO-1, NQO-1, GCLC, and SOD-1 was reduced after PM2.5 exposure in the placental trophoblast cell line HTR8/SVneo cells (*Figure 5C*). The expression of GSH, SOD, and MDA in HTR8/SVneo was consistent with the above results in the mice placental tissues (*Figure 5D*). The ROS levels were measured by cellular immunofluorescence staining and flow cytometry, and the results showed that there was a significant increase in the PM2.5-exposed cells (*Figure 5E and F*). Based on these results, it was hypothesized that PM2.5 induced oxidative stress damage by disrupting the intracellular antioxidant system, contributing to a large accumulation of intracellular ROS. As mentioned above, severe damage to intracellular mitochondrial morphology was observed by transmission electron microscopy (*Figure 3A*). Therefore, we detected the mitochondrial ROS levels using the Mito-SOX probe. The results revealed a considerable increase in the level of ROS in the mitochondria with increasing PM2.5

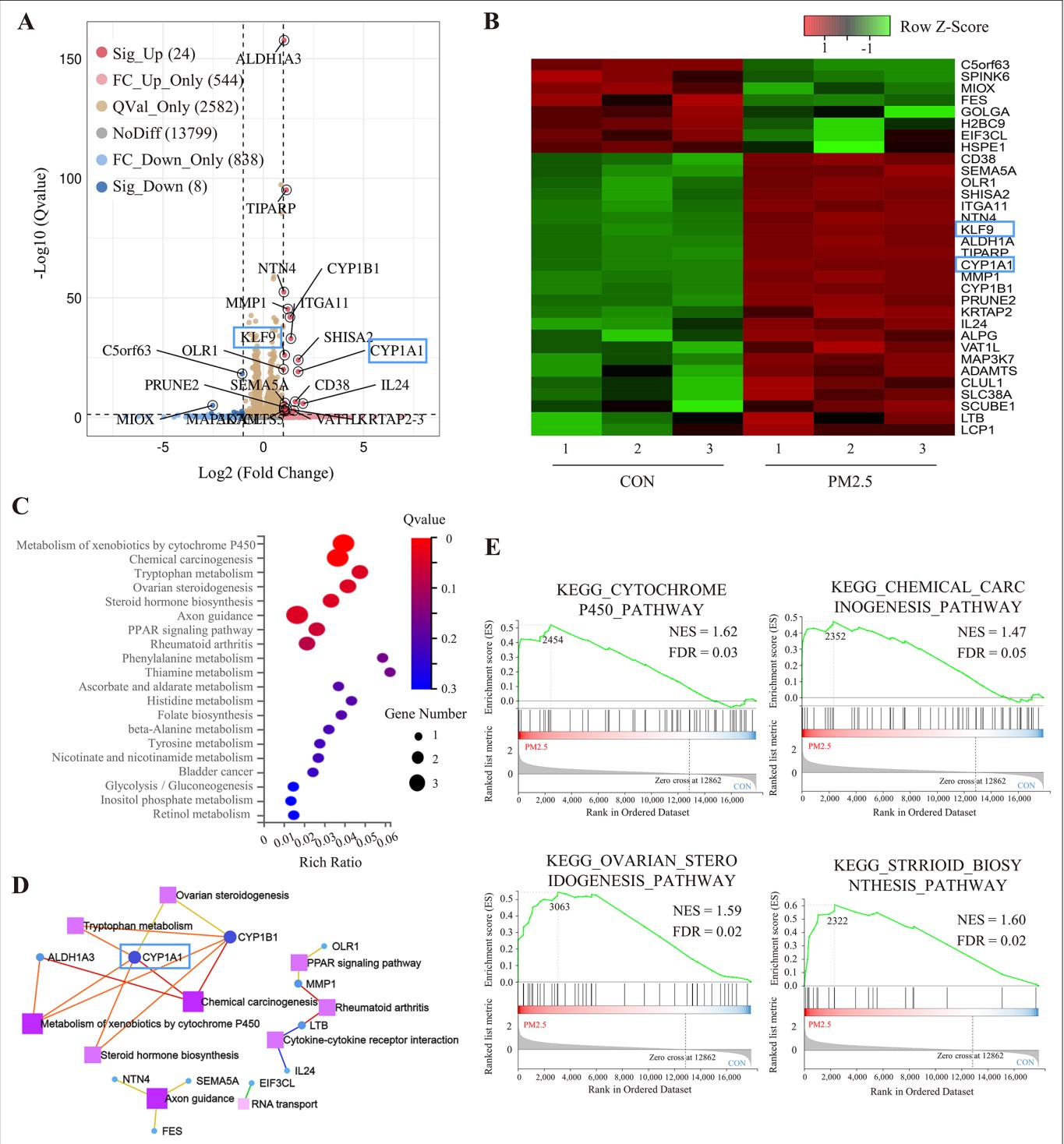

**Figure 4.** RNA sequencing and bioinformatic analysis of PM2.5-induced genomic alterations in HTR8/SVneo cells. (**A**) Volcano plot of the differentially expressed genes (DEGs) in the RNA-Seq (Qvalue <0.05; |Log2 (fold change) |>1). (**B**) Heat map visualization of sequencing data with the Row Z-Score scaling method for gene expression between samples. (**C**) Bulb map of KEGG analysis for the DEGs. The rich ratio represented the enrichment degree of DEGs. (**D**) Topological networks based on altered gene enrichment analysis. (**E**) GSEA of RNA-Seq data showed the enrichment of genes in the pathway. The zero-cross line indicates the point in which the difference between expression in the PM2.5-treated and control groups is zero. NES, normalized enrichment score; FDR, false discovery rate.

The online version of this article includes the following figure supplement(s) for figure 4:

**Figure supplement 1.** RNA sequencing and bioinformatic analysis.

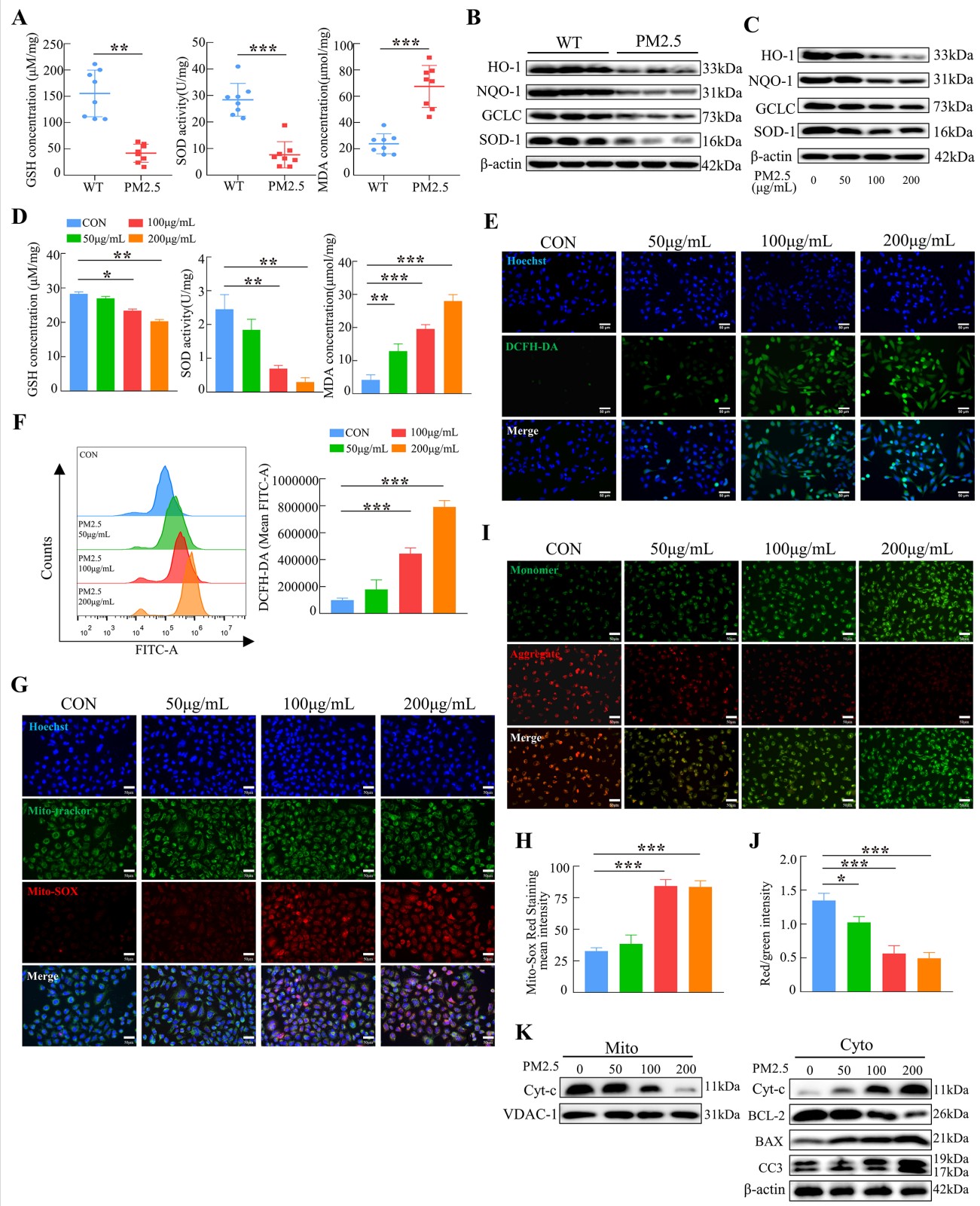

**Figure 5.** PM2.5 impaired the biological functions of HTR8/SVneo cells by triggering oxidative stress. (**A**) Detection of GSH, SOD, and MDA expression levels in the placenta of wild-type and PM2.5 exposed mice (n=8, one randomly selected placental sample per mouse). (**B**) Expression of antioxidant-related proteins HO-1, NQO-1, GCLC, and SOD-1 in the placenta of wild-type and PM2.5-exposed mice by western blotting (n=3). (**C**) Expression of antioxidant-related proteins HO-1, NQO-1, GCLC, and SOD-1 in HTR8/SVneo cells treated with different concentrations of PM2.5 (50 μg/mL, 100 μg/

*Figure 5 continued on next page*

*Figure 5 continued*

mL, and 200 µg/mL) by western blotting. (**D**) Detection of GSH, SOD, and MDA expression levels in HTR8/SVneo treated with different concentrations of PM2.5 (50 µg/mL, 100 µg/mL, 200 µg/mL). (**E**) Detection of intracellular ROS levels by DCFH-DA probe staining following treatment with different concentrations of PM2.5 (50 µg/mL, 100 µg/mL, 200 µg/mL). The nuclei were counter stained with Hoechst (blue), and ROS were stained with DCFH-DA (green). Scale bar, 50 µm. (**F**) Quantitative detection of intracellular ROS levels by flow cytometry after staining with DCFH-DA. The histogram indicated the mean levels of FITC fluorescence in each group. (**G**) Mitochondrial ROS levels in HTR8/SVneo treated with different concentrations of PM2.5 (50 µg/mL, 100 µg/mL, 200 µg/mL). Hoechst was used to counterstain the nuclei; Mito-tracker was used to stain the mitochondria; Mito-SOX was used to stain the mitochondrial ROS. Scale bar, 50 µm. (**H**) The histogram indicated the Mito-SOX Red Staining mean intensity in each group. (**I**) Mitochondrial membrane potential was assessed using JC-1 staining following treatment with different concentrations of PM2.5 (50 µg/mL, 100 µg/mL, 200 µg/mL). When the mitochondrial membrane potential was high, JC-1 aggregated in the mitochondrial matrix, forming J-aggregates, which fluoresced red; when the mitochondrial membrane potential was low, JC-1 did not aggregate in the mitochondrial matrix, thus, JC-1 was present as a monomer, which fluoresced green. Scale bar, 50 µm. (**J**) The histogram indicated the Red/Green intensity in each group. (**K**) Expression of mitochondrial apoptosis-associated proteins in the mitochondria and cytoplasm using western blotting following treatment with different concentrations of PM2.5 (50 µg/mL, 100 µg/mL, 200 µg/mL). Cytochrome C (Cyt-C) expression was detected in the mitochondria, while Cyt-C, BCL-2, BAX and Cleaved-caspase 3 (CC3) expression levels in the cytoplasm were detected. The mitochondrial marker VDAC and the cytosol marker β-actin were used to identify the purity of the mitochondria in the extracts (*, $p < 0.05$; **, $p < 0.01$; ***, $p < 0.001$).

The online version of this article includes the following source data for figure 5:

**Source data 1.** Labelled gel images.

**Source data 2.** Raw unlabelled gel images.

concentration (*Figure 5G and H*). Previous studies found that the accumulation of mitochondrial ROS led to a decrease in mitochondrial membrane potential, resulting in the impairment of mitochondrial function and causing the development of mitochondrial apoptosis in cells (*Hsieh et al., 2019*; *Rizwan et al., 2020*). The JC-1 probe was used to detect changes in the levels of the mitochondrial membrane potential. As shown in the *Figure 5I and J*, the mitochondrial membrane potential in the control group of HTR8/SVneo cells was high, and after the addition of PM2.5, the red fluorescence of JC-1 diminished and gradually showed green fluorescence, representing a decrease in the mitochondrial membrane potential. Consistently, we extracted and separated the mitochondrial proteins and cytoplasmic proteins to examine the expression of mitochondrial apoptosis-related proteins via western blotting. The results indicated that PM2.5 induced the translocation of cytochrome C from the mitochondria to the cytoplasm, and the expression of BCL-2 was decreased, whilst BAX and cleaved-caspase 3 expression were elevated (*Figure 5K*). This suggested that PM2.5 triggered the occurrence of mitochondrial apoptosis by causing oxidative stress in HTR8/SVneo cells.

To further confirm our hypothesis, N-Acetyl-L-cysteine (NAC), a widely used oxidative stress inhibitor, was used to detect whether it can reverse the effects of PM2.5-induced impairment. The addition of NAC simultaneously increased intracellular GSH and SOD levels, and diminished MDA production (*Figure 6A*). Cellular immunofluorescence and flow cytometry revealed that NAC reduced the elevated ROS levels caused by PM2.5 (*Figure 6B and C*). NAC also resulted in the reverse of cell dysfunction on the proliferation, apoptosis, invasion and angiogenesis when compared with the PM2.5 exposed group (from *Figure 6D to M*), The results of western blotting indicated that the addition of NAC reversed the PM2.5-induced increase in the expression of the pro-apoptotic proteins BAX and cleaved-caspase 3 and prevented the transfer of cytochrome C from the mitochondria to the cytoplasm, whilst increasing the expression of the anti-apoptotic protein BCL-2 (*Figure 6N*). In summary, these results demonstrated that PM2.5 resulted in cellular functional damage by triggering oxidative stress in trophoblasts.

## PM2.5 induces mitochondrial apoptosis through oxidative stress damage in trophoblast cells via CYP1A1

KEGG enrichment analysis showed that PM2.5 had the most significant effect on the metabolism of xenobiotics by cytochrome P450, primarily affecting CYP1A1, CYP1B1 and ALDH1A3 (*Figure 4D*). The proteins encoded by these three genes were detected by western blotting, and the results showed that CYP1A1 expression was most significantly elevated (*Figure 7—figure supplement 1*). To investigate the role of CYP1A1 in PM2.5-induced defects, we first examined the expression of CYP1A1 in mice placental tissue by immunofluorescence staining and western blotting (*Figure 7A and B*). The results showed that, compared to the control group, the expression of CYP1A1 was

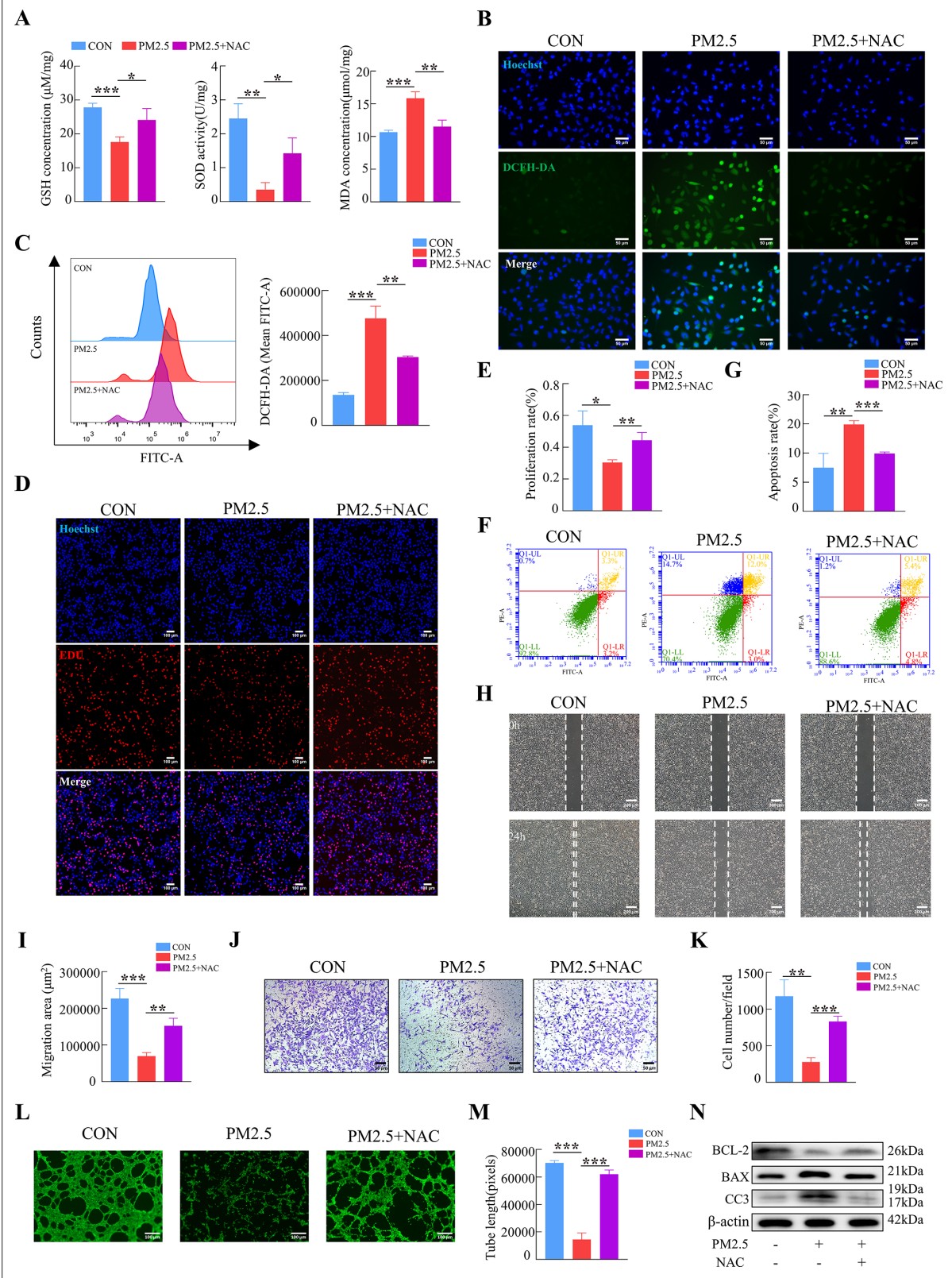

**Figure 6.** NAC reversed PM2.5-induced impairment of HTR8/SVneo cell function by inhibiting oxidative stress. (**A**) GSH, SOD, and MDA expression levels in HTR8/SVneo cells treated with PM2.5 (100μg/mL) and NAC (5 mM). (**B**) Intracellular ROS levels following treatment with PM2.5 (100 μg/mL) and NAC (5 mM). The nuclei were stained blue with Hoechst, the intracellular ROS in cells were stained green with DCFH-DA. Scale bar, 50 μm. (**C**) Quantitative detection of intracellular ROS levels by flow cytometry after staining with DCFH-DA. The histogram indicated the mean FITC

*Figure 6 continued on next page*

*Figure 6 continued*

fluorescence of each group (100 µg/mL PM2.5 and 5 mM NAC). (**D**) Images showing the proliferation of the HTR8/SVneo cells treated with PM2.5 (100 µg/mL) and NAC (5 mM) by EDU assay. The nuclei were stained with DAPI (blue), and the proliferating cells were stained with EDU (red). Scale bar, 100 µm. (**E**) Quantitative analysis of the proliferation rate of HTR8/SVneo cells. (**F**) Detection of the percentage of apoptotic cells treated with PM2.5 (100 µg/mL) and NAC (5 mM) using flow cytometry assay. (**G**) The histogram analysis showed the apoptotic rate of cells in each group. (**H**) Representative images of the wound healing assay of HTR8/SVneo cells at the 0 and 24 hr. Scale bar, 200 µm. (**I**) The histogram showed the migration area (µm²) in each group. (**J**) Representative images of the Transwell invasion assays after 24 hr in the different groups. Scale bar, 50 µm (**K**) The histogram showed the cell counts/field in each group. (**L**) Representative images showing tube formation at different concentrations of PM2.5 after 4 hr. Scale bar, 100 µm. (**M**) The histogram showed the quantification of tube length of HTR8/SVneo cells. (**N**) The protein expression of BCL-2, BAX, Cleaved-Caspase3 by western blotting (*, p<0.05; **, p<0.01; ***, p<0.001).

The online version of this article includes the following source data for figure 6:

**Source data 1.** Labelled gel images.

**Source data 2.** Raw unlabelled gel images.

significantly increased by PM2.5 exposure. Similarly, in PM2.5-exposed HTR8/Svneo cells, the levels of *CYP1A1* mRNA transcripts and protein were both elevated as revealed by qRT-PCR (*Figure 7C*), immunofluorescence staining (*Figure 7D*) and western blotting (*Figure 7E*), which confirmed that CYP1A1 could play an important role in PM2.5 induced trophoblast dysfunction. Next, we used two siRNAs to knock down *CYP1A1* expression in HTR8/Svneo (*Figure 7F*) and then detected whether the effect of PM2.5 on trophoblast cells was modified. The results showed that the elevated levels of ROS caused by PM2.5 were reduced following *CYP1A1* knockdown (*Figure 7G and H*). The elevated mitochondrial ROS levels caused by PM2.5 were also reduced after *CYP1A1* knockdown as shown by the Mito-SOX staining (*Figure 7I and J*). Meanwhile, JC-1 staining also revealed that the knockdown of *CYP1A1* reversed the decrease in mitochondrial membrane potential caused by PM2.5 (*Figure 7K and L*). Western blotting results showed that PM2.5 led to a decrease in BCL-2 expression, an increase in BAX expression, the translocation of cytochrome C from the mitochondria to the cytoplasm and the activation of downstream cleaved-caspase3 expression. When *CYP1A1* was knocked down, these changes were significantly reversed (*Figure 7M*). These results confirmed that CYP1A1 was essential in oxidative stress damage and mitochondrial apoptosis in HTR8/Svneo cells caused by PM2.5.

Furthermore, we tested whether the knockdown of *CYP1A1* reduced the effect of PM2.5 on the biological functions of HTR8/Svneo cells. EDU, Annexin V-FITC/PI staining, wound-healing, Transwell and tube formation assays results suggested that *CYP1A1* knockdown can reverse PM2.5-induced impairment of HTR8/Svneo biological functions, such as proliferation, invasion, migration, angiogenesis, and apoptosis (*Figure 8*). Together, these results showed that CYP1A1 played an essential role in PM2.5-induced oxidative stress damage in HTR8/Svneo cells, and knockdown of *CYP1A1* expression not only reversed PM2.5-induced HTR8/Svneo oxidative stress and mitochondrial apoptosis, but also alleviated PM2.5-induced changes to cellular biological functions.

## KLF9 binds to the CYP1A1 promoter region to induce transcription

RNA-Seq showed that PM2.5 could regulate *CYP1A1* expression at the transcriptional level, so we examined the stability of *CYP1A1* mRNA by qRT-PCR after treatment with actinomycin D to determine whether the increase in *CYP1A1* expression was related to its post-transcriptional regulation by PM2.5. The results showed that the half-lives of *CYP1A1* mRNA in the PM2.5-exposed and non-exposed groups were 5.62 hr and 7.08 hr, respectively, and the difference was not significant (*Figure 9A*). We then assessed the transcriptional activity of the *CYP1A1* promoter using a dual luciferase reporter assay and found enhanced transcriptional activity of *CYP1A1* in the PM2.5 exposed cells compared with the control cells (*Figure 9B*). Previous studies have shown that the expression of some cytochrome P450 family genes are regulated by transcription factors (TFs) (*Degrelle et al., 2022*; *Liu et al., 2022*; *Rannug, 2022*; *Shivaram et al., 2023*). Thus, whether PM2.5 modified *CYP1A1* expression via TFs in trophoblast cells was assessed. Using the JASPAR database (https://jaspar.genereg.net/), HOCOMOCO MoLoTool (https://molotool.autosome.org/), and the hTFtarget database (http://bioinfo.life.hust.edu.cn/hTFtarget#!/), we predicted dozens of TFs that could bind to the *CYP1A1* promoter region (TSS, –532/+88). Interestingly, these TFs were further confined when mapped to the DEGs in the RNA-Seq data, and KLF9 emerged as the only potential TF. Next, we attempted to knock down or over-express KLF9 to determine its role in the regulation of CYP1A1 by qRT-PCR

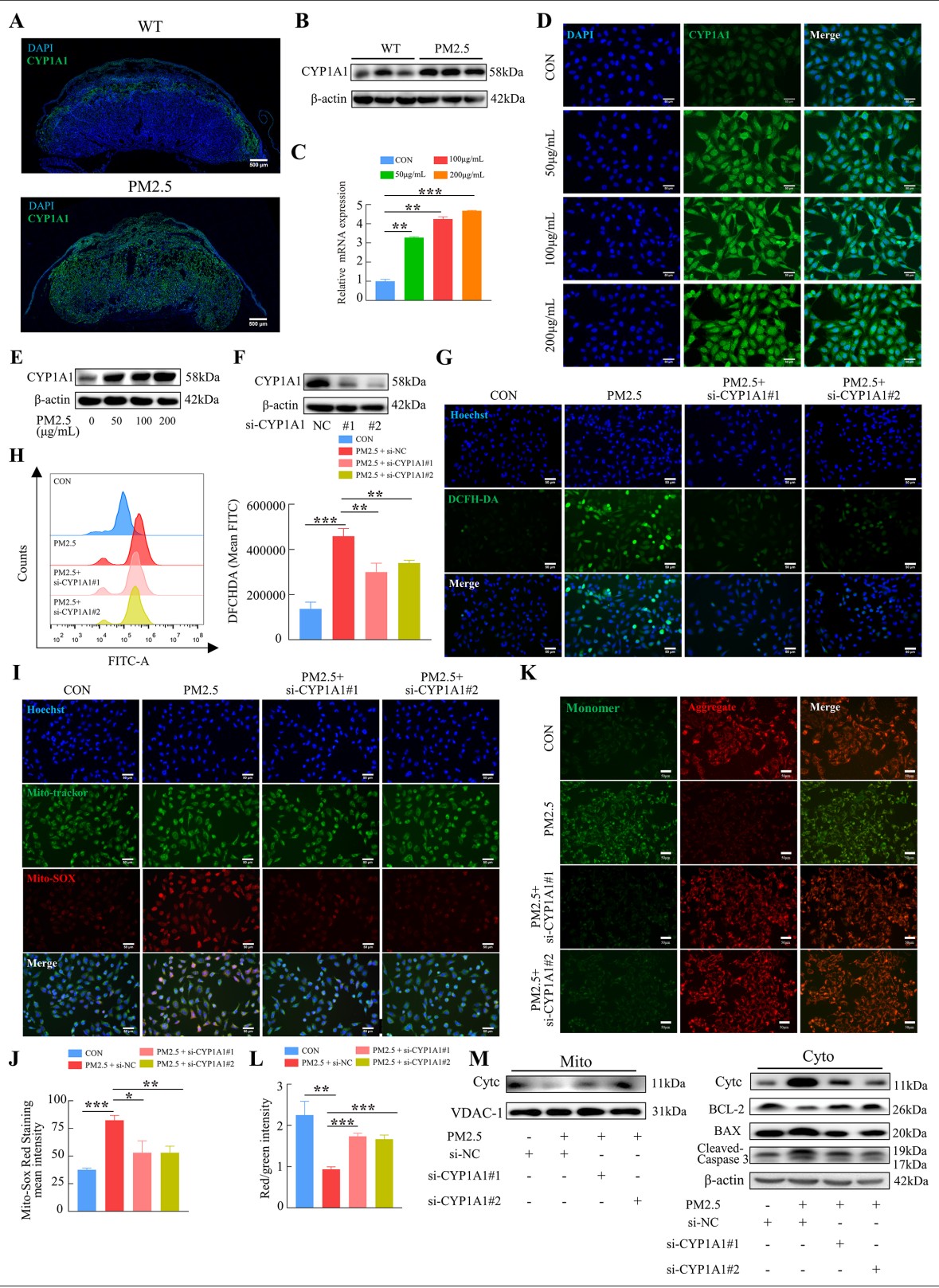

**Figure 7.** PM2.5 caused mitochondrial apoptosis through oxidative stress damage in HTR8/SVneo cells via CYP1A1. (**A**) Immunofluorescence staining of CYP1A1 in mice placental tissue sections. The nuclei were stained with DAPI (blue). The CYP1A1 expressing cells were stained with a specific antibody (green). Scale bar, 500 µm. (**B**) The protein expression levels of CYP1A1 in the placental tissues were detected by western blotting (n=3). (**C**) Relative mRNA expression levels of *CYP1A1* in HTR8/SVneo cells were treated with different concentrations of PM2.5 (50 µg/mL, 100 µg/mL, 200 µg/mL).

*Figure 7 continued on next page*

*Figure 7 continued*

(**D**) Immunofluorescence images of CYP1A1 in HTR8/SVneo cells were treated with different concentrations of PM2.5 (50 μg/mL, 100 μg/mL, 200 μg/mL). The nuclei were stained with DAPI (blue), and CYP1A1 was stained with specific antibody (green). Scale bar, 50 μm. (**E**) The protein expression levels of CYP1A1 in HTR8/SVneo cells treated with different concentrations of PM2.5 (50 μg/mL, 100 μg/mL, 200 μg/mL). (**F**) The effects of *CYP1A1* knockdown using two siRNAs by western blotting. (**G**) Quantitative detection of intracellular ROS by flow cytometry after staining with DCFH-DA. The histogram showed the mean FITC in each group. (**H**) Representative images of DCFH-DA staining for intracellular ROS (PM2.5: 100 μg/mL). The nuclei were stained with Hoechst (blue), and the ROS in cells were stained with DCFH-DA (green). Scale bar, 50 μm. (**I**) Mitochondrial ROS levels in HTR8/SVneo (PM2.5: 100 μg/mL). Hoechst was used to label the nuclei of cells in blue; Mito-tracker was used to label the mitochondrial sites; Mito-sox was used to label the mitochondrial ROS. Scale bar, 50 μm. (**J**) The histogram indicated the Mito-SOX Red Staining mean intensity in each group. (**K**) Detection of mitochondrial membrane potential by JC-1 staining (PM2.5: 100 μg/mL). Scale bar, 50 μm. (**L**) The histogram indicated the Red/Green intensity in each group. (**M**) Mitochondrial apoptosis-associated protein expression in mitochondria and cytoplasm using western blotting (PM2.5: 100 μg/mL). Cyt-C expression was detected in mitochondria, and Cyt-C, BCL-2, BAX and Cleaved-caspase 3 (CC3) expression levels in the cytoplasm were detected. The mitochondrial marker VDAC and the cytosol marker β-actin were used to identify the purity of the mitochondria in the extracts. (si-*CYP1A1*#1, si-*CYP1A1*#2: siRNAs to knockdown *CYP1A1*; si-NC: siRNAs Negative Control; *, p<0.05; **, p<0.01; ***, p<0.001).

The online version of this article includes the following source data and figure supplement(s) for figure 7:

**Source data 1.** Labelled gel images.

**Source data 2.** Raw unlabelled gel images.

**Figure supplement 1.** Gene expression in HTR8/SVneo cells under PM2.5 treatment.

**Figure supplement 1—source data 1.** Labelled gel images.

**Figure supplement 1—source data 2.** Raw unlabelled gel images.

(*Figure 9C and D*) and western blotting (*Figure 9E*). Notably, the dual-luciferase assay revealed that over-expression of KLF9 increased the transcriptional activity of *CYP1A1* (*Figure 9F*). To identify the transcriptional regulatory elements of *CYP1A1* that are responsive to KLF9, the binding sites of KLF9 to the promoter region of *CYP1A1* were mutated and the reporter plasmid was co-transfected with the pRL-TK into HTR8/Svneo cells that stably overexpressed KLF9. We found that mutation of either site#1 (-456 bp to −440 bp) or mutation site #2 (-162 bp to −146 bp) did not affect the promotive effect of KLF9 on the transcription of *CYP1A1*, but this was not the case for site #3 (-64 bp to −49 bp). Consistently, further simultaneous mutation of the three sites resulted in an abrogation of the promotive effect of KLF9 (*Figure 9F*). These results suggested that a response element of the *CYP1A1* promoter located in the −64 bp to −49 bp region, GAAGGAGGCGTGGCC, was required for the transcription of of CYP1A1 meditated by KLF9. Subsequently, we investigated the binding of KLF9 to the *CYP1A1* promoter using ChIP analysis. The chromatin was precipitated with antibodies specific for KLF9 or IgG, and PCR analysis with the primers (spanning the −96/+16 bp region of the *CYP1A1* promoter) showed that KLF9 was able to bind directly to the *CYP1A1* promoter (*Figure 9G and H*). Furthermore, we observed a positive correlation between KLF9 and CYP1A1 expression in the trophoblast layer of human placental tissue by immunohistochemical fluorescence co-staining in 30 randomly selected pregnant placental tissues in the clinic (*Figure 9I and J*). Taken together, these results indicated that KLF9 could bind to the *CYP1A1* promoter region and drive its transcriptional activity in human trophoblast cells, highlighting an important mechanistic axis for PM2.5 regulation of CYP1A1 and its downstream effects.

## KLF9/CYP1A1 activation and signaling are necessary for PM2.5-induced oxidative stress damage and trophoblast mitochondrial apoptosis

To investigate the possible role of KLF9 in PM2.5-induced trophoblast dysfunction, we first demonstrated that KLF9 expression in mouse placenta was markedly increased in response to PM2.5 stimulation, particularly in the labyrinth layer (*Figure 10A and B*). Mirroring the findings in mice, the expression of KLF9 in HTR8/Svneo cells was increased by PM2.5 treatment (*Figure 10C*). Cellular immunofluorescence showed that the increase in KLF9 expression was primarily observed in the nucleus (*Figure 10D*). The protein expression level of KLF9 in the nucleus was increased with the concentration of PM2.5 by western blotting (*Figure 10—figure supplement 1A*). Next, we knocked down the expression of *KLF9* by two siRNAs in HTR8/Svneo cells. Using DCFH-DA, we found that the knockdown of *KLF9* abrogated the increase in intracellular ROS levels induced by PM2.5 (*Figure 10E*

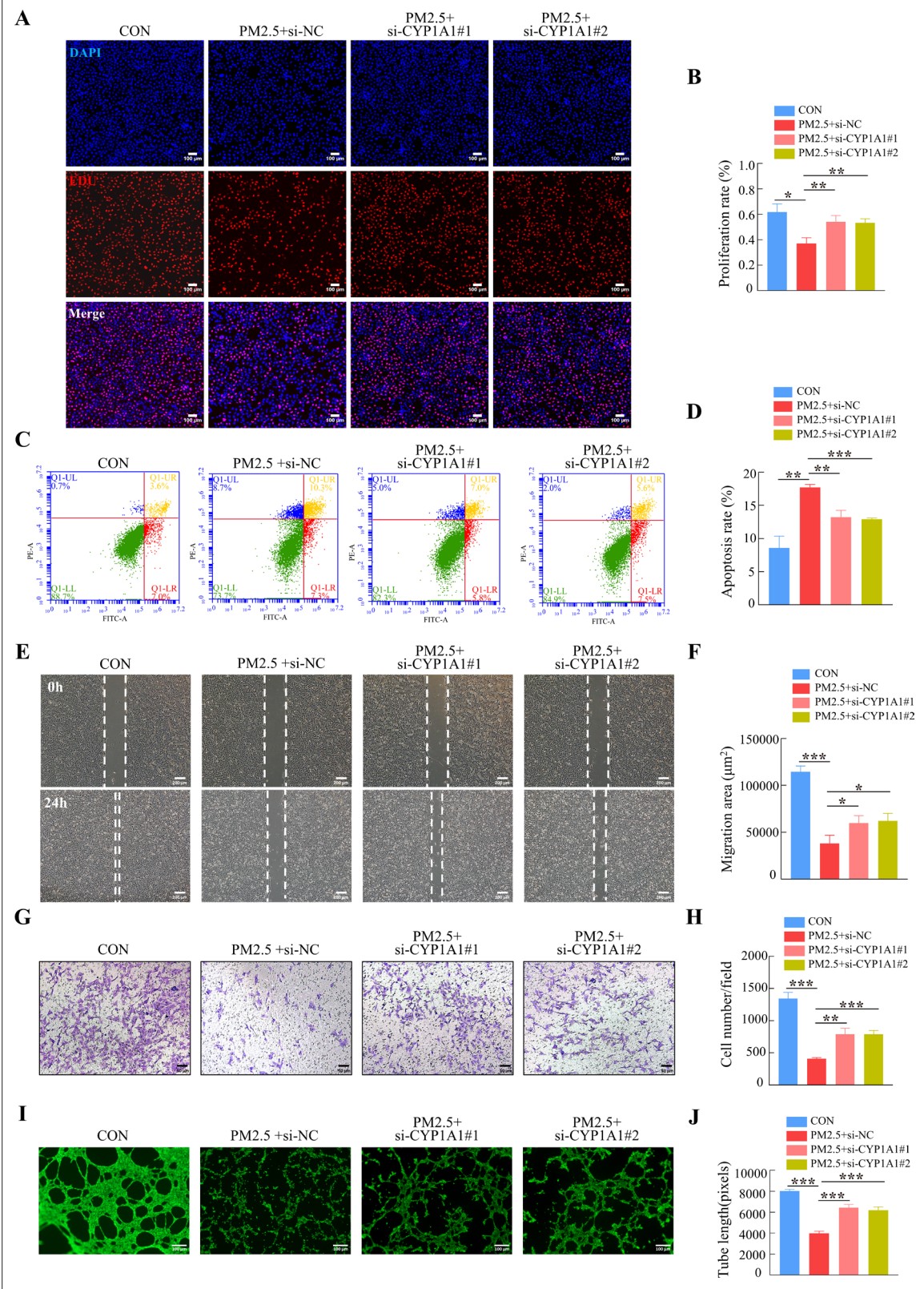

**Figure 8.** *CYP1A1* knockdown reversed HTR8/SVneo cell dysfunction induced by PM2.5. (**A**) Images showing the proliferation of HTR8/SVneo (PM2.5: 100 μg/mL). The nuclei were stained with DAPI in blue, the cells in the proliferation stage were stained with EDU in red. Scale bar, 100 μm. (**B**) The histogram analysis showed the proliferation rate of the cells in each group. (**C**) Detection of the percentage of apoptotic cells by flow cytometry (PM2.5: 100 μg/mL). (**D**) The histogram analysis indicated the apoptosis rate of the cells in each group. (**E**) Representative images of the wound healing

*Figure 8 continued on next page*

*Figure 8 continued*

assay of HTR8/SVneo cells at the 0 and 24 hr time points (PM2.5: 100 μg/mL). Scale bar, 200 μm. (**F**) The histogram showed the migration area (μm²) in each group. (**G**) Representative images of the Transwell invasion assay after 24 hr in the different groups. Scale bar, 50 μm. (**H**) The histogram indicated the cell counts/field in each group. (**I**) Representative images showing cell tube formation after 4 hr. Scale bar, 100 μm. (**J**) Histogram showed the quantification of the tube length formed by HTR8/SVneo cells. (*, p<0.05; **, p<0.01; ***, p<0.001).

*and F*). *KLF9* knockdown also prevented mitochondrial ROS production (*Figure 10G1*), resulted in an increase in mitochondrial potential (*Figure 10H and J*), and blocked the translocation of cytochrome C from the mitochondria to the cytoplasm. As a result, mitochondrial apoptosis was also reduced by *KLF9* knockdown as revealed by the upregulation of BCL-2 expression, downregulation of BAX and cleaved-caspase 3 expression (*Figure 10K*). Consistently, knockdown of *KLF9* also abrogated the disruption of trophoblast function by PM2.5 (*Figure 10—figure supplement 1B–I*). To delineate the link between KLF9 and CYP1A1 after PM2.5 exposure, *KLF9* was then knocked down in PM2.5-exposed HTR8/Svneo cultured with lentivirus-mediated ectopic expression of CYP1A1. Notably, the suppression actions by KLF9 silencing on PM2.5-induced mitochondrial apoptosis were restored by CYP1A1 overexpression, suggesting the effect of KLF9 in regulating mitochondrial apoptosis was mediated by CYP1A1 (*Figure 10L*). In conclusion, the results confirmed that PM2.5 caused oxidative stress damage, and mitochondrial apoptosis with impairment of cell biological functions through the KLF9/CYP1A1 signaling pathway in HTR8/Svneo.

## Discussion

Exposure to PM2.5 during pregnancy has been widely reported to result in an increased risk of adverse outcomes such as gestational hypertension, fetal growth restriction, preterm birth and still-birth. Studies in Florida have shown that the closer one lives to a PM2.5 emitting plant, the greater the risk of low birth weight in the fetus (*Salihu et al., 2012*). A study of more than 350,000 newborns in Ohio found that exposure to PM2.5 even in late gestation resulted in stillbirths (*Grippo et al., 2018*). Studies conducted in Beijing and Shanghai suggested that PM2.5 exposure during pregnancy is strongly related to smaller ultrasound parameters of fetal growth. Similarly, our previous research analyzed the association between preterm birth and environmental air pollutants in Jinan, which revealed a strong correlation between preterm birth and PM2.5 (*Wang et al., 2022*). To investigate the impact of ambient PM2.5 on pregnancy in depth, we extracted PM2.5 from the atmosphere of Jinan and constructed a PM2.5-exposed pregnant mouse model using intratracheal perfusion in vivo. The selection of appropriate PM2.5 exposure dosage in mice was critical for our experiments. We combined the PM2.5 exposure level of pregnant women from the Jinan (*Wang et al., 2022*) (where we collected PM2.5 particles) with physiological indicators of mice (*Vermillion et al., 2018*), then multiplied by 100-fold uncertainty factor to obtain the corresponding PM2.5 exposure dosage during mice pregnancy. The 100-fold uncertainty factor ( =10 fold interspecies difference ×10 fold interindividual variation) is used to convert a no-observed-adverse-effect level (NOAEL) from an animal toxicity study to a safe value for human intake (ADI), which is the criteria for determining experimental dosages in toxicological studies involving animals. It was originally proposed, over 60 years ago, by *Lehman and Fitzhugh, 1954*, and still forms the basis of the uncertainty factors which are in use today. Also, the 100-uncertainty factor is considered one of the criteria for in vivo experiments in authoritative studies related to PM2.5 (*Chen et al., 2022a*; *Zhang et al., 2018*; *Zhang et al., 2017*). We found that PM2.5 caused different adverse pregnancy outcomes in mice, such as an increased rate of still-births, a reduced number of fetuses, and a reduced fetal size. Additionally, we revealed that PM2.5 caused placental oxidative stress damage and trophoblast apoptosis. In addition, we performed RNA-Seq analysis and verified that PM2.5 regulated CYP1A1-mediated oxidative stress and mitochondrial apoptosis via the transcription factor KLF9 in vitro, which resulted in structural damage and functional alterations to the placenta and contributes to a poor outcome of the pregnancy.

It is well known that the placenta is a vital organ for securing the growth and development of the fetus, and its dysfunction often results in adverse perinatal outcomes. The placenta is also the target for a multitude of environmental pollutants such as PAHs, heavy metals, microplastics, perfluoroalkyl substances, and trichloroethylene, which can accumulate in the placenta and disrupt its function, thereby affecting the survival and growth of the fetus (*Vrooman et al., 2016*). PM2.5 is a mixture

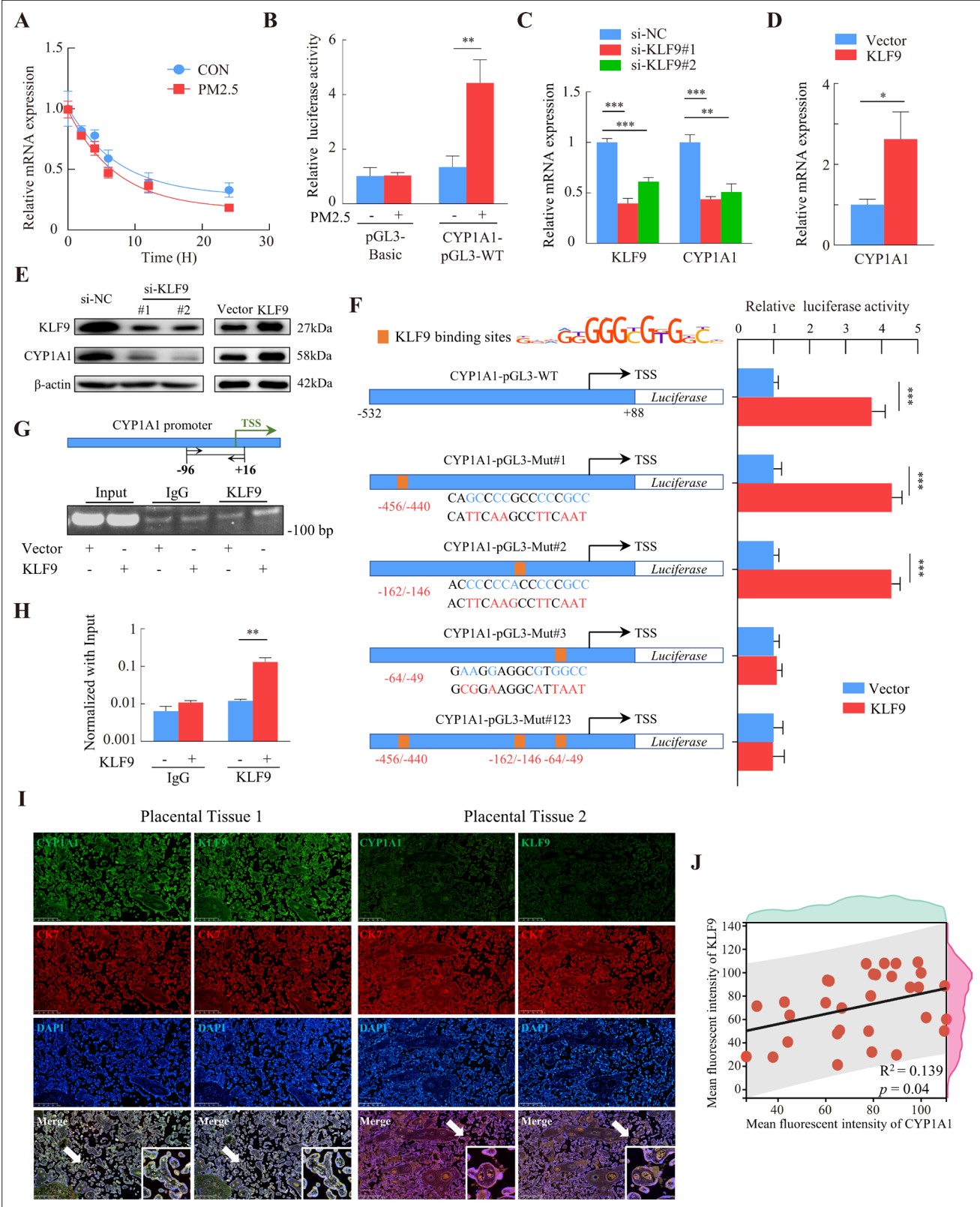

**Figure 9.** KLF9 bound to a specific region of the *CYP1A1* promoter to drive transcriptional activity. (**A**) HTR8/SVneo cells in the PM2.5-exposed or non-exposed groups were treated with 20 μM actinomycin D, and RNA was collected at different times for qRT-PCR to detect mRNA expression, and the non-linear fitted curves showed similar half-lives of *CYP1A1* mRNA. (**B**) HTR8/SVneo cells were cotransfected with CYP1A1-pGL3-WT and pRL-TK (internal control), and then exposed to 100 μg/mL PM2.5 for 24 hr. Dual luciferase assays were used to detect the fluorescence intensity. (**C and**

*Figure 9 continued on next page*

*Figure 9 continued*

D) Expression of *CYP1A1* mRNA in *KLF9* knockdown and over-expression HTR8/SVneo cells as detected by qRT-PCR. (**E**) Western blotting was used to detect the modulation of CYP1A1 protein expression by KLF9. (**F**) Different human *CYP1A1* promoter-luciferase reporter gene structures together with pRL-TK were co-transfected in HTR8/SVneo cells stably over-expressing KLF9. Schematic graph on the left showed the binding sites of the *CYP1A1* promoter predicted from the motif of KLF9 and mutant reporter genes constructed based on sites. The bars on the right showed the relative luciferase activity of different *CYP1A1* promoters in KLF9 overexpressing cells (KLF9) compared to control cells (Vector) using dual-luciferase reporter gene assays. (**G**) Chromatin from KLF9 over-expression HTR8/SVneo cells was subjected to ChIP assay using KLF9 antibody or control IgG. PCR amplification with primers spanning the –96/+16 bp region of the *CYP1A1* promoter was performed. A 2% agarose gel electrophoresis was performed on PCR products. (**H**) RT-qPCR analysis quantitatively demonstrated that KLF9 overexpression increased its binding to the endogenous *CYP1A1* promoter. (**I**) Immunofluorescence staining showed co-staining of CYP1A1 (green) or KLF9 (green), with CK7 (red) and DAPI (blue) in human placental tissue. Scale bar, 500 µm. (**J**) Data shown were representative results of three repeated assays and are represented as mean ± SD. (si-*KLF9*#1, si-*KLF9*#2: siRNAs to knockdown *KLF9* expression; si-NC: siRNAs Negative Control; * p<0.05, ** p<0.01, *** p<0.001).

The online version of this article includes the following source data for figure 9:

**Source data 1.** Labelled gel images.

**Source data 2.** Raw unlabelled gel images.

of several environmental pollutants, and due to the small size, it can penetrate the alveoli and pass through the bloodstream to be deposited in the placenta, making the placenta an organ of interest for studying the impact of PM2.5 on pregnancy. Previous studies have identified that prenatal exposure to PM2.5 resulted in an increased odds of having a placental abruption, inflammation, and hyperco-agulability with vascular thrombosis and imbalances of immune cells in the placenta (*Ibrahimou et al., 2017*; *Liu et al., 2016*). Further, we found that PM2.5-induced oxidative stress damage in mouse placenta was strongly demonstrated by a significant reduction in GSH and SOD levels, an elevation in MDA production, and decreased expression of antioxidant proteins, such as HO-1, GCLC, NQO-1, and SOD-1. The levels of apoptosis were also increased in the placental trophoblast cells, so we selected the HTR8/Svneo trophoblast cell line for PM2.5 cytotoxicity assays in vitro. HTR8/Svneo cells are perpetuated by the SV40 Tag and not only maintain many of the essential and key features of the parental cells, but are also devoid of any features of neoplastic transformation (*Bilban et al., 2010*; *Dong et al., 2021*). Therefore, HTR8/Svneo cells are a more suitable model for this study than choriocarcinoma BeWo, JEG-3, and JAR cells. In the present study, PM2.5 caused severe oxidative stress damage in HTR8/Svneo cells, as evidenced by decreased SOD and GSH levels, significantly elevated intracellular MDA and ROS levels, and decreased expression of antioxidant-related proteins. Consistently, previous studies have also reported that PM2.5 or particulate air pollution could cause trophoblast cytotoxicity, inflammation and oxidative stress (*Nääv et al., 2020*; *Familari et al., 2019*). However, its downstream signaling pathways and their potential mechanisms have not been explored in detail in previous studies. In the current study, we revealed that PM2.5 not only induced the induction of elevated intracellular ROS levels, but also significantly increased mitochondrial ROS levels. This resulted in a decrease in mitochondrial membrane potential, triggering the flow of cytochrome C from the mitochondria to the cytosol and inducing the downstream cell mitochondrial apoptosis. Our study demonstrated that exposure to PM2.5 induced oxidative stress in placental trophoblast, leading to mitochondrial apoptosis and impairment of cell biological functions.

In contrast to our results, many studies have found that PM2.5 did not trigger oxidative stress. For instance, Liu Y et al. found that PM2.5 evoked placental inflammation and hypercoagulability with vascular thrombosis but not oxidative stress in rats (*Liu et al., 2016*). Scientists may have reached these different conclusions because PM2.5 surfaces absorb different environmental pollutants. Studies have indicated that the morphological characteristics of PM2.5 are closely related to its components. For example, organic pollutant particles are irregularly clustered, fly ash particles from coal combustion were mostly spherical in shape, and construction particles were mostly sharp-edge (*Ghadikolaei et al., 2020*; *Liu et al., 2009*). SEM of the PM2.5 samples in this study showed that they primarily consisted of flocculent structures. Elemental composition by EDS revealed that the elements O, N, and C were prominent in the gathered PM2.5. These results, in addition to the presence of a chemical plant nearby the sampling site, strongly suggested that a large number of organic pollutants had adhered to the surface of the PM2.5 we collected. Studies have shown that the toxicological effects of PM2.5 on cells could depend on the pollutant composition of PM2.5. In particular, organic pollutants can lead towards the development of oxidative stress in cells. For example, PM2.5 containing

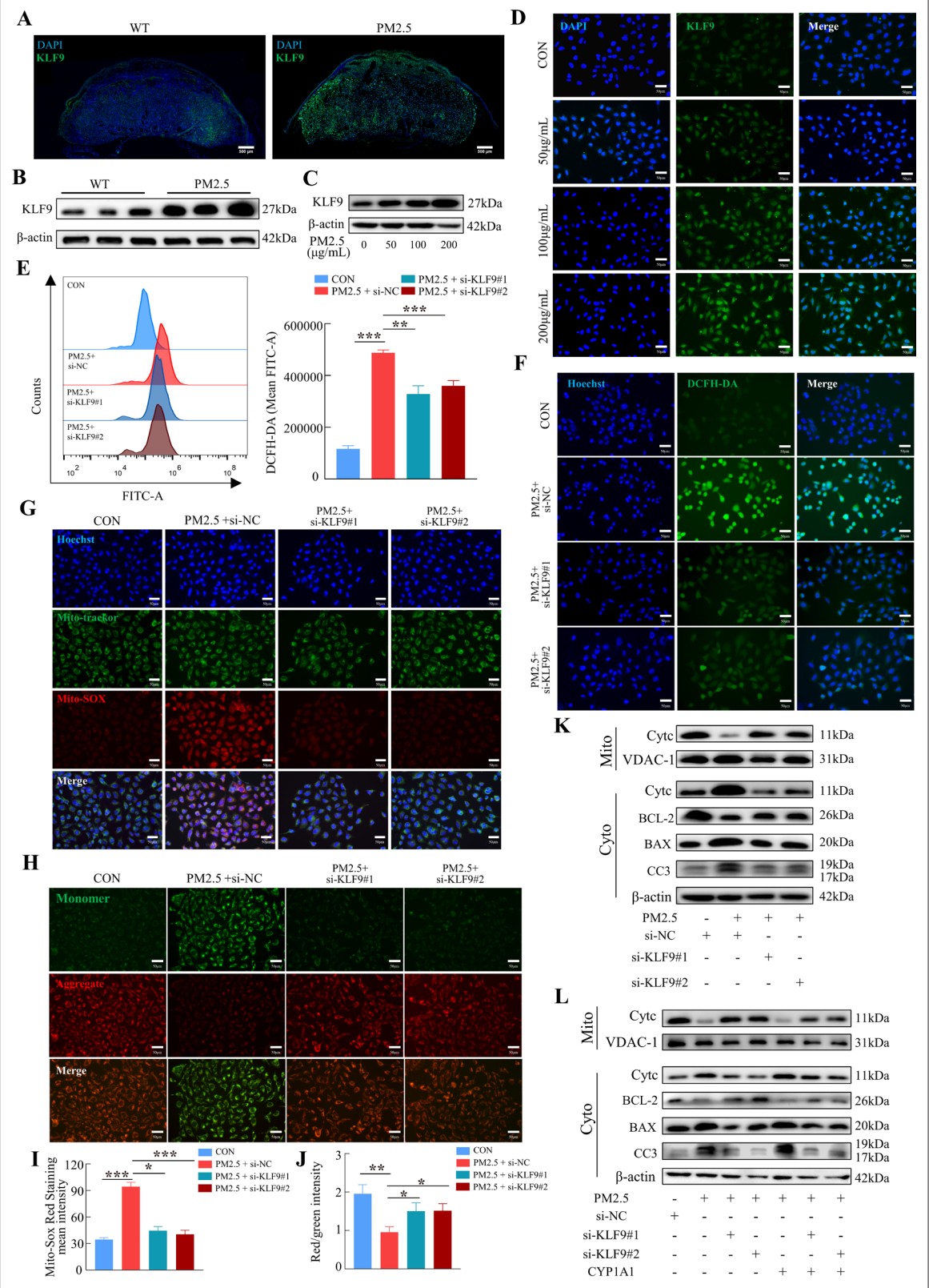

**Figure 10.** The KLF9/CYP1A1 signaling pathway was essential in PM2.5-induced oxidative stress damage and mitochondrial apoptosis in HTR8/SVneo cells. (**A**) Expression of KLF9 in mice placental tissue sections was detected by immunofluorescence staining. The nuclei were stained with DAPI in blue. KLF9 was stained with a specific antibody (green). Scale bar, 500μm. (**B**) Detection of KLF9 expression in mouse placental tissues by western blotting (n=3). (**C**) The expression of KLF9 in HTR8/SVneo at different concentrations of PM2.5 (50 μg/mL, 100 μg/mL, 200 μg/mL) were measured using western

*Figure 10 continued on next page*

*Figure 10 continued*

blotting. (**D**) Immunofluorescence images of KLF9 in HTR8/SVneo treated with different concentrations of PM2.5 (50 µg/mL, 100 µg/mL, 200 µg/mL). The nuclei were stained with DAPI (blue), and KLF9 was stained with specific antibody (green). Scale bar, 50 µm. (**E**) Quantitative detection of intracellular ROS by flow cytometry after staining with DCFH-DA. The histogram showed the mean FITC in each group. (**F**) Representative images of DCFH-DA staining for intracellular ROS (PM2.5: 100 µg/mL). The nuclei were stained with Hoechst (blue), the ROS in cells were stained with DCFH-DA (green). Scale bar, 50 µm. (**G**) Mitochondrial ROS levels in HTR8/SVneo (PM2.5: 100 µg/mL). Hoechst: labelling the nuclei of cells in blue; Mito-tracker: labelling the mitochondrial sites; Mito-sox: labelling the mitochondrial ROS. Scale bar, 50 µm. (**H**) Mitochondrial membrane potential was assessed based on JC-1 staining (PM2.5: 100 µg/mL). Scale bar, 50 µm. (**I**) The histogram indicated the Mito-SOX Red Staining mean intensity in each group. (**J**) The histogram indicated the Red/Green intensity in each group. (**K**) Mitochondrial apoptosis-associated protein expression levels in the mitochondria and cytoplasm were assessed using western blotting after knockdown of *KLF9* (PM2.5: 100 µg/mL). Cyt-C expression was detected in mitochondria and cytoplasm, and BCL-2, BAX and Cleaved-caspase 3 (CC3) expression levels in the cytoplasm were detected. The mitochondrial marker VDAC and the cytosol marker β-actin were used to identify the purity of mitochondria in the extract. (**L**) Detection of mitochondrial apoptosis-associated protein expression in mitochondria and cytoplasm after *KLF9* knockdown and *CYP1A1* overexpression using western blotting.

The online version of this article includes the following source data and figure supplement(s) for figure 10:

**Source data 1.** Labelled gel images.

**Source data 2.** Raw unlabelled gel images.

**Figure supplement 1.** KLF9 was involved in PM2.5-induced dysfunction in HTR8/SVneo cells.

**Figure supplement 1—source data 1.** Labelled gel images.

**Figure supplement 1—source data 2.** Raw unlabelled gel images.

benzo[b]fluoranthene, chrysene, and fluoranthene impair the antioxidant system of lung epithelial cells, leading to increased cellular ROS levels and reduced cellular activity *Deng et al., 2013*; exposure to PM2.5, which contains high concentrations of, metals, and polar organic compounds, can lead to elevated levels of 8-OHdG in the urine and elevated levels of oxidative stress *Wei et al., 2009*; motor vehicle exhaust in traffic congestion contributes to elevated levels of organic pollutants in PM2.5 levels, causing severe oxidative stress damage and genotoxicity to human lung cells (*Oh et al., 2011*). In the present study, we observed that PM2.5 exposure caused intracellular and mitochondrial oxidative stress. We speculated that this was due to the organic pollutants on the surface of PM2.5.

Mitochondria were involved in a wide range of critical cellular functions, including the oxidative phosphorylation of ATP synthesis and the metabolic degradation of sugars and lipids, and it is also damaged by a variety of environmental pollutants. PM2.5 can trigger oxidative stress damage to cells with increased levels of ROS, which in turn continues to damage mitochondrial membranes, further leading to increased release of cytochrome C from mitochondria and ultimately apoptosis. PM2.5 has been found to cause cellular mitochondrial damage in a variety of studies. For example, PM2.5 induced mitochondrial oxidative stress damage in hepatic stellate cells through the PINK1/Parking pathway, which led to cellular autophagy and liver fibrosis (*Qiu et al., 2019*). PM2.5 also induced an increase in ROS levels and a decrease in mitochondrial membrane potential in human bronchial epithelial cells, causing the cells to undergo mitochondrial apoptosis (*Shan et al., 2022*).A significant increase in PM2.5-exposed mitochondrial ROS was also found in macrophages, which inhibited M2 polarization and induced immune disorders (*Zhao et al., 2016*). In our study, it was apparent that the mitochondrial morphology of the cells was significantly altered, including a thickening and shortening of the cristae and a reduction in the volume. Importantly, Mito-SOX and JC-1 fluorescent probe staining revealed that PM2.5 induced an elevation in mitochondrial ROS and reduced mitochondrial membrane potential (MMP) in HTR8/SVneo cells. Moreover, mitochondrial-mediated endogenous apoptosis was also observed. Our results reinforced that PM2.5 promoted mitochondrial oxidative stress and mitochondrial apoptosis in trophoblast cells.

CYP1A1 is an important metabolic enzyme involved in the metabolism and degradation of PAHs (*Ugartondo et al., 2021*). It was found that CYP1A1 contributes to oxidative stress in cells. For example, CYP1A1 induction enhanced gefitinib-induced oxidative stress and apoptosis in A549 cells and was implicated with chronic obstructive pulmonary disease (*Callaway et al., 2020*). Furthermore, aromatic hydrocarbons in PM2.5 could increase the expression level of CYP1A1, inducing oxidative stress and inflammation in several cells, such as human bronchial epithelial cells (*Gu et al., 2021*), neutrophils (*Vogel et al., 2016*), and epithelial cells (*Toydemir et al., 2021*), amongst others. Activation of CYP1A1 mediates oxidative stress exacerbating the production and accumulation of toxic metabolic intermediates such as ROS and causes mitochondrial apoptosis (*Zhao et al., 2020*). In this

study, RNA-Seq analysis and validation tests indicated that CYP1A1 in the cytochrome P450 pathway was critically involved in PM2.5-induced cellular damage. Knockdown of *CYP1A1* expression not only abolished PM2.5-induced oxidative stress and mitochondrial apoptosis but also alleviated the disruption of cell biological functions via PM2.5.

Transcription factors play an important part in the modulation of the cytochrome P450 pathway. For example, AHR is an essential ligand-activated transcription factor of the cytochrome P450 pathway, controlling the expression of CYP1A1, CYP1B1, CYP1A2 and other genes in the cytochromes P450 family (**Torti et al., 2021**; **Al-Dhfyan et al., 2017**; **Zhang et al., 2022**; **Jin et al., 2021**). AHR could be activated by many environmental pollutants, including PM2.5 (**Ren et al., 2020**). Before the AHR is activated, it forms a complex with two heat shock protein 90 (Hsp90) molecules in the cytoplasm. Following activation, the AHR breaks free from binding and enters the nucleus, forming a dimer with the AHR nuclear translocator protein (Arnt) and binding to an enhancer to form a xenobiotic response element (XRE) involved in the regulation of cytochromes P450 family genes. In addition, multiple studies have found that intracellular CYP1A1 expression is regulated by various transcription factors, such as upstream stimulatory factor 1 (USF1), and Nuclear Factor erythroid 2-Related Factor 2 (Nrf2) (**Familari et al., 2019**; **Kyoreva et al., 2021**; **Takahashi and Kamataki, 2001**). In our study, actinomycin D was used to show that PM2.5 regulates CYP1A1 expression at transcriptional initiation rather than by modifying its mRNA stability. Then we used three databases, JASPAR, HOCOMOCO MoLo-Tool, and hTFtarget, to predict the upstream transcription factors of CYP1A1 and compared them with the RNA-Seq sequencing results. As a result, we found that KLF9 emerged as the only potential transcriptional regulator of CYP1A1.

Kruppel-like factor 9 (KLF9), also named Basic Transcription Element Binding protein 1 (BTEB1), is a transcription factor regulates development, differentiation and apoptosis by binding to GC-rich sites via three $C_2H_2$-type zinc fingers. KLF9 transcriptionally activates or represses downstream genes depending on the cellular environment and partner co-regulators. For example, KLF9 activated the gluconeogenic program in primary hepatocytes by directly binding to the *PAC1A* promoter (**Cui et al., 2019**). KLF9 also regulates p53 gene expression positively by binding to the GC box proximal to the *P53* promoter in HepG2 and SK-Hep1 cells (**Sun et al., 2014**). Conversely, KLF9 works as a transcriptional repressor inhibiting AKT transcription in prostate cancer cells, thereby suppressing the growth of PCa cells (**Shen et al., 2014**). There are very few studies on the relationship between KLF9 and CYP1A1, those that have assessed this have only reported it in rat liver cells. KLF9 was characterized as a trans-repressor of CYP1A1 gene in rat liver (**Imataka et al., 1992**). Importantly, our study has revealed that KLF9 exerted a positive regulatory effect on CYP1A1 in humans, which is in contrast to its previously reported transcriptional repression of CYP1A1 in rat liver. Furthermore, we identified the –64 bp to –49 bp region of the *CYP1A1* promoter, GAAGGAGGCGTGGCC, was the binding site of the KLF9 protein. In support of the above findings, we observed that KLF9 expression in pregnancy placental tissue was positively correlated with CYP1A1, which further supports that KLF9 regulates CYP1A1 expression in trophoblasts. Moreover, *KLF9* knockdown was able to reverse the oxidative stress and mitochondrial apoptosis caused by PM2.5, as well as the impaired proliferation, migration, invasion, and tube formation of trophoblasts. Indeed, KLF9 has been consistently reported to exacerbate oxidative stress damage in cells by disrupting ROS scavenging. For example, the knockdown of *KLF9* in cardiomyocytes protects them from ischemic injury (**Yan et al., 2019**), and a KLF9-dependent increase in ROS can result in cell death in lung tissues (**Zucker et al., 2014**) and induce proliferation of melanoma (**Bagati et al., 2019**). There are also numerous studies indicating the critical role of KLF9 in toxicological research. For example, Yue Gu et al. found that Klf9 is involved in BLM-induced pulmonary toxicity in human lung fibroblasts, Daqian Yang et al. identified that KLF9 was essential in allicin resisting against arsenic trioxide-Induced hepatotoxicity, but little is known regarding its role in the occurrence of PM2.5-induced toxicological processes. Here, our study not only identified KLF9 as a transcription factor of CYP1A1 for the first time in humans, but also revealed a novel mechanism of oxidative stress in trophoblast cells induced by PM2.5, providing a new target for future clinical treatment.

Due to the limitations of our laboratory, our current research unavoidably possesses certain deficiencies. For instance, the administration of intratracheal instillation may induce adverse effects on pregnant mice. In the future study, we will collaborate with other research institutions to employ meteorological and environmental animal exposure system that not only mitigate harm to pregnant

mice but also more realistically simulate the process of inhaling PM2.5 particles in humans. We also lack the utilization of primary cells in our in vitro experiments. Furthermore, our current article lacks investigations on the regulatory effect of PM2.5 on KLF9. These limitations should be further investigated in future.

In conclusion, we collected PM2.5 from the urban atmosphere of Jinan city and found that it caused various adverse gestational outcomes in mice as well as impaired placental structure and increased placental trophoblast apoptosis. In the trophoblast cell line HTR8/SVneo, PM2.5 induced oxidative stress damage and mitochondrial apoptosis, and affected cell proliferation, invasion, migration and angiogenesis. The KLF9/CYP1A1 transcriptional axis was involved in PM2.5-induced oxidative stress and mitochondrial apoptosis. This is not only the first study to demonstrate the molecular mechanism of PM2.5-induced oxidative stress damage in trophoblast cells, but also the first time to identify that KLF9 acts as a transcriptional factor positively modulating the expression of CYP1A1 in humans.

## Materials and methods
### PM2.5 collection, morphological characterization, and elemental composition examination

The PM2.5 high volume sampling system (Staplex PM2.5 SSI, USA) was installed near the Xincheng apartment complex, Jinan City, Shandong Province, China (**Figure 1A**). Structurally, the area was densely populated by heavy traffic, and surrounded by chemical factories, and cement plants. From October 2020 to April 2021, PM2.5 samples were collected on fiberglass fiber filters, and the filters were exchanged each 48 h. The filters were then cut into 1 cm$^2$ pieces, and 100 ml double distilled water was added for sonication. The suspension was purified by filtration through 8 layers gauze. And the filtrate was gathered in a 50 mL tube and lyophilized for 24 hr under vacuum. PM2.5 was stored at 4 °C. When PM2.5 were used in subsequent experiments, they were dissolved in PBS to prepare 5 mg/mL suspensions, and mixed thoroughly under ultrasonication for 20 min. The high-resolution SEM (JEOL JSM-6700F, Tokyo, Japan) attached with EDS (APOLLO XL, USA) was used to measure the size, shape, elemental composition and surface morphology of PM2.5 particles. Based on the manufacturer's guidelines, PM2.5 was suspended in a solution of n-hexane with the aid of ultrasonic treatment to obtain a uniformly distributed PM2.5 solution. Carbon coating was then performed and measurements were carried out using automatic mode. Analysis of the elemental composition of PM2.5 by scanning electron microscopy images of three randomly selected areas using EDS.

### Intratracheal instillation of PM2.5 in mice and specimen collection

Kunming pregnant mice (6–8 weeks old, weighing 50–56 g) were used in the present study. All animal experiments were permitted by the Research Ethics Committee approval of Maternal and Child Health Care Hospital of Shandong Province. Prior to the start of the study, mice were maintained in animal chambers under standard husbandry conditions for 1 week. The mice are housed in an environment which is guaranteed to be free of pathogens and with constant temperature and humidity. The study conformed to the principles for laboratory animal research and approved by the Maternal and Child Health Care Hospital of Shandong Province (permit NO. 2021–116). All pregnant mice were divided randomly into control and PM2.5-treated group. At 8 weeks, the average tidal volume of Kunming mice was about 0.25 mL, and the frequency of per mouse's respiratory was about 163 /min, thus, the total air intake per day was 0.25×163 × 60 mins×24 hrs=58,680 mL ≈ 0.0587 m$^3$/day (**Vermillion et al., 2018**). It has been reported that the PM2.5 exposure of pregnant women in Jinan in 2020 was about 64 µg/m$^3$ daily (**Wang et al., 2022**). Therefore, the total PM2.5 intake though out the whole pregnancy of mice was about 0.0587 m$^3$/day ×64 µg/m$^3$ × 20 days×100 (uncertainty factor)=7511 µg. The 100 fold uncertainty factor = 10 fold interspecies difference ×10 fold interindividual variation (**Dorne and Renwick, 2005**; **Zhang et al., 2017**). We weighed 7511 µg PM2.5 particles and dissolved it in 60 µL PBS buffer to prepare a PM2.5 suspension, which was subsequently divided into three equal portions. Pregnant mice were anaesthetized by intraperitoneal injection of 0.5% pentobarbital sodium (50 mg/kg) on 1.5 d, 7.5 d, and 12.5 d of pregnancy (corresponding to first, second and third trimester of human), followed by intratracheal instillation of 20 µL PM2.5 suspension, and the control group was intratracheally instilled with the same volume of PBS (n=8 for per group). The mice were euthanized by inhalation of 100% isoflurane on pregnancy 15.5 d and the placenta and pups were

extracted (*Figure 2A*). After weighing and counting the weight and number of mouse placentas and fetal mice, each mouse placenta was cut from the middle and divided into two parts, one was stored in 4% paraformaldehyde, and the other was frozen in liquid nitrogen and stored at –80 °C.

## Human placenta samples collection

The clinical specimens were collected from January 2020 to December 2021 at The Maternal and Child Healthcare Hospital of Shandong Province (*Supplementary file 1*). The study protocol complied with the ethical norms for the research of clinical specimens. The study protocol was approved by the Ethics Committee of Maternal and Child Healthcare Hospital of Shandong Province (permit NO. 2020–115).All pregnant women participating in the study signed an informed consent document. Placental samples were selected from 31 healthy pregnant women aged 22–33 years old. Immediately after clinical collection, the placental samples were obtained from the maternal side avoiding hemorrhaged, infarcted, or calcified tissues. The collected tissues were rinsed with cold PBS, then fixed with 4% paraformaldehyde, and finally paraffin-embedded and sectioned.

## Immunohistochemical staining of placental tissue

The placental tissue slices were dehydrated in a gradient of xylene and ethanol, and Hematoxylin eosin (HE) staining (Beyotime Institute of Biotechnology, China) was performed to detect histopathological changes, according to the manufacturer's instructions. The slices were dewaxed with xylene and hydrated using an ethanol gradient before being permeabilized with 0.5% Triton X-100.The slices were then blocked in 10% goat serum, and incubated with one of the primary antibodies listed in *Supplementary file 4* (1:100) overnight at 4 °C. The slices were incubated with secondary antibodies for 60 min at room temperature in the following day. Finally, the slices were washed with PBS and stained with DAPI (Beyotime Institute of Biotechnology, China). Representative images were captured using an upright fluorescence microscope (Olympus Corporation, Japan).

## Cell culture and PM2.5 treatment

The human trophoblast cell line HTR8/SVneo, originating from human placental trophoblast cells, was purchased from The American Type Culture Collection (CRL-3271, ATCC, USA). The HTR8/SVneo cells we used were identified by STR and tested negative for mycoplasma contamination. HTR8/SVneo was cultured in RPMI-1640 medium (Shanghai BasalMedia Technologies Co., Ltd, China) supplemented with 10% fetal bovine serum (Invitrogen; Thermo Fisher Scientific, Inc, USA), 1% penicillin and streptomycin (Beyotime Institute of Biotechnology, China), and 1% 100 mM sodium pyruvate (Beyotime Institute of Biotechnology, China). Subsequently, the cells were seeded in 6-well plates (Guangzhou Jet Biofiltration Co.,Ltd, China) with $3 \times 10^5$ cells/well, and different concentrations of PM2.5 (50 µg/mL, 100 µg/mL, 200 µg/mL) were added to the culture medium for 24 hr to construct a PM2.5 exposure cell model. N-acetyl-l-cysteine (NAC) (A7250, MillporeSigma, USA) was added to detect the rescue effect on oxidative stress caused by PM2.5 in HTR/SVneo cells.

## Cell counting Kit-8 (CCK8) assay

After cells were cultured with PM2.5 for 24 hr, 10 µL/well CCK-8 solution (E-CK-A362, Elabscience Biotechnology Co.,Ltd, China) was added and incubated for 2 h. Then, the absorbance was measured at 450 nm using a microplate reader (Norgen Biotek Corp., Canada).

## 5-Ethynyl-2'-deoxyuridine (EDU) assay

EDU assays were used to assess cell proliferation visually based on fluorescent staining. Cells were incubated with 1×Edu working solution for 2 hr at 37 °C (C0075, Beyotime Institute of Biotechnology, China); then 4% paraformaldehyde was added for 15 min by 0.3% Triton, both at room temperature. Subsequently, 0.5 mL Click reaction solution was added for 20 min. Finally, DAPI was added to stain the nuclei. After incubation in the dark for 15 min, the 96-well plate was imaged using HCA (high content analysis) to quantitatively assess proliferation.

## Apoptosis assay

According to the manufacturer's instructions (559763, BD Biosciences, Inc, USA),after suspending cells in 100 µL 1×binding buffer, Annexin V-FITC and PI were used to stain cells at room temperature

in the dark for 15 min. Finally, 300 µL of 1×binding buffer was added and the cells were analyzed using the FACSCalibur flow cytometer (BD Bio-sciences, USA) for detection.

## Wound-healing assay

After treatment under the different conditions, the cells were seeded (7×10⁴ cells/100 µL) into a 4-Well Culture-Insert (Ibidi GmbH, Germany). The culture inserts were gently removed after 12 hr. Afterwards, medium containing different concentrations of PM2.5 and no FBS was added. Representative images of cell migration were obtained by a microscope (4×objective) at 0 and 24 hr and the wound healing area was calculated by ImageJ (National Institutes of Health, Bethesda, USA) software.

## Invasion assay

For invasion assays, a Transwell insert (8 µm pores, Corning, Inc, USA) was placed in 24-well plates. 60 µL (1 mg/mL). Matrigel matrix (356234, Becton, Dickinson and Company, USA) was added to the upper chamber plate After the matrigel was solidified, 1×10⁵ cells were suspended in100µL medium without serum and seeded in the upper chamber. Then600 µL supplemented media was added to the bottom chamber. The cells and matrigel were removed from the chamber with a cotton swab after 12 hr cell culture at 37 °C, and each chamber were fixed with 4% paraformaldehyde and stained with 1% crystal viole. After washing, the representative images were captured by microscope, and counted the cells in five randomly selected fields of view.

## Tube formation assay

A total of 3×10⁴ cells were seeded into96-well plates which is coated with Matrigel. After 4 hr of incubation, 5 µM Calcein Acetoxymethyl Ester (C2012, Beyotime Institute of Biotechnology, China) was added to the plate and incubated for 15 min at 37 °C. Tube formation was observed by fluorescence microscope and the tube length was calculated by Image J.

## Intracellular reactive oxygen species (ROS) assay

Using 2', 7'-dichloro-dihydro-fluorescein diacetate (DCFH-DA) probe (C2938, Invitrogen; Thermo Fisher Scientific, Inc, USA) to detect intracellular ROS levels. The cells were seeded in a six-well plate (3×10⁵ cells/well). After treatment of cells as described above, 5 mM DCFH-DA was added for 30 min at 37 °C. The nuclei were counterstained with Hoechest (33258, Beyotime Institute of Biotechnology, China) at 37 °C for 15 min. Finally, representative images were captured using an upright fluorescence microscope (n=3 per group). Meanwhile, ROS levels were could also be detected by flow cytometry.

## Mitochondrial superoxide (MitoSOX) assay

Mitochondrial superoxide (MitoSOX) assays were conducted to detect the mitochondrial ROS levels. The cells were treated as required and subsequently seeded in 96-well plates with 8×10³ cells/well. After incubation for 12 hr at 37 °C, the cells were incubated in HBSS supplemented with 5 µM MitoSOX regent (M36008, Invitrogen; Thermo Fisher Scientific, Inc, USA) and 5 µM Mito-Tracker green regent (M7514, Invitrogen; Thermo Fisher Scientific, Inc, USA) for 30 min at 37 °C. Mito-Tracker green regent is often used as an intracellular mitochondria-specific fluorescent probe. Next, we removed the MitoSOX regent and stained the nuclei with Hoechst for 15 min at 37 °C. Finally, a fluorescence microscope was used to observe the fluorescence intensity of MitoSOX and image of the cells.

## Mitochondrial membrane potential (MMP) assay

JC-1 (5,5',6,6'-Tetrachloro-1,1',3,3'-tetraethyl-imidacarbocyanine iodide) probe (C2005, Beyotime Institute of Biotechnology, China) was used to detect mitochondrial membrane potential (MMP). The cells were seeded in a 96-well or 24-well plate. Cells were incubated in RPMI-1640 medium with 10 µM JC-1 probe for 15 min at 37 °C and then fluorescence intensity was observed by fluorescence microscopy.

## Lentiviral package and transduction

Lentiviruses were produced by transfecting HEK293T cells with two helper plasmids (psPAX2 and pMD2.G) and the pLVX-Puro vectors expressing KLF9 or CYP1A1 (performed by Tsingke Biotechnology Co., Ltd, China) (*Shuen et al., 2015*). The virus-containing medium was collected 48 hr and

72 hr after transfection, respectively, and filtered through a 0.45 µm membrane and mixed overnight at 4 °C with 5×PEG8000. After centrifugation at 5000 x *g* for 20 min, the supernatant was carefully removed, and the precipitated virus was resuspended in in pre-chilled PBS. Next, HTR8/SVneo cells were incubated in ix-well plates with 2 mL medium containing viral particles in the presence of 5 µg/mL polybrene. These cells were further cultured after 1 week with the addition of puromycin (starting from 0.5 µg/mL and increasing sequentially), and the HTR8/SVneo cell line with stable ectopic expression of KLF9 or CYP1A1 was screened.

siRNAs against *KLF9*, and *CYP1A1* were performed by Tsingke Biotechnology Co., Ltd., China. Detailed information about siRNA was listed in *Supplementary file 2*. For transient transfection, 70%–80% confluent cells were transfected with indicated siRNAs using Lipofectamine RNAi MAX (Thermo Fisher Scientific, 13778150) for 48 hr according to the manufacturer's instructions.

## Dual-luciferase reporter assay

Dual luciferase gene reporter analysis was performed using a plasmid in which the human *CYP1A1* promoter (TSS, –532 to +88) or its mutant sequence was inserted between the KpnI and XhoI restriction sites of the pGL3 basic expression vector (*Figure 9F*). Cells were transfected using Lipofectamine 3000 (L3000150, Thermo Fisher Scientific, Inc USA) according to the manufacturer's instructions. To correct for transfection efficiency, cells were co-transfected with the pRL-TK vector encoding Renilla luciferase and CYP1A1-pGL3-WT or the respective mutant plasmid. After 48 hr, firefly and Renilla luciferase activities in cell lysates were measured on a microplate luminometer using a dual luciferase reporter assay kit (E1910, Promega Corporation, USA) according to the manufacturer's protocol.

## High-throughput sequencing and bioinformatics analysis

Two groups of HTR8/SVneo cells were cultured in six-well plates exposed to PM2.5-free RPMI-1640 or PM2.5 100 µg/mL for 24 hr. To ensure the accuracy of data interpretation and analysis, three biological replicates were established for the control and treated cells. Total RNA was isolated and purified using TRIzol reagent (15596018, Invitrogen; Thermo Fisher Scientific, Inc, USA) according to the manufacturer's instructions. The sequencing was completed by Beijing Genomics Institute (BGI). The raw sequencing data was subjected to filtration using SOAPnuke in order to obtain clean reads. The clean reads were mapped to the reference genome using HISAT, and aligned to the assembled unique gene set using Bowtie2. Subsequently, the data analysis and mapping were performed utilizing the sequencing company's proprietary system (https://biosys.bgi.com). The gene expression was quantified utilizing the RNA-Seq by Expectation-Maximization (RSEM) algorithm, and differentially expressed genes (DEGs) were identified by the R-Bioconductor package DESeq2 with pre-set Q value<0.05 and |log2[Fold Change]|≥1. KEGG enrichment analysis of annotated DEGs was performed using phyper based on Hypergeometric test, and the significance values of pathways were strictly threshold corrected by Q values (Q values≤0.05). Gene Set Enrichment Analysis (GSEA) was performed on control and treatment groups based on KEGG database data. All parameters were kept at default settings, except for the maximum size of the filtering threshold which was adjusted to 5000. Gene features with FDR Q values≤0.25 were deemed statistically significant. The RNA sequencing data were deposited into the Gene Expression Omnibus (GEO) database (accession number: GSE237795).

## mRNA degradation assay

Actinomycin D (GC16866, Glpbio, USA) was used to detect mRNA degradation. HTR8/SVneo cells pre-exposed or not exposed to PM2.5 (100 µg/mL) were treated with 20 µM actinomycin D (RNA synthesis inhibitor) for 0, 3, 6, 9, 18, or 24 hr, respectively, and *CYP1A1* mRNA levels were measured by qRT-PCR. The half-life of mRNA was calculated by non-linearly fitting the relative expression of the time gradient by GraphPad Prism Version 8.0 (GraphPad Software, Inc).

## Chromatin immunoprecipitation (ChIP)

ChIP assays were performed on HTR8/SVneo cells stably overexpressing KLF9 using a ChIP assay kit (P2078, Beyotime Institute of Biotechnology, China) according to the manufacturer's instructions. HTR8/SVneo cells ($2×10^6$ cells) were cross-linked in 10 cm dishes, then the nuclear lysates were sonicated to shear DNA to approximately 200–300 bp fragments, followed by immunoprecipitation overnight at 4 °C using the anti-IgG or anti-KLF9 antibodies listed in *Supplementary file 4*. The

immunoprecipitated DNA was purified using a DNA purification kit (DP214, Tiangen Biotech Co.,Ltd, China) and then analyzed by PCR using the primers specified in *Supplementary file 3*.

### Tunel staining

TUNEL staining (E-CK-A320, Elabscience Biotechnology Co.,Ltd, China) was used to detect cell and tissue apoptosis. TPlacental tissue slices were dewaxed with xylene and hydrated using an ethanol gradient. According to the manufacturer's instructions, the proteinase K working solution and DNase I working solution were dropped onto the slices. After the addition of the TDT reaction solution, the cells were allowed to react at 37 °C for 60 min in the dark, and DAPI was added to stain the cell nuclei. Finally, staining was observed using a fluorescence microscope and imaged.

### Analysis of GSH, MDAcontent and SOD activity

Glutathione Assay Kit (MAK440, MilliporeSigma, USA), Mn-SOD Assay Kit with WST-8 (S0103, Beyotime Institute of Biotechnology, China), and MDA Assay Kit (S0131S, Beyotime Institute of Biotechnology, China) were used to analyze the glutathione (GSH), superoxide dismutase (SOD), and malondialdehyde (MDA) activity in cells, according to the manufacturer's protocol.

### Western blot assay

Western blotting was applied to detect the protein expression levels in cells or tissues, as described previously (*Li et al., 2022*). The antibodies used were listed in *Supplementary file 4*.

### Statistical analyses

The data were analyzed by GraphPad Prism version 8.0 (GraphPad Software, Inc, USA) and presented as the mean ± SEM. All experiments were repeated at least times. The differences between groups were compared by an unpaired two-tailed Student's t-test or an ANOVA. Correlation analysis was performed using Pearson's correlation analysis. Differences with a p value less than 0.05 were considered to indicate a statistically significant difference.

## Acknowledgements

The authors are grateful to members at Jinan Environmental Monitoring Center of Shandong Province for their help in providing PM2.5 samples. This work was supported by the foundation of National Natural Science Foundation of China (82301903), the foundation of the Maternal and Child Health Care Hospital of Shandong Province High-level talent incubation program (2022RS07) and the scientific research foundation of the Maternal and Child Health Care Hospital of Shandong Province (YJKY2022-024).

## Additional information

### Funding

| Funder | Grant reference number | Author |
| --- | --- | --- |
| The National Natural Science Foundation of China | 82301903 | Shuxian Li |
| Maternal and Child Health Care Hospital of Shandong Province | High-level talent incubation program (2022RS07) | Meihua Zhang |
| Maternal and Child Health Care Hospital of Shandong Province | YJKY2022-024 | Shuxian Li |

The funders had no role in study design, data collection and interpretation, or the decision to submit the work for publication.

## Author contributions
Shuxian Li, Conceptualization, Data curation, Writing – original draft, Funding acquisition; Lingbing Li, Formal analysis, Supervision; Changqing Zhang, Writing – review and editing; Huaxuan Fu, Resources; Shuping Yu, Data curation; Meijuan Zhou, Zhenya Fang, Validation; Junjun Guo, Software; Anna Li, Investigation; Man Zhao, Formal analysis; Meihua Zhang, Supervision, Funding acquisition, Validation; Xietong Wang, Project administration, Supervision

## Author ORCIDs
Shuxian Li ⓘ https://orcid.org/0000-0003-2601-6126
Lingbing Li ⓘ https://orcid.org/0000-0002-0640-8328
Xietong Wang ⓘ https://orcid.org/0000-0002-8811-3792

## Ethics
Human subjects: The qualification and experience of researcher meet the test requirements; the research project is accordance with the scientific and ethical principles; the method of obtaining informed consent is right. The study was approved by the Research Ethics Committee approval of Maternal and Child Health Care Hospital of Shandong Province Approval Number: 2020-115.

The experimental design is in accordance with the principles of animal protection, experimental animal welfare ethics and other ethical requirements, Applicants are committed to abide by the relevant experimental animal ethics, and accept the supervision and inspection of the Committee at any time. Experiment related personnel qualification and experiment related units are appropriate. Varieties, quality grade and specifications of animals used in experiments are appropriate. Research Ethics Committee approval of Maternal and Child Health Care Hospital of Shandong Province Approval Number: 2021-116.

## Decision letter and Author response
Decision letter https://doi.org/10.7554/eLife.85944.sa1
Author response https://doi.org/10.7554/eLife.85944.sa2

---

# Additional files

## Supplementary files
- Supplementary file 1. The gestation and characteristics of the pregnant women (n=31).
- Supplementary file 2. The sequence of siRNAs used in this study.
- Supplementary file 3. The sequence of primers used in this study.
- Supplementary file 4. The information of antibodies used in this study.
- MDAR checklist

## Data availability
The RNA sequencing data were deposited into the Gene Expression Omnibus (GEO) database (accession number: GSE237795).

The following dataset was generated:

| Author(s) | Year | Dataset title | Dataset URL | Database and Identifier |
|---|---|---|---|---|
| Shuxian L, Lingbing L, Meihua Z, Xietong W | 2023 | PM2.5 leads to adverse pregnancy outcomes by inducing trophoblast oxidative stress and mitochondrial apoptosis via KLF9/CYP1A1 transcriptional axis | https://www.ncbi.nlm.nih.gov/geo/query/acc.cgi?acc=GSE237795 | NCBI Gene Expression Omnibus, GSE237795 |

---

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

# Appendix 1

## Appendix 1—key resources table

| Reagent type (species) or resource | Designation | Source or reference | Identifiers | Additional information |
| --- | --- | --- | --- | --- |
| Gene (*Homo sapiens*) | CYP1A1 | GenBank | HGNC:HGNC:2595 | |
| Gene (*Homo sapiens*) | KLF9 | GenBank | HGNC:HGNC:1123 | |
| Antibody | anti-CYP1A1 (Rabbit polyclonal) | Proteintech | Cat#13241–1-AP | WB (1:1000) IHC (1:500) |
| Antibody | anti-KLF9 (Rabbit polyclonal) | Abcam | Cat#ab227920 | WB (1:1000) IHC (1:500) ChIP (1:1000) |
| Antibody | anti-HO-1 (Rabbit polyclonal) | Cohesion Biosciences | Cat#CQA2561 | WB (1:1000) |
| Antibody | anti-NQO-1 (Rabbit polyclonal) | Cohesion Biosciences | Cat#CPA1342 | WB (1:1000) |
| Antibody | anti-GCLC (Rabbit polyclonal) | Cohesion Biosciences | Cat#CPA2092 | WB (1:1000) |
| Antibody | anti-SOD-1 (Rabbit polyclonal) | Cohesion Biosciences | Cat#CPA1476 | WB (1:1000) |
| Antibody | anti-CK-7 (Rabbit monoclonal) | Abcam | Cat#ab68459 | IHC (1:500) |
| Antibody | anti-β-actin (Mouse monoclonal) | Proteintech | Cat#66009 | WB (1:1000) |
| Antibody | anti-Lamin B | Proteintech | Cat#12987–1-AP | WB (1:1000) |
| Cell line (*Homo sapiens*) | HTR8-SVneo (*Homo sapiens*) | ATCC | NO.CRL-3271 | |
| Sequence-based reagent | siCYP1A1#1_F | This paper | siRNA sequence | GGUAUGUGGUGGUAUCAGUTT |
| Sequence-based reagent | siCYP1A1#1_R | This paper | siRNA sequence | ACUGAUACCACCACAUACCTT |
| Sequence-based reagent | siCYP1A1#2_F | This paper | siRNA sequence | CCUUCAAGGACCUGAAUGATT |
| Sequence-based reagent | siCYP1A1#2_R | This paper | siRNA sequence | UCAUUCAGGUCCUUGAAGGTT |
| Sequence-based reagent | siKLF9#1_F | This paper | siRNA sequence | GCCCAUUACAGAGUGCAUATT |
| Sequence-based reagent | siKLF9#1_R | This paper | siRNA sequence | UAUGCACUCUGUAAUGGGCTT |
| Sequence-based reagent | siKLF9#2_F | This paper | siRNA sequence | GGAGUGACCACCUCACAAATT |
| Sequence-based reagent | siKLF9#2_R | This paper | siRNA sequence | UUUGUGAGGUGGUCACUCCTT |
| Sequence-based reagent | CYP1A1_F | This paper | PCR primers | TGGCATCCTCTACAGACTCCTG |
| Sequence-based reagent | CYP1A1_R | This paper | PCR primers | CTTCAGGTTGCGTGCCATCTCA |
| Sequence-based reagent | KLF9_F | This paper | PCR primers | CTACAGTGGCTGTGGGAAAGTC |
| Sequence-based reagent | KLF9_R | This paper | PCR primers | CTCGTCTGAGCGGGAGAACTTT |

*Appendix 1 Continued on next page*

*Appendix 1 Continued*

| Reagent type (species) or resource | Designation | Source or reference | Identifiers | Additional information |
|---|---|---|---|---|
| Sequence-based reagent | CYP1A1-promoter_F | This paper | PCR primers | CTGCTTCTCCCTCCATCT |
| Sequence-based reagent | CYP1A1-promoter _R | This paper | PCR primers | GGAACTGTCACCTTCAGG |
| Commercial assay or kit | GSH | MilliporeSigma | MAK440 | |
| Commercial assay or kit | SOD | Beyotime Institute of Biotechnology, China | S0103 | |
| Commercial assay or kit | MDA | Beyotime Institute of Biotechnology, China | S0131S | |
| Commercial assay or kit | EDU | Beyotime Institute of Biotechnology, China | C0075 | |
| Commercial assay or kit | Apoptosis kit | BD Biosciences | 559763 | |
| Commercial assay or kit | TUNEL staining | Elabscience Biotechnology | E-CK-A320 | |
| Commercial assay or kit | ChIP assay kit | Beyotime Institute of Biotechnology | P2078 | |
| Commercial assay or kit | Dual luciferase reporter assay kit | Promega Corporation | E1910 | |
| Recombinant DNA reagent | pLVX-Puro (plasmid) | purchased from Tsingke biotechnology | | plasmid sequences were verified by Tsingke biotechnology |
| Recombinant DNA reagent | pLVX-CYP1A1- 3×FLAG (plasmid) | purchased from Tsingke biotechnology | | plasmid sequences were verified by Tsingke biotechnology |
| Recombinant DNA reagent | pLVX-KLF9- 6×His (plasmid) | purchased from Tsingke biotechnology | | plasmid sequences were verified by Tsingke biotechnology |
| Chemical compound, drug | NAC | MillporeSigma | A7250 | |
| Chemical compound, drug | JC-1 | Beyotime Institute of Biotechnology | C2005 | |
| Chemical compound, drug | Actinomycin D | Beyotime Institute of Biotechnology | GC16866 | |
| Software, algorithm | Image J | National Institutes of Health | V 1.8.0 | |
| Software, algorithm | GraphPad | GraphPad Software | V 8.0 | |

