## [Editor Report]

This study offers a valuable finding using a mouse model exposed to PM2.5 samples collected from highly polluted city air. The solid evidence provided strongly supports the assertions of the authors that PM2.5 triggers a KLF9/CYP1A1 signaling pathway, resulting in placental dysfunction, oxidative stress, mitochondrial issues, and adverse gestational outcomes. This research holds substantial relevance for medical biologists engaged in studying environmental factors impacting maternal and fetal health.

---

## [Decision Letter]

**Decision letter after peer review:**

Thank you for submitting your article "PM2.5 leads to adverse pregnancy outcomes by inducing trophoblast oxidative stress and mitochondrial apoptosis via KLF9/CYP1A1 transcriptional axis" for consideration by *eLife*. Your article has been reviewed by 3 peer reviewers, one of whom is a member of our Board of Reviewing Editors, and the evaluation has been overseen by Diane Harper as the Senior Editor.

Essential revisions:

1) One concern from this study is regarding the experimental design, particularly the dosage of PM2.5 used in this paper, as well as the timing and frequency of administration of this compound. The authors should perform further experiments at a lower dose or administration frequency to represent a more physiologically relevant scenario.

2) The metformin data lack rigour, and the authors would remove them and use these data to focus on a new study.

3) There are several concerns regarding the figures accompanying the results, including some that are not self-explanatory.

*Reviewer #1 (Recommendations for the authors):*

Air pollutant such as particulate matter PM2.5 is considered one of the most severe toxic associated with various adverse pregnancy outcomes. Most present research on PM2.5 in adverse effects on human pregnancy is focused on the epidemiological aspects, remaining partially unraveling the underlying molecular mechanisms. In this research, Zhang Wang and colleagues used PM2.5 collected from the urban region of Jinan, China, to establish pollutant exposure experiments by in vivo animal models and in vitro trophoblast cells. All experiments were evaluated by several in vitro techniques, including RNAseq analysis and in silico exploration of differentially expressed genes to infer the effect of PM2.5 on the mice pregnancy and human trophoblast cells.

One caution of these studies is the dosage calculations for the in vivo and in vitro experimental design to evaluate the effect of PM 2.5. In the mice model, they administered doses of daily PM2.5 inhalation several times higher than a pregnant woman in Jinan might experience. On the other hand, an equivalent high dosage was applied to trophoblasts to extract total RNA and perform the RNAseq. However, the in silico data analysis only detected 32 differentially expressed genes using a Log2FC of no more than 1, denoting the subtle effect between the control and the treatments.

Despite this, through in vivo experiments, the authors show that the administration of PM2.5 by intratracheal route leads to significant changes in the pregnancy of mice, significantly increased fetal mortality, decreased fetal weight and number, and damaged placental structure. The in vitro experiments showed increased apoptosis in trophoblastic line HTR8/SVneo cells.

Through RNAseq experiments, the authors were able to infer two genes, KLF9 and CYP1A1, from the set of genes differentially expressed between trophoblast cells treated with 100 ug of PM2.5 compared to untreated controls. In addition, the authors demonstrated by in vitro ChIP assay the binding of KLF9 to the CYP1A1 promoter, proving that KLF9 is a positive modulator of CYP1A1 expression triggered by PM2.5. Consequently, the authors suggested activating this KLF9/CYP1A1 pathway promotes the empirically observed effects of oxidative stress and cell death in trophoblasts and the placenta of pregnant mice. But this study has not explored the mechanisms by which PM2.5 activates KLF9.

By additional experiments at the end of this work, the authors found that a single empirically deduced dose of metformin can reverse the toxic effects of PM2.5 on trophoblast cells of the HTR8/SVneo lineage. They observed that cells treated with metformin resulted in reduced expression of KLF9 and CYP1A1 compared to PM2.5-exposed cells. However, this experiment shows limitations, such as the single dose of the inhibitory drug used and a bias in the genetic mechanisms by focusing only on the two genes, KLF9 and CP1A1.

This study used PM2.5 collected from the urban region of Jinan, China, to establish pollutant exposure experiments by in vivo animal models and trophoblast cells. This relevant study applies several in vitro and in silico techniques from these models. The authors identify that PM2.5 activates the KLF9/CYP1A1 signaling pathway causing oxidative stress damage with mitochondrial apoptosis, which can be correlated to poor pregnancy outcomes observed in mice.

I have the following comments:

1. The in vivo experimental designs of PM 2.5 dosage calculations are not unreliable to me. Because the in vitro and in vivo experiments were administered doses of daily PM2.5 inhalation huge times higher (2.503 ug) than a pregnant woman in Jinan might experience (3.6 ug). The inferences from this experimental design so different from the natural investigation in humans should be better explained. Also, it should be justified in comparison with some experiments with the proper doses experienced by pregnant women in Jinan.

On the other hand, the dosage applied to trophoblasts for the extraction of total RNA and consequent RNAseq only detected 32 differentially expressed genes by using a Log2FC of 1, denoting the subtle effect between the control and the treatments. Under natural exposure conditions of daily PM2.5 (3.6 ug) instead of the current in vitro design (100 ug), it would be interesting to explore the gene expression results. Authors should discuss all these experimental limitations more rigorously.

2. Regarding the English written in the manuscript, I found several misspelled words, such as typo errors. Also, the authors need to check a few wrong or missing prepositions, punctuation, and the agreement between subject and verb in some sentences. Furthermore, please check words in the original language?

3. In the manuscript, beginning with the abstract, it would be essential to change the meaning of the inferences obtained from the analysis of the RNAseq experiments. Since this type of in vitro experiment is only exploratory, it then raises hypotheses that must be confirmed a posteriori through other in vivo experiments. In the abstract the current text reads: "we comprehensively analyzed the transcriptional landscape of HTR8/SVneo cells exposed to PM2.5 through RNA-Seq and confirmed that PM2.5 triggered oxidative stress and mitochondrial apoptosis to damage HTR8/SVneo cell biological functions through CYP1A1." The meaning could be changed by using this statement "we comprehensively analyzed the transcriptional landscape of HTR8/SVneo cells exposed to PM2.5 through RNA-Seq and observed that PM2.5 triggered overexpression of pathways involved in oxidative stress and mitochondrial apoptosis to damage HTR8/SVneo cell biological functions through CYP1A1." Or "we comprehensively analyzed the transcriptional landscape of HTR8/SVneo cells exposed to PM2.5 through RNA-Seq and confirmed by validation tests that PM2.5 triggered oxidative stress and mitochondrial apoptosis to damage HTR8/SVneo cell biological functions through CYP1A1."

4. The methodology for the bioinformatics analysis of the RNA-seq experiments needs to be better described. The authors should better detail the data pre-processing steps (to get quality control), including the software and the parameters applied until achieved at Differentially Expressed Genes (DEGs), such as read alignments, counts, normalization, and DEGs.

Although the analyses of the KEGG pathway enrichment analysis of differentially expressed genes and Gene Set Enrichment Analysis (GSEA) were carried out in a particular company, it is also desirable that the details of the analyses are described in the methodology with their respective parameters.

5. Some bioinformatics analyses deserve to be better explained. This is because this manuscript reports an interdisciplinary work, and it is expected that non-experts in bioinformatics can understand the findings and correlations between the different results. For example, explain in more detail the analyses and how to interpret each result shown in figure 4, from A to H, particularly curves and cutoff values.

6. Figures:

6.1. I feel that the figures presented accompanying the results are not self-explanatory. There are too many tags that the non-expert reader cannot understand because they have not dominated the meaning. To cite just one example here: Figure 9D: si-NC – si-KLF9#1 – si-KLF9#2. The authors need to put the meaning of the figures' tags in the respective captions. Remembering again that this is an interdisciplinary paper.

6.2. There is a critical error in Figure 4E, where the authors show the Z-score scale in the heat map. While in the manuscript text they describe that 24 genes were up-regulated and eight genes down-regulated in the PM 2.5 samples, in Figure 4E, we see the opposite. In Figure 4E, we see 24 genes in red that represent an expression level lower than the mean, whereas eight green genes that represent an expression level above the mean related to PM 2.5 samples compared to control samples. Please check this figure and identify and correct this critical error.

6.3. Figure 9G is very confusing to me. This figure contains a left and right panel that the reader must interpret as correlating. The left panel shows the CYP1A1 promoter region with several deletions that either resulted in failure of binding by FT KLF9 or not. Therefore, in the right panel, the reader would expect to see which mutations prevent FT KLF9-mediated CY1A1 expression. However, in Figure 9G on the right panel, the authors showed the expression of KLF9 or control. So, in this scenario, the picture gets confused. I suggest that the authors correct this figure to clarify this experiment's results.

Also, in the explanation of this figure 9, there is a mistake in the sense of writing, e.g., "These results suggested that a response element of the CYP1A1 promoter located in the -64 bp to -49 bp region, GAAGGAGGCGTGGCC, was required for the transcription of KLF9." A change to the following is recommended: "…, was required for the transcription of CYP1A1 meditated by KLF9."

6.4. Figures 9 H and I do not show what the authors claim in the text: "The chromatin was precipitated with specific antibodies for KLF9 or IgG, and PCR analysis with the indicated primers showed that KLF9 was able to bind directly to the CYP1A1 promoter (Figure 9H and I)."

The authors need to show the figure that corresponds with this statement. It would be interesting to perform the ChIP analysis experiment using wild-type and mutant constructs of the CY1A1 promoter, confirming the results of the region of this promoter that effectively binds KLF9.

7. Discussion: related to this sentence "To our knowledge, this study is the first to report that PM2.5 caused mitochondrial apoptosis via inducing oxidative stress, which in-turn impaired a series of biological functions such as invasion, migration, and angiogenesis in placental trophoblasts." Actually, this one paper Front. Endocrinol., 12 March 2020 Sec. Translational Endocrinology Volume 11 – 2020 https://doi.org/10.3389/fendo.2020.00075. Please double check, since they also described oxidative damage in trophoblasts and correct corresponding.

8. Inferences report to metformin:

Inferences about the application of the drug metformin were raised from experiments testing a single dose at a high level (20 mM). The authors only concentrated on the target genes highlighted in this work (CYP1A1 and KLF9).

I suggest that metformin evaluation should be better explored in another paper. In this scenario, the results are not very robust and complicate the focus of this paper, which turned out to be very large. Metformin could be explored by tests using more treatments and even performing new RNAseq experiments on trophoblast cells exposed to the drug at different doses.

*Reviewer #2 (Recommendations for the authors):*

Epidemiological studies have linked an increase in air pollutants, including fine particulate matter (PM2.5) to adverse pregnancy and postnatal outcomes. However, the molecular details of this are unclear, partially because there is no established mouse model in which to investigate the effects of PM2.5 on mammalian pregnancy. Li and Li et al. begin this study with the collection and characterization of PM2.5 from a high-volume sampling system set up in Jinan City, China. Upon collection of this material, the authors began their study using scanning electron microscopy and elemental analysis to define the properties of PM2.5. The authors use this material to establish a mouse model and cellular system to test the molecular effects of PM2.5 on pregnancy, postnatal mouse health, and trophoblast cells. One caveat of these studies is the dosage of PM2.5 given to the mice; the authors calculate dosages based on their estimation of PM2.5 exposure of pregnant women in Jinan in 2020, but include a 100-fold "uncertainty factor," which substantially increases the amount of PM2.5 given to the mice in their model system. This, along with the multiple timepoints at which PM2.5 is administered, suggests that the authors may be dosing mice with up to 400 times higher levels of PM2.5 than humans experience, on average, based on their own calculations. It is unclear if more physiologically relevant doses of PM2.5 (less the uncertainty factor) would have such stark effects on pregnancy outcomes as are shown in this study.

Nonetheless, the authors show that administration of PM2.5 through intratracheal inhalation leads to increased fetal mortality, decreased fetal weight and number, damaged placental structure, and increased trophoblast apoptosis in mice and in isolated trophoblast cells. To investigate the molecular mechanisms underlying these phenotypes, the authors performed RNAseq on PM2.5-treated trophoblasts and identify two genes, CYP1A1 and KLF9, that are transcriptionally upregulated upon PM2.5 exposure. The authors find that genetic ablation of CYP1A1 diminishes oxidative stress and intrinsic cell death in PM2.5-treated cells, suggesting upregulation of this cytochrome P450 family member at least partially drives toxicity in this model. The authors then link KLF9, a transcription factor also upregulated in PM2.5-treated trophoblasts, to CYP1A1 expression, mapping the KLF9 responsive element in the CYP1A1 promoter. These data suggest that PM2.5 exposure induces KLF9-mediated transcription of CYP1A1 to promote oxidative stress and cell death in trophoblasts and in the placenta of pregnant mice. In a final set of experiments, the authors find that a relatively high dose of 20 mM metformin can ameliorate some toxic effects of PM2.5 in trophoblast cells, and that this correlates with a decrease in PM2.5-induced CYP1A1 and KLF9 expression.

Collectively, the data presented by the authors convincingly demonstrate that: 1. The chosen dose of PM2.5 administered to mice causes significant increases in fetal mortality and morbidity; 2. Treatment of trophoblasts with PM2.5 causes oxidative stress, increases in apoptosis, and results in the transcriptional upregulation of CYP1A1 and KLF9; 3. KLF9 binds to the CYP1A1 promoter, driving its expression upon PM2.5 exposure, and that 4. CYP1A1 upregulation is necessary for PM2.5-induced oxidative stress. The authors also claim that "metformin could eliminate the toxicity induced by PM2.5 via the KLF9/CYP1A1 transcriptional regulatory axis" in their model trophoblast cell line, although this link is correlative as presented within the current study. While the data strongly implicate KLF9 and CYP1A1 to PM2.5-mediated toxicity, the mechanisms by which KLF9 senses PM2.5 were not established in this study, nor were the effects of KLF9 on gene products other than CYP1A1, which may also contribute to the phenotypes seen within this mouse model. Finally, though the reasoning for testing metformin in this disease model is not fully clear, there does seem to be therapeutic benefit at high doses of metformin to ameliorate phenotypes associated with PM2.5-mediated toxicity.

This is an important study on the effects of PM2.5 on pregnancy outcomes in mice. Overall, the authors use multiple independent systems to test the effects of PM2.5 in both mice and in trophoblast cells, adding rigor to the approach. The authors put forth a number of claims in the abstract, most of which are well justified:

1. "…PM2.5 induced adverse gestational outcomes such as increased fetal mortality rates, decreased fetal number and weight, damaged placental structure, and increased apoptosis of trophoblasts."

2. "…PM2.5 induced dysfunction of the trophoblast cell line HTR8/SVneo, including its proliferation, apoptosis, invasion, migration, and angiogenesis."

3. "…PM2.5 triggered oxidative stress and mitochondrial apoptosis to damage HTR8/SVneo cell biological functions through CYP1A1."

4. "…PM2.5 stimulated KLF9, a transcription factor identified as binding to CYP1A1 promoter region."

5. "…metformin could eliminate the toxicity induced by PM2.5 via the KLF9/CYP1A1 transcriptional regulatory axis in HTR8/SVneo."

The first four of these claims are well justified. The data presented may support the last claim on metformin treatment, but the data supporting this are correlative (see revision suggestions below). While the data collected support the hypothesis that PM2.5 leads to oxidative stress via the upregulation of KLF9 and CYP1A1, I have four main concerns that the authors could address to strengthen this work, which I outline below.

1. I am concerned with the dosages of particulate matter administered to the mice and cell models, and that these exceed physiologically relevant doses based on the calculations the authors provide in the methods section. The authors calculate a single dose of administration based on the average daily PM2.5 inhalation rate, and multiply it by 20 to reflect the full-term pregnancy in a mouse. However, the authors then use a 100-fold 'uncertainty factor' and administer this dose four times, despite the fact that the full pregnancy duration had already been factored into their calculations. Thus, one dose is ~100x the full exposure of PM2.5, but the authors administering this four times increases the potential exposure to ~400x what a pregnant woman in Jinan may experience. It is not hard to imagine that an excess dose may cause more severe pathologies than what women actually experience through PM2.5 inhalation. The authors should explain and justify why they used such high doses based on their own calculations and should repeat some key experiments with lower doses of PM2.5 to reflect physiologically relevant exposures.

2. The metformin data at the end of the manuscript are unnecessary, and I believe they should be removed from the manuscript. While the authors demonstrate that a single high dose (20 mM) metformin can alleviate select phenotypes induced by PM2.5, these data are not rigorous enough to justify the use of metformin to alleviate PM2.5 toxicity in mice. Collection of such data would be beyond the scope of this manuscript, as it is already 11 figures worth of data. Furthermore, the authors suggest that metformin acts through the CYP1A1/KLF9 signaling axis, but they only show correlative data that these two proteins change at one dose of metformin administered for one timepoint. While these two key PM2.5 targets do change in response to metformin, significantly more data would need to be collected to demonstrate that these targets are required for this response. Finally, the justification for looking at metformin is obscure, and it is unclear why the authors chose this compound other than their statement that metformin is a 'miracle' drug. Collectively, the lack of rationale, effects only at a single high dose, and correlative nature of involvement with the KLF9/CYP1A1 pathway leave this line of investigation with insufficient rigor. As a substantial amount of work would be required to alleviate this, it would be advisable for the authors to remove this portion of the manuscript so as to be able to more fully characterize these phenotypes in a future study.

3. Some data would be more powerful if shown in a more quantitative manner. This includes the MitoSOX microscopy (Figures 5G, 7I, 10G) and the JC-1 microscopy (Figures 5H, 7J, 10F). There are also portions of the manuscript in which the authors claim "significant" changes in their data, but these experiments are not accompanied by statistical analysis (Figure 7B… "The results showed that, compared to the control group, the expression of CYP1A1 was significantly increased by PM2.5 exposure.") The authors should change the verbiage (e.g., 'substantially') or perform the relevant statistical analysis (which would be preferred).

4. The paper is too long, particularly in the discussion and conclusions (8 pages). The manuscript would be better if it were shortened.

*Reviewer #3 (Recommendations for the authors):*

This article investigated the PM2.5-induced toxicity on pregnancy outcomes and trophoblasts and revealed a novel mechanism by which PM2.5 caused trophoblast mitochondrial damage through KLF9/CYP1A1 transcriptional axis. Furthermore, this was the first time to identify KLF9 acted as the transcription factor positively modulating CYP1A1 in humans. This article also suggested a potential therapeutic effect of metformin. Overall, the work is logical and well supported by the data organized, and sufficiently innovative.

I have a few suggestions to improve the manuscript, please see below.

1. The authors constructed a PM2.5-exposed pregnant mice model by intratracheal instillation. The concentrations setting of PM2.5 were described in detail, but the selection of PM2.5 treatment time points (1.5 d, 7.5 d, and 12.5 d of pregnancy) was not explained. The authors need to describe the reason why these time points were chosen. In addition, the intratracheal instillation did not simulate PM2.5 environmental exposure properly, could the authors construct the animal model of PM2.5 exposure using a meteorological and environmental animal exposure system? (Ran Z, An Y, Zhou J, Yang J, Zhang Y, Yang J, et al. Subchronic exposure to concentrated ambient PM2.5 perturbs gut and lung microbiota as well as metabolic profiles in mice. Environ Pollut 2021, 272: 115987)

2. In result 3.9, the authors concluded that metformin reversed oxidative stress damage caused by PM2.5 through the KLF9/CYP1A1 transcriptional axis, but there were no results on the impact of metformin on the transcriptional level of CYP1A1. The authors should provide additional experiments to illustrate this issue and also add pertinent content to the discussion.

3. As the authors described in the paper, trophoblast biological functions such as invasion, migration, and tube formation were essential for placental development. The authors did not conduct relevant studies exploring the toxic effects of metformin in protecting trophoblast cells from PM2.5. The authors should perform further experiments to optimize the effects of metformin.

4. KLF9 is the focus of this manuscript to explore the toxic effects of PM2.5, but there are not many studies describing the role of KLF9 in toxicology in the Discussion section. For example, YUE GU et al. found that Klf9 is involved in BLM-induced pulmonary toxicity in human lung fibroblasts (Gu Y, Wu YB, Wang LH, Yin JN. Involvement of Kruppel-like factor 9 in bleomycin-induced pulmonary toxicity. Mol Med Rep 2015, 12(4): 5262-5266.) and Daqian Yang et al. identified that KLF9 was essential in allicin resisting against arsenic trioxide-Induced hepatotoxicity (Yang D, Lv Z, Zhang H, Liu B, Jiang H, Tan X, et al. Activation of the Nrf2 Signaling Pathway Involving KLF9 Plays a Critical Role in Allicin Resisting Against Arsenic Trioxide-Induced Hepatotoxicity in Rats. Biol Trace Elem Res 2017, 176(1): 192-200). Previously published evidence of KLF9-dependent toxicological responses needs to be cited more clearly in the manuscript.

5. Is it possible to evaluate the role of KLF9 in placental toxicity caused by PM2.5 using knockout mice models?

6. The authors used only trophoblast cell line HTR8/SVneo for in vitro experiments. Could it be possible to add trophoblast primary cells or other primary cells such as HUVEC? The authors should mention this and refer to this point in the manuscript.

7. In result 3.8, the authors described "Cellular immunofluorescence showed that the increase in KLF9 expression was primarily observed in the nucleus (Figure 10D)". The authors assumed that KLF9 functioned as a transcription factor through nuclear translocation. To better corroborate this conclusion, I suggest that the authors should extract nuclear protein and detect the expression of KLF9 quantitatively.

8. AHR has been recognized as a receptor of environmental pollutants and a mediator of chemical toxicity including PM2.5. Meanwhile, AHR is an essential ligand-activated transcription factor of CYP1A1, which is also mentioned in the discussion of the manuscript. I am interested in the expression of AHR in trophoblast cells under PM2.5 exposure, does PM2.5 also increase AHR expression? Is it possible for the authors to conduct research on the role of PM2.5 on AHR?

[Editors’ note: further revisions were suggested prior to acceptance, as described below.]

Thank you for resubmitting your work entitled "PM2.5 leads to adverse pregnancy outcomes by inducing trophoblast oxidative stress and mitochondrial apoptosis via KLF9/CYP1A1 transcriptional axis" for further consideration by *eLife*. Your revised article has been evaluated by Diane Harper (Senior Editor) and a Reviewing Editor.

The manuscript has been improved, but there are some remaining issues that need to be addressed, as outlined below:

To ensure transparency and reproducibility, we kindly request the supplemental material containing the complete dataset of the RNA-Seq analysis, including the total of 17,795 genes identified and the corresponding statistical parameters.

*Reviewer #1 (Recommendations for the authors):*

This study used PM2.5 collected from the urban region of Jinan, China, to establish pollutant exposure experiments by in vivo animal models and trophoblast cells. This relevant study applies several in vitro and in silico techniques from these models. The authors identify that PM2.5 activates the KLF9/CYP1A1 signaling pathway causing oxidative stress damage with mitochondrial apoptosis, which can be correlated to poor pregnancy outcomes observed in mice.

In general, the authors have adequately addressed the primary issues I raised in this revised version.

However, the importance of the authors providing a supplementary Table containing the complete dataset of the RNA-Seq analysis is necessary. In the current scientific landscape, transparency and reproducibility of experiments are vital principles that promote the advancement of knowledge and enable the scientific community to build upon previous findings.

By providing the entire dataset, including the total of 17,795 genes identified in the sequencing, along with the corresponding statistical parameters of the differential expression analysis (DEG) such as P-value, FDR, and Log2FC, you will contribute significantly to the transparency and reproducibility of your study. This will enable other researchers to validate and replicate your findings, facilitating scientific progress.

---

## [Author Response]

Essential revisions:1) One concern from this study is regarding the experimental design, particularly the dosage of PM2.5 used in this paper, as well as the timing and frequency of administration of this compound. The authors should perform further experiments at a lower dose or administration frequency to represent a more physiologically relevant scenario.

Thank you very much for your advice on our in vivo experiments. We apologize for the simplicity of the writing in our previous manuscript, which did not clearly explain the dosage, timing and frequency in our experimental design. Please allow me to provide detailed explanation of each aspect as follows.

Firstly, regarding the dosage of PM2.5. We collected the PM2.5 particles from Jinan, China, so we combined PM2.5 exposure level of pregnant women in Jinan with the physiological indicators of mice, multiplied by 100-fold uncertainty factor to set the dosage in our in vivo experiments. To make it easier to understand, we have revised the PM2.5 dosage calculation process in detail in the method section:

“At 8 weeks, the average tidal volume of Kunming mice was about 0.25 mL, and the frequency of per mouse’s respiratory was about 163/min, thus, the total air intake per day was 0.25 x 163 x 60 mins x 24 hrs = 58680 mL ≈ 0.0587 m^3^/day[41]. It has been reported that the PM2.5 exposure of pregnant women in Jinan in 2020 was about 64 μg/m^3^ daily [40]. Therefore, the total PM2.5 intake though out the whole pregnancy of mice was about 0.0587 m^3^/day × 64 μg/m^3^ × 20 days × 100 (uncertainty factor) = 7511 μg. The 100 fold uncertainty factor = 10-fold interspecies difference ×10-fold interindividual variation [95] [60]. We weighed 7511 μg PM2.5 particles and dissolved it in 60 μL PBS buffer to prepare a PM2.5 suspension, which was subsequently divided into three equal portions. Pregnant mice were anaesthetized by intraperitoneal injection of 0.5% pentobarbital sodium (50 mg/kg) on 1.5 d, 7.5 d, and 12.5 d of pregnancy (corresponding to first, second and third trimester of human), followed by intratracheal instillation of 20 μL PM2.5 suspension, and the control group was intratracheally instilled with the same volume of PBS (n = 8 for per group).” (P24 L563-572).

The calculation of our PM2.5 dosage is as same as many authoritative studies related PM2.5, such as Chen Q et al. J Exp Clin Cancer Res 2022; Li Y et al. Ecotoxicol Environ Saf, 2021; Li J et al. Sci Total Environ, 2020; Zhang J et al. Sci Total Environ 2018; Zhang Y et al. Sci Total Environ 2017. In these studies, the dosage of PM2.5 in the in vivo experiments was also determined by combining the ambient PM2.5 level with the physiological indicators of mice, multiplying by 100 fold uncertainty factor. The only difference of the dosage between these study and our manuscript is the ambient PM2.5 exposure level. They employed ambient PM2.5 levels recommended by WHO, and we employed ambient PM2.5 level of pregnant women in Jinan area (where we collected PM2.5 particules).

The editor and reviewers concluded the dosage in our in vivo experiments was higher than a pregnant woman in Jinan that was mainly because our multiplication of the 100-fold uncertainty factor. The 100-fold uncertainty factor was derived from the integration of toxicokinetics and toxicodynamics, resulting in a multiplication of 10-fold interspecies difference with 10-fold interindividual variation. It was initially proposed by Lehman and Fitzhugh (Lehman, A.J et al. Association of the Food Drug Officials Quartely Bulletin 1954) over 60 years ago to convert a no-observed-adverse-effect level (NOAEL) from an animal toxicity study to a safe value for human intake (ADI). Scientists later applied the 100-fold uncertainty factor extensively to the selection of toxic dosage for animal experiments in toxicological studies, such as. Lautz L.S et al. Toxicol Lett, 2021; Arnot J.A. et al. J Expo Sci Environ Epidemiol, 2022; Tome D. et al. Curr Opin Clin Nutr Metab Care, 2020; Cooper A.B. et al. Regul Toxicol Pharmacol, 2019. Meanwhile, the Unite States Environmental Protection Agency (EPA) also identified the 100-fold uncertainty factor as the criteria for animal study in conducting a human health risk assessment (https://www.epa.gov/risk/conducting-human-health-risk-assessment). In authoritative studies related to PM2.5, the 100-fold uncertainty factor is also considered as one of the criteria for in vivo experiments (Chen Q et al. J Exp Clin Cancer Res 2022; Li Y. et al. Ecotoxicol Environ Saf, 2021; Li J. et al. Sci Total Environ, 2020; Zhang J et al. Sci Total Environ 2018; Zhang Y et al. Sci Total Environ 2017.). Therefore, the dosage of our PM2.5 in vivo experiment was not high but relatively reasonable as it covered the interspecies difference and interindividual variation using a scientific methodology. We have also added the content about 100-fold uncertainty factor to the Discussion section:

“The selection of appropriate PM2.5 exposure dosage in mice was critical for our experiments. We combined the PM2.5 exposure level of pregnant women from the Jinan [40] (where we collected PM2.5 particles) with physiological indicators of mice [41], then multiplied by 100-fold uncertainty factor to obtain the corresponding PM2.5 exposure dosage during mice pregnancy. The 100-fold uncertainty factor ( = 10-fold interspecies difference × 10-fold interindividual variation) is used to convert a no-observed-adverse-effect level (NOAEL) from an animal toxicity study to a safe value for human intake (ADI), which is the criteria for determining experimental dosages in toxicological studies involving animals. It was originally proposed, over 60 years ago, by Lehman and Fitzhugh[57], and still forms the basis of the uncertainty factors which are in use today. Also, the 100-uncertainty factor is considered one of the criteria for in vivo experiments in authoritative studies related to PM2.5 [58-60].” (P16 L383-392).

In conclusion, we think that the dosage of PM2.5 in our in vivo experiments was selected appropriately.

Secondly, regarding the time and frequency of PM2.5. Compared to the human gestation period of 280 days, the mouse gestation period is only 20 days. By aligning the developmental stages of mouse and human embryos, the 1.5^th^, 7.5^th^, and 12.5^th^ mouse gestational days which we selected to apply intratracheal instillation were designated approximately as the first, second, and third trimesters of human pregnancy (Amack JD et al. Cell Commun Signal 2021; Bunnell TM et al. Cytoskeleton (Hoboken) 2010; Xu C. et al. Environ Health Perspect 2022). This is to comprehensively investigate its impact throughout the gestation period. Due to the fragile condition of pregnant rats, we divided the PM2.5 dosage of the entire gestation period into three equal portions and conducted the three intratracheal instillations in order to minimize potential harm caused by the procedure. In many toxicological studies on embryonic development, same or similar time points were also chosen for in vivo experiments (Li R. et al. Chemosphere, 2018; Tata B Nat Med 2018; Koren O et al. Cell 2012). For a better understanding of the time points, I have rewritten "Pregnant mice were anaesthetized by intraperitoneal injection of 0.5% pentobarbital sodium (50 mg/kg) on 1.5 d, 7.5 d, and 12.5 d of pregnancy (corresponding to first, second and third trimester of human), followed by intratracheal instillation of 20 μL PM2.5 suspension" in method section (P24 L569-571).

Reference:

Amack JD. Cellular dynamics of EMT: lessons from live in vivo imaging of embryonic development. Cell Commun Signal 2021, 19(1): 79;Arnot JA, Toose L, Armitage JM, Sangion A, Looky A, Brown TN*, et al.* Developing an internal threshold of toxicological concern (iTTC). J Expo Sci Environ Epidemiol 2022, 32(6): 877-884.Bunnell TM, Ervasti JM. Delayed embryonic development and impaired cell growth and survival in Actg1 null mice. Cytoskeleton (Hoboken) 2010, 67(9): 564-572.Chen Q, Wang Y, Yang L, Sun L, Wen Y, Huang Y*, et al.* PM2.5 promotes NSCLC carcinogenesis through translationally and transcriptionally activating DLAT-mediated glycolysis reprograming. J Exp Clin Cancer Res 2022, 41(1): 229;Cooper AB, Aggarwal M, Bartels MJ, Morriss A, Terry C, Lord GA*, et al.* PBTK model for assessment of operator exposure to haloxyfop using human biomonitoring and toxicokinetic data. Regul Toxicol Pharmacol 2019, 102: 1-12;Koren O, Goodrich JK, Cullender TC, Spor A, Laitinen K, Backhed HK*, et al.* Host remodeling of the gut microbiome and metabolic changes during pregnancy. Cell 2012, 150(3): 470-480.Lautz LS, Jeddi MZ, Girolami F, Nebbia C, Dorne J. Metabolism and pharmacokinetics of pharmaceuticals in cats (Felix sylvestris catus) and implications for the risk assessment of feed additives and contaminants. Toxicol Lett 2021, 338: 114-127.Lehman A.J., O.G. Fitzhugh, 100-fold margin of safety, Association of the Food Drug Officials Quartely Bulletin (1954).Li Y, Batibawa JW, Du Z, Liang S, Duan J, Sun Z. Acute exposure to PM(2.5) triggers lung inflammatory response and apoptosis in rat. Ecotoxicol Environ Saf 2021, 222: 112526.Li J, Hu Y, Liu L, Wang Q, Zeng J, Chen C. PM2.5 exposure perturbs lung microbiome and its metabolic profile in mice. Sci Total Environ 2020, 721: 137432.Li R, Wang X, Wang B, Li J, Song Y, Luo B*, et al.* Gestational 1-nitropyrene exposure causes fetal growth restriction through disturbing placental vascularity and proliferation. Chemosphere 2018, 213: 252-258.Tata B, Mimouni NEH, Barbotin AL, Malone SA, Loyens A, Pigny P*, et al.* Elevated prenatal anti-Mullerian hormone reprograms the fetus and induces polycystic ovary syndrome in adulthood. Nat Med 2018, 24(6): 834-846.Tome D. Admissible daily intake for glutamate. Curr Opin Clin Nutr Metab Care 2020, 23(2): 133-137.Xu C, Ma H, Gao F, Zhang C, Hu W, Jia Y*, et al.* Screening of Organophosphate Flame Retardants with Placentation-Disrupting Effects in Human Trophoblast Organoid Model and Characterization of Adverse Pregnancy Outcomes in Mice. Environ Health Perspect 2022, 130(5): 57002;Zhang J, Liu J, Ren L, Wei J, Duan J, Zhang L*, et al.* PM(2.5) induces male reproductive toxicity via mitochondrial dysfunction, DNA damage and RIPK1 mediated apoptotic signaling pathway. Sci Total Environ 2018, 634: 1435-1444;Zhang Y, Hu H, Shi Y, Yang X, Cao L, Wu J*, et al.* (1)H NMR-based metabolomics study on repeat dose toxicity of fine particulate matter in rats after intratracheal instillation. Sci Total Environ 2017, 589: 212-221;

2) The metformin data lack rigour, and the authors would remove them and use these data to focus on a new study.

Many Thanks for your advice. In our study, we have observed a robust correlation between PM2.5 and adverse pregnancy outcomes; however, there is currently no clinical intervention or prophylactic medication available to alleviate this association. To mitigate the incidence of unfavorable pregnancy outcomes caused by atmospheric pollution, we performed some research of metformin. However, our investigation of metformin was insufficient because of time limit. We only utilized a single concentration of metformin and solely examined its mechanistic effects on KLF9 and CYP1A1. Therefore, we decided to delete the content of metformin from the manuscript according to the reviewers’ suggestion. We will investigate the mechanism of metformin in the treatment of functional impairment of trophoblasts caused by PM2.5 through cell function experiments combined with new RNA sequencing in the future study.

3) There are several concerns regarding the figures accompanying the results, including some that are not self-explanatory.

Many thanks for your helpful suggestion. We have implemented revisions item-by-item in response to the suggestions provided by reviewers.

Reviewer #1 (Recommendations for the authors):Air pollutant such as particulate matter PM2.5 is considered one of the most severe toxic associated with various adverse pregnancy outcomes. Most present research on PM2.5 in adverse effects on human pregnancy is focused on the epidemiological aspects, remaining partially unraveling the underlying molecular mechanisms. In this research, Zhang Wang and colleagues used PM2.5 collected from the urban region of Jinan, China, to establish pollutant exposure experiments by in vivo animal models and in vitro trophoblast cells. All experiments were evaluated by several in vitro techniques, including RNAseq analysis and in silico exploration of differentially expressed genes to infer the effect of PM2.5 on the mice pregnancy and human trophoblast cells.One caution of these studies is the dosage calculations for the in vivo and in vitro experimental design to evaluate the effect of PM 2.5. In the mice model, they administered doses of daily PM2.5 inhalation several times higher than a pregnant woman in Jinan might experience. On the other hand, an equivalent high dosage was applied to trophoblasts to extract total RNA and perform the RNAseq. However, the in silico data analysis only detected 32 differentially expressed genes using a Log2FC of no more than 1, denoting the subtle effect between the control and the treatments.

Thank you very much for your suggestion. We apologize for your confusion about our in vivo and in vitro dosages and RNA-Seq results due to the simplicity of our previous manuscript. I will explain in detail about the dosages and RNA-Seq results as follows.

Firstly, about the dosage in in vivo experiments. We collect the PM2.5 particles from Jinan, China, so we combined PM2.5 exposure of pregnant women in Jinan with the physiological indicators of mice, multiplied by 100-fold uncertainty factor to set the dosage in our in vivo experiments. The previous version was too simple for the reader to understand, so we have revised the PM2.5 dosage calculation process in detail in the method section:

“At 8 weeks, the average tidal volume of Kunming mice was about 0.25 mL, and the frequency of per mouse’s respiratory was about 163/min, thus, the total air intake per day was 0.25 x 163 x 60 mins x 24 hrs = 58680 mL ≈ 0.0587 m^3^/day[41]. It has been reported that the PM2.5 exposure of pregnant women in Jinan in 2020 was about 64 μg/m^3^ daily [40]. Therefore, the total PM2.5 intake though out the whole pregnancy of mice was about 0.0587 m^3^/day × 64 μg/m^3^ × 20 days × 100 (uncertainty factor) = 7511 μg. The 100 fold uncertainty factor = 10-fold interspecies difference ×10-fold interindividual variation [95] [60]. We weighed 7511 μg PM2.5 particles and dissolved it in 60 μL PBS buffer to prepare a PM2.5 suspension, which was subsequently divided into three equal portions. Pregnant mice were anaesthetized by intraperitoneal injection of 0.5% pentobarbital sodium (50 mg/kg) on 1.5 d, 7.5 d, and 12.5 d of pregnancy (corresponding to first, second and third trimester of human), followed by intratracheal instillation of 20 μL PM2.5 suspension, and the control group was intratracheally instilled with the same volume of PBS (n = 8 for per group).” (P24 L563-572).

The calculation of our PM2.5 dosage is as same as the calculation in many authoritative studies related PM2.5, such as Chen Q et al. J Exp Clin Cancer Res 2022; Li Y et al. Ecotoxicol Environ Saf, 2021; Li J et al. Sci Total Environ, 2020; Zhang J et al. Sci Total Environ 2018; Zhang Y et al. Sci Total Environ 2017. In these studies, the dosage of PM2.5 in the in vivo experiments was also determined by combining the ambient PM2.5 level with the physiological indicators of mice, multiplying by 100 times the uncertainty factor. The only difference of the dosage between these study and our manuscript is the ambient PM2.5 exposure level. They employed ambient PM2.5 level recommended by WHO, while we employed ambient PM2.5 levels of pregnant women in Jinan area ( where we collected PM2.5 particules). The reviewer considered the doses of daily PM2.5 inhalation of mice in our in vivo experiments were several times higher than a pregnant woman in Jinan mainly because our multiplication of the 100-fold uncertainty factor. The 100-fold uncertainty factor is one of the criteria commonly used in animal experiments in toxicological studies. It was derived from the integration of toxicokinetics and toxicodynamics, resulting in a multiplication of 10-fold interspecies difference with 10-fold interindividual variation. The 100-fold uncertainty factor was initially proposed by Lehman and Fitzhugh (Lehman, A.J et al. Association of the Food Drug Officials Quartely Bulletin 1954) over 60 years ago to convert a no-observed-adverse-effect level (NOAEL) from an animal toxicity study to a safe value for human intake (ADI). Scientists later applied the 100-fold uncertainty factor extensively to the selection of toxic dosage for animal experiments in toxicological studies, such as Lautz L.S et al. Toxicol Lett, 2021; Arnot J.A. et al. J Expo Sci Environ Epidemiol, 2022; Tome D et al. Curr Opin Clin Nutr Metab Care, 2020; Cooper A.B et al. Regul Toxicol Pharmacol, 2019. Meanwhile, the Unite States Environmental Protection Agency (EPA) also identified the 100-fold uncertainty factor as the criteria for animal study in conducting a human health risk assessment (https://www.epa.gov/risk/conducting-human-health-risk-assessment). In authoritative studies related to PM2.5 in vivo experiments, the 100-fold uncertainty factor is also considered as one of the criteria for in vivo experiments (Chen Q et al. J Exp Clin Cancer Res 2022; Li Y et al. Ecotoxicol Environ Saf, 2021; Li J et al. Sci Total Environ, 2020; Zhang J et al. Sci Total Environ 2018; Zhang Y et al. Sci Total Environ 2017.). Therefore, the dosage of our PM2.5 in vivo experiment was not high but relatively reasonable as it covered the interspecies difference and interindividual variation using a scientific methodology. We have also added the content about 100-fold uncertainty factor to the Discussion section:

“The selection of appropriate PM2.5 exposure dosage in mice was critical for our experiments. We combined the PM2.5 exposure level of pregnant women from the Jinan [40] (where we collected PM2.5 particles) with physiological indicators of mice [41], then multiplied by 100-fold uncertainty factor to obtain the corresponding PM2.5 exposure dosage during mice pregnancy. The 100-fold uncertainty factor ( = 10-fold interspecies difference × 10-fold interindividual variation) is used to convert a no-observed-adverse-effect level (NOAEL) from an animal toxicity study to a safe value for human intake (ADI), which is the criteria for determining experimental dosages in toxicological studies involving animals. It was originally proposed, over 60 years ago, by Lehman and Fitzhugh[57], and still forms the basis of the uncertainty factors which are in use today. Also, the 100-uncertainty factor is considered one of the criteria for in vivo experiments in authoritative studies related to PM2.5 [58-60]” (P16 L383-392).

In conclusion, we think that the dosage of PM2.5 in our in vivo experiments was selected appropriately.

Secondly, the PM2.5 dosage in our in vitro experiments has been also meticulously considered. The objective of our in vitro experiments was to investigate the impact of PM2.5 on human placental trophoblast cells. During the normal human physiological activity, PM2.5 particles inhaled are deposited within the lungs and subsequently transported to the placenta via circulation of blood. Although we are able to measure the concentration of PM2.5 in the air and estimate the dosage of PM2.5 that reaches the lungs, the PM2.5 particles are continuously accumulating in the lungs and there were no definitive studies to determine the precise dosage of PM2.5 that accumulated in the lungs or the dosage of PM2.5 that entered different organs via the bloodstream. The current in vitro studies of PM2.5 were mainly designed on a series of concentration gradients based on empirical or previous literature to investigate the toxic effects of PM2.5 on cell function (Wang Y et al., J Dermatol Sci 2021; Shan H et al. Ecotoxicol Environ Saf 2022; Wang Y et al. Sci Total Environ 2020). Therefore, we refered to previous authoritative literature and combined experiments on cell biological functions (e.g. PM2.5 on cell proliferation, apoptosis, invasion, migration, tube formation, etc.) to explore the effects of PM2.5 on trophoblast cells. In our study, we set three concentration gradients of 50 μg/mL, 100 μg/mL, 200 μg/mL (which was described in the Results section (P8 L187-189)) based on previous studies (Duan S et al. J Hazard Mater 2020; Guo X et al. Environ Pollut 2022, Qiu YN et al. Ecotoxicol Environ Saf 2019; Hu T et al. Environ Toxicol 2021; Zhao C et al. Sci Total Environ 2020). According to the results on the trophoblastic biological functions at these concentrations, we found that the trophoblastic biological function was impaired in a concentration-dependent manner. At a concentration of 50μg/mL, cell functions were slightly impaired, while at concentrations up to 200 μg/mL, all cell functions were severely impaired. According to the results of the CCK8 assays, the median lethal dose (LD50) of PM2.5 treatment for 24 h on trophoblast cells in our study was determined to be 105.2 µg/mL (as follows). LD50 represents the dose at which a substance is lethal for 50% of tested subjects. Toxicological studies typically employed concentrations in close proximity to the LD50 when investigating cellular functions or molecular mechanisms (Han C et al. Ecotoxicol Environ Saf 2023; Duan S et al. J Nanobiotechnology 2021; Akbar MU et al. ACS Omega 2022). Therefore, we chose concentration of 100 µg/mL for RNA-Seq sequencing and subsequent exploration of the molecular mechanism. In conclusion, the selection of PM2.5 dosage for our in vitro experiments is also scientific and evidence-based.

**Author response image 1. sa2fig1:** The LD50 of PM2. 5 treated trophoblasts (LD50 = 105.2 µg/mL).

Thirdly, about RNA-Seq results. In our in vitro experiments, we chose 100 μg/mL for our RNA-Seq experiment and the exploration of the molecular mechanism according to the LD50 value. In our RNA-Seq results, 32 differentially expressed genes were filtered out according to the pre-defined conditions of absolute value of log 2 (Fold Change) ≥1 and p-value <0.05 ( not Log2FC of no more than 1 as mentioned by reviewer). Although the number of differential genes shown by our RNA-Seq was not large, this result did not suggest that the effect of PM2.5 on trophoblast cells is minimal, as our previous biological function experiments had definitely demonstrated that PM2.5 altered the biological functions of cell proliferation, apoptosis, invasion, migration, and tube formation. We have also observed that the number of differentially expressed genes identified through RNA-Seq in numerous authoritative publications is not large. For example, in the article of “Phan BN et al., Nat Neurosci 2020”, there were 36 DEGs in P1 group; in the article of “Dai X et al., Front Med (Lausanne) 2022”, there were 43 DEGs; in the article of “Gu J et al.,. Biomed Res Int 2017”, 49 DEGs were identified; in the article of “Duan S et al., J Hazard Mater 2021”, 75 DEGs were identified. Although only a limited number of differential genes have been identified through RNA-Seq in these studies, their functional significance cannot be overstated. In line with our research, the number of differential genes sequenced by our RNA-Seq was limited, but we detected a crucial role for KLF9 and CYP1A1 in the oxidative stress-induced damage to trophoblasts triggered by PM2.5. Meanwhile, the RNA-Seq sequencing results revealed the altered transcript levels of genes, and we speculate that PM2.5 has the potential to affect the protein expression of genes even more due to the Western Blot results in our manuscript showed that PM2.5 significantly altered the protein expression level of KLF9, CYP1A1, cytochrome C, BCL^-^2, BAX, et al. Therefore, in our future study, we will perform proteomics sequencing analysis to further reveal the mechanisms by which PM2.5 affects trophoblastic biological function.

Reference:

Chen Q, Wang Y, Yang L, Sun L, Wen Y, Huang Y*, et al.* PM2.5 promotes NSCLC carcinogenesis through translationally and transcriptionally activating DLAT-mediated glycolysis reprograming. J Exp Clin Cancer Res 2022, 41(1): 229;Cooper AB, Aggarwal M, Bartels MJ, Morriss A, Terry C, Lord GA*, et al.* PBTK model for assessment of operator exposure to haloxyfop using human biomonitoring and toxicokinetic data. Regul Toxicol Pharmacol 2019, 102: 1-12;Dai X, Yang Z, Zhang W, Liu S, Zhao Q, Liu T, et al. Identification of diagnostic gene biomarkers related to immune infiltration in patients with idiopathic pulmonary fibrosis based on bioinformatics strategies. Front Med (Lausanne) 2022, 9: 959010.Duan S, Zhang M, Sun Y, Fang Z, Wang H, Li S*, et al.* Mechanism of PM(2.5)-induced human bronchial epithelial cell toxicity in central China. J Hazard Mater 2020, 396: 122747.Duan S, Zhang M, Li J, Tian J, Yin H, Wang X*, et al.* Uterine metabolic disorder induced by silica nanoparticles: biodistribution and bioactivity revealed by labeling with FITC. J Nanobiotechnology 2021, 19(1): 62.Gu J, Li T, Zhao L, Liang X, Fu X, Wang J, et al. Identification of Significant Pathways Induced by PAX5 Haploinsufficiency Based on Protein-Protein Interaction Networks and Cluster Analysis in Raji Cell Line. Biomed Res Int 2017, 2017: 5326370.Guo X, Lin Y, Lin Y, Zhong Y, Yu H, Huang Y*, et al.* PM2.5 induces pulmonary microvascular injury in COPD via METTL16-mediated m6A modification. Environ Pollut 2022, 303: 119115.Han C, Pei H, Sheng Y, Wang J, Zhou X, Li W*, et al.* Toxicological mechanism of triptolide-induced liver injury: Caspase3-GSDME-mediated pyroptosis of Kupffer cell. Ecotoxicol Environ Saf 2023, 258: 114963.Koren O, Goodrich JK, Cullender TC, Spor A, Laitinen K, Backhed HK*, et al.* Host remodeling of the gut microbiome and metabolic changes during pregnancy. Cell 2012, 150(3): 470-480.Lautz LS, Jeddi MZ, Girolami F, Nebbia C, Dorne J. Metabolism and pharmacokinetics of pharmaceuticals in cats (Felix sylvestris catus) and implications for the risk assessment of feed additives and contaminants. Toxicol Lett 2021, 338: 114-127.Lehman A.J., O.G. Fitzhugh, 100-fold margin of safety, Association of the Food Drug Officials Quartely Bulletin (1954).Li Y, Batibawa JW, Du Z, Liang S, Duan J, Sun Z. Acute exposure to PM(2.5) triggers lung inflammatory response and apoptosis in rat. Ecotoxicol Environ Saf 2021, 222: 112526.Li J, Hu Y, Liu L, Wang Q, Zeng J, Chen C. PM2.5 exposure perturbs lung microbiome and its metabolic profile in mice. Sci Total Environ 2020, 721: 137432.Phan BN, Bohlen JF, Davis BA, Ye Z, Chen HY, Mayfield B, et al. A myelin-related transcriptomic profile is shared by Pitt-Hopkins syndrome models and human autism spectrum disorder. Nat Neurosci 2020, 23(3): 375-385.Shan H, Li X, Ouyang C, Ke H, Yu X, Tan J*, et al.* Salidroside prevents PM2.5-induced BEAS-2B cell apoptosis via SIRT1-dependent regulation of ROS and mitochondrial function. Ecotoxicol Environ Saf 2022, 231: 113170.Tome D. Admissible daily intake for glutamate. Curr Opin Clin Nutr Metab Care 2020, 23(2): 133-137.Wang Y, Tang N, Mao M, Zhou Y, Wu Y, Li J*, et al.* Fine particulate matter (PM2.5) promotes IgE-mediated mast cell activation through ROS/Gadd45b/*JNK* axis. J Dermatol Sci 2021, 102(1): 47-57;Wang Y, Tang M. PM2.5 induces autophagy and apoptosis through endoplasmic reticulum stress in human endothelial cells. Sci Total Environ 2020, 710: 136397.Zhang J, Liu J, Ren L, Wei J, Duan J, Zhang L*, et al.* PM(2.5) induces male reproductive toxicity via mitochondrial dysfunction, DNA damage and RIPK1 mediated apoptotic signaling pathway. Sci Total Environ 2018, 634: 1435-1444;Zhang Y, Hu H, Shi Y, Yang X, Cao L, Wu J*, et al.* (1)H NMR-based metabolomics study on repeat dose toxicity of fine particulate matter in rats after intratracheal instillation. Sci Total Environ 2017, 589: 212-221;

Despite this, through in vivo experiments, the authors show that the administration of PM2.5 by intratracheal route leads to significant changes in the pregnancy of mice, significantly increased fetal mortality, decreased fetal weight and number, and damaged placental structure. The in vitro experiments showed increased apoptosis in trophoblastic line HTR8/SVneo cells.Through RNAseq experiments, the authors were able to infer two genes, KLF9 and CYP1A1, from the set of genes differentially expressed between trophoblast cells treated with 100 ug of PM2.5 compared to untreated controls. In addition, the authors demonstrated by in vitro ChIP assay the binding of KLF9 to the CYP1A1 promoter, proving that KLF9 is a positive modulator of CYP1A1 expression triggered by PM2.5. Consequently, the authors suggested activating this KLF9/CYP1A1 pathway promotes the empirically observed effects of oxidative stress and cell death in trophoblasts and the placenta of pregnant mice. But this study has not explored the mechanisms by which PM2.5 activates KLF9.

Thank you very much for your suggestion in our manuscript. In our study, we employed RNA-Seq and validation assays to identify CYP1A1 as a crucial gene involved in PM2.5-induced oxidative stress in trophoblast cells. Subsequently, bioinformatic predictions were utilized to determine its transcription factor KLF9, which was then confirmed through Chip and Dual-luciferase assays to regulate the transcriptional activity of CYP1A1. As the reviewer suggested, the research on the regulation of KLF9 by PM2.5 was missing. On one hand, exploring PM2.5 regulation of KLF9 would take a lot of time, and on the other hand, adding PM2.5 regulation of KLF9 to the current article might complicate our research focus. We will explore the regulation of PM2.5 on KLF9 and whether KLF9 can affect trophoblastic biological function through other signaling pathways in the future work. We have added this limitation in the Discussion section:

“Furthermore, our current article lacks investigations on the regulatory effect of PM2.5 on KLF9. These limitations should be further investigated in future.” (P22 L525-527).

Thank you again for your suggestions.

By additional experiments at the end of this work, the authors found that a single empirically deduced dose of metformin can reverse the toxic effects of PM2.5 on trophoblast cells of the HTR8/SVneo lineage. They observed that cells treated with metformin resulted in reduced expression of KLF9 and CYP1A1 compared to PM2.5-exposed cells. However, this experiment shows limitations, such as the single dose of the inhibitory drug used and a bias in the genetic mechanisms by focusing only on the two genes, KLF9 and CP1A1.

Many thanks for your advice. In our study, in order to resolve the clinical issue of how to prevent PM2.5-induced damage in pregnancy, we identified metformin by communicating with clinicians. Our experimental results confirmed that metformin indeed reduced oxidative stress damage and apoptosis in trophoblast cells caused by PM2.5. We then combined with the KLF9/CYP1A1, a signalling pathway that we had demonstrated that was involved in PM2.5-induced oxidative stress damage in trophoblast cells, to preliminarily validate the mechanism of metformin. However, our investigation of metformin was insufficient because of time limit. We only utilized a single concentration of metformin and solely examined its mechanistic effects on KLF9 and CYP1A1. Therefore, as suggested by other reviewers and editors, we decided to delete the content of metformin from the manuscript. Considering metformin could be a promising avenue for the clinical management of adverse pregnancy outcomes associated with PM2.5 exposure. So, we will explore the effects of different concentrations of metformin and investigate the mechanism of metformin through cell function experiments combined with new RNA sequencing in the future work.

This study used PM2.5 collected from the urban region of Jinan, China, to establish pollutant exposure experiments by in vivo animal models and trophoblast cells. This relevant study applies several in vitro and in silico techniques from these models. The authors identify that PM2.5 activates the KLF9/CYP1A1 signaling pathway causing oxidative stress damage with mitochondrial apoptosis, which can be correlated to poor pregnancy outcomes observed in mice.I have the following comments:1. The in vivo experimental designs of PM 2.5 dosage calculations are not unreliable to me. Because the in vitro and in vivo experiments were administered doses of daily PM2.5 inhalation huge times higher (2.503 ug) than a pregnant woman in Jinan might experience (3.6 ug). The inferences from this experimental design so different from the natural investigation in humans should be better explained. Also, it should be justified in comparison with some experiments with the proper doses experienced by pregnant women in Jinan.On the other hand, the dosage applied to trophoblasts for the extraction of total RNA and consequent RNAseq only detected 32 differentially expressed genes by using a Log2FC of 1, denoting the subtle effect between the control and the treatments. Under natural exposure conditions of daily PM2.5 (3.6 ug) instead of the current in vitro design (100 ug), it would be interesting to explore the gene expression results. Authors should discuss all these experimental limitations more rigorously.

Thanks a lot for your comments on our study. We apologize for your confusion about our results due to the simplicity of our writing in the previous manuscript. Please kindly allow me to provide a detailed explanation of the dosage used in both in vivo and in vitro experiments, as well as the RNA-Seq sequencing results.

Firstly, about the dosage in in vivo experiments. We collected the PM2.5 particles from Jinan, China, so we combined PM2.5 exposure of pregnant women in Jinan with the physiological indicators of mice, multiplied by 100-fold uncertainty factor to set the dosage in our in vivo experiments. The previous version was too simple for the readers to understand, so we have revised the PM2.5 dosage calculation process in detail in the method section:

“At 8 weeks, the average tidal volume of Kunming mice was about 0.25 mL, and the frequency of per mouse’s respiratory was about 163/min, thus, the total air intake per day was 0.25 x 163 x 60 mins x 24 hrs = 58680 mL ≈ 0.0587 m^3^/day[41]. It has been reported that the PM2.5 exposure of pregnant women in Jinan in 2020 was about 64 μg/m^3^ daily [40]. Therefore, the total PM2.5 intake though out the whole pregnancy of mice was about 0.0587 m^3^/day × 64 μg/m^3^ × 20 days × 100 (uncertainty factor) = 7511 μg. The 100 fold uncertainty factor = 10-fold interspecies difference ×10-fold interindividual variation [95] [60]. We weighed 7511 μg PM2.5 particles and dissolved it in 60 μL PBS buffer to prepare a PM2.5 suspension, which was subsequently divided into three equal portions. Pregnant mice were anaesthetized by intraperitoneal injection of 0.5% pentobarbital sodium (50 mg/kg) on 1.5 d, 7.5 d, and 12.5 d of pregnancy (corresponding to first, second and third trimester of human), followed by intratracheal instillation of 20 μL PM2.5 suspension, and the control group was intratracheally instilled with the same volume of PBS (n = 8 for per group).” (P24 L563-572).

The calculation of our PM2.5 dosage is as same as the calculation in many authoritative studies related PM2.5, such as Chen Q et al. J Exp Clin Cancer Res 2022; Li Y et al. Ecotoxicol Environ Saf, 2021; Li J et al. Sci Total Environ, 2020; Zhang J et al. Sci Total Environ 2018; Zhang Y et al. Sci Total Environ 2017. In these studies, the dosage of PM2.5 in the in vivo experiments was also determined by combining the ambient PM2.5 level with the physiological indicators of mice, multiplying by 100 times the uncertainty factor. The only difference of the dosage between these study and our manuscript is the ambient PM2.5 exposure level. They employed ambient PM2.5 level recommended by WHO, while we employed ambient PM2.5 levels of pregnant women in Jinan area ( where we collected PM2.5 particules).

The reviewer considered the doses of daily PM2.5 inhalation of mice in our in vivo experiments were higher than a pregnant woman in Jinan that was mainly because our multiplication of the 100-fold uncertainty factor. The 100-fold uncertainty factor is one of the criteria commonly used in animal experiments in toxicological studies. It was derived from the integration of toxicokinetics and toxicodynamics, resulting in a multiplication of 10-fold interspecies difference with 10-fold interindividual variation. The 100-fold uncertainty factor was initially proposed by Lehman and Fitzhugh (Lehman, A.J et al. Association of the Food Drug Officials Quartely Bulletin 1954) over 60 years ago to convert a no-observed-adverse-effect level (NOAEL) from an animal toxicity study to a safe value for human intake (ADI). Scientists later applied the 100-fold uncertainty factor extensively to the selection of toxic dosage for animal experiments in toxicological studies, such as Lautz L.S et al. Toxicol Lett, 2021; Arnot J.A. et al. J Expo Sci Environ Epidemiol, 2022; Tome D et al. Curr Opin Clin Nutr Metab Care, 2020; Cooper A.B et al. Regul Toxicol Pharmacol, 2019. Meanwhile, the Unite States Environmental Protection Agency (EPA) also identified the 100-fold uncertainty factor as the criteria for animal study in conducting a human health risk assessment (https://www.epa.gov/risk/conducting-human-health-risk-assessment). In authoritative studies related to PM2.5 in vivo experiments, the 100-fold uncertainty factor is also considered as one of the criteria for in vivo experiments (Chen Q et al. J Exp Clin Cancer Res 2022; Li Y et al. Ecotoxicol Environ Saf, 2021; Li J et al. Sci Total Environ, 2020; Zhang J et al. Sci Total Environ 2018; Zhang Y et al. Sci Total Environ 2017.). Therefore, although the dosage in our in vivo experiment may appear higher than the PM2.5 dosage absorbed by normal pregnant women, it was determined through a scientific methodology that taked into consideration interspecies difference and interindividual variation. We have also added this component to the Discussion section:

“The selection of appropriate PM2.5 exposure dosage in mice was critical for our experiments. We combined the PM2.5 exposure level of pregnant women from the Jinan [40] (where we collected PM2.5 particles) with physiological indicators of mice [41], then multiplied by 100-fold uncertainty factor to obtain the corresponding PM2.5 exposure dosage during mice pregnancy. The 100-fold uncertainty factor ( = 10-fold interspecies difference × 10-fold interindividual variation) is used to convert a no-observed-adverse-effect level (NOAEL) from an animal toxicity study to a safe value for human intake (ADI), which is the criteria for determining experimental dosages in toxicological studies involving animals. It was originally proposed, over 60 years ago, by Lehman and Fitzhugh[57], and still forms the basis of the uncertainty factors which are in use today. Also, the 100-uncertainty factor is considered one of the criteria for in vivo experiments in authoritative studies related to PM2.5 [58-60]” (P16 L383-392).

Secondly, about the dosage of PM2.5 in our in vitro experiments. The objective of our in vitro experiments was to investigate the impact of PM2.5 on human placental trophoblast cells. During the normal human physiological activity, PM2.5 particles inhaled are deposited within the lungs and subsequently transported to the placenta via circulation of blood. Although we are able to measure the concentration of PM2.5 in the air and estimate the dosage of PM2.5 that reaches the lungs, the PM2.5 particles are continuously accumulating in the lungs and there were no definitive studies to determine the precise dosage of PM2.5 that accumulated in the lungs or the dosage of PM2.5 that entered different organs via the bloodstream. The current in vitro studies of PM2.5 were mainly designed on a series of concentration gradients based on empirical or previous literature to investigate the toxic effects of PM2.5 on cell function (Wang Y et al., J Dermatol Sci 2021; Shan H et al. Ecotoxicol Environ Saf 2022; Wang Y et al. Sci Total Environ 2020).Therefore, we refered to previous authoritative literature and combined experiments on cell biological functions (e.g. PM2.5 on cell proliferation, apoptosis, invasion, migration, tube formation, etc.) to explore the effects of PM2.5 on trophoblast cells. In our study, we set three concentration gradients of 50 μg/mL, 100 μg/mL, 200 μg/mL based on previous studies (Duan S et al. J Hazard Mater 2020; Guo X et al. Environ Pollut 2022, Qiu YN et al. Ecotoxicol Environ Saf 2019; Hu T et al. Environ Toxicol 2021; Zhao C et al. Sci Total Environ 2020), which was described in the Results section (P8 L187-189). According to the results on the trophoblastic biological functions at these concentrations, we found that the trophoblastic biological function was impaired in a concentration-dependent manner. At a concentration of 50μg/mL, cell functions were slightly impaired, while at concentrations up to 200 μg/mL, all cell functions were severely impaired. According to the results of the CCK8 assays, the median lethal dose (LD50) of PM2.5 treatment for 24 h on trophoblast cells in our study was determined to be 105.2 µg/mL (as follows). LD50 represents the dose at which a substance is lethal for 50% of tested subjects. Toxicological studies typically employed concentrations in close proximity to the LD50 when investigating cellular functions or molecular mechanisms (Han C et al. Ecotoxicol Environ Saf 2023; Duan S et al. J Nanobiotechnology 2021; Akbar MU et al. ACS Omega 2022). Therefore, we chose concentration of 100 µg/mL for RNA-Seq sequencing and subsequent exploration of the molecular mechanism. In conclusion, the selection of PM2.5 dosage for our in vitro experiments is also scientific and evidence-based.

Thirdly, about RNA-Seq results. In our in vitro experiments, we chose 100 μg/mL for our RNA-Seq experiment and the exploration of the molecular mechanism according to the LD50 value. In our RNA-Seq results, 32 differentially expressed genes were filtered out according to the pre-defined conditions of absolute value of log 2 (Fold Change)≥1 and p-value <0.05. Although the number of differential genes shown by our RNA-Seq was not large, this result did not suggest that the effect of PM2.5 on trophoblast cells is minimal, as our previous biological function experiments had definitely demonstrated that PM2.5 altered the biological functions of cell proliferation, apoptosis, invasion, migration, and tube formation. We have also observed that the number of differentially expressed genes identified through RNA-Seq in numerous authoritative publications is not large. For example, in the article of “Phan BN et al., Nat Neurosci 2020”, there were 36 DEGs in P1 group; in the article of “Dai X et al., Front Med (Lausanne) 2022”, there were 43 DEGs; in the article of “Gu J et al.,. Biomed Res Int 2017”, 49 DEGs were identified; in the article of “Duan S et al., J Hazard Mater 2020”, there were 75 DEGs were identified. Although only a limited number of differential genes have been identified through RNA-Seq in the literature, their functional significance cannot be overstated. In line with our research, the number of differential genes sequenced by our RNA-SEq was limited, but we detected a crucial role for KLF9 and CYP1A1 in the oxidative stress-induced damage to trophoblasts triggered by PM2.5. The RNA-Seq sequencing results revealed the altered transcript levels of genes, and we speculate that PM2.5 has the potential to affect the protein expression of genes even more due to the Western Blot results in our manuscript showed that PM2.5 significantly altered the protein expression level of KLF9, CYP1A1, cytochrome C, BCL^-^2, BAX, et al. Therefore, in our future study, we will perform proteomics sequencing analysis to further reveal the mechanisms by which PM2.5 affects trophoblastic biological function.

Reference:

Akbar MU, Badar M, Zaheer M. Programmable Drug Release from a Dual-Stimuli Responsive Magnetic Metal-Organic Framework. ACS Omega 2022, 7(36): 32588-32598.Arnot JA, Toose L, Armitage JM, Sangion A, Looky A, Brown TN*, et al.* Developing an internal threshold of toxicological concern (iTTC). J Expo Sci Environ Epidemiol 2022, 32(6): 877-884.Chen Q, Wang Y, Yang L, Sun L, Wen Y, Huang Y*, et al.* PM2.5 promotes NSCLC carcinogenesis through translationally and transcriptionally activating DLAT-mediated glycolysis reprograming. J Exp Clin Cancer Res 2022, 41(1): 229;Cooper AB, Aggarwal M, Bartels MJ, Morriss A, Terry C, Lord GA*, et al.* PBTK model for assessment of operator exposure to haloxyfop using human biomonitoring and toxicokinetic data. Regul Toxicol Pharmacol 2019, 102: 1-12;Dai X, Yang Z, Zhang W, Liu S, Zhao Q, Liu T, et al. Identification of diagnostic gene biomarkers related to immune infiltration in patients with idiopathic pulmonary fibrosis based on bioinformatics strategies. Front Med (Lausanne) 2022, 9: 959010.Duan S, Zhang M, Sun Y, Fang Z, Wang H, Li S*, et al.* Mechanism of PM(2.5)-induced human bronchial epithelial cell toxicity in central China. J Hazard Mater 2020, 396: 122747.Duan S, Zhang M, Li J, Tian J, Yin H, Wang X*, et al.* Uterine metabolic disorder induced by silica nanoparticles: biodistribution and bioactivity revealed by labeling with FITC. J Nanobiotechnology 2021, 19(1): 62.Gu J, Li T, Zhao L, Liang X, Fu X, Wang J, et al. Identification of Significant Pathways Induced by PAX5 Haploinsufficiency Based on Protein-Protein Interaction Networks and Cluster Analysis in Raji Cell Line. Biomed Res Int 2017, 2017: 5326370.Guo X, Lin Y, Lin Y, Zhong Y, Yu H, Huang Y*, et al.* PM2.5 induces pulmonary microvascular injury in COPD via METTL16-mediated m6A modification. Environ Pollut 2022, 303: 119115.Han C, Pei H, Sheng Y, Wang J, Zhou X, Li W*, et al.* Toxicological mechanism of triptolide-induced liver injury: Caspase3-GSDME-mediated pyroptosis of Kupffer cell. Ecotoxicol Environ Saf 2023, 258: 114963.Hu T, Zhu P, Liu Y, Zhu H, Geng J, Wang B*, et al.* PM2.5 induces endothelial dysfunction via activating NLRP3 inflammasome. Environ Toxicol 2021, 36(9): 1886-1893.Koren O, Goodrich JK, Cullender TC, Spor A, Laitinen K, Backhed HK*, et al.* Host remodeling of the gut microbiome and metabolic changes during pregnancy. Cell 2012, 150(3): 470-480.Lautz LS, Jeddi MZ, Girolami F, Nebbia C, Dorne J. Metabolism and pharmacokinetics of pharmaceuticals in cats (Felix sylvestris catus) and implications for the risk assessment of feed additives and contaminants. Toxicol Lett 2021, 338: 114-127.Lehman A.J., O.G. Fitzhugh, 100-fold margin of safety, Association of the Food Drug Officials Quartely Bulletin (1954).Li Y, Batibawa JW, Du Z, Liang S, Duan J, Sun Z. Acute exposure to PM(2.5) triggers lung inflammatory response and apoptosis in rat. Ecotoxicol Environ Saf 2021, 222: 112526.Li J, Hu Y, Liu L, Wang Q, Zeng J, Chen C. PM2.5 exposure perturbs lung microbiome and its metabolic profile in mice. Sci Total Environ 2020, 721: 137432.Phan BN, Bohlen JF, Davis BA, Ye Z, Chen HY, Mayfield B, et al. A myelin-related transcriptomic profile is shared by Pitt-Hopkins syndrome models and human autism spectrum disorder. Nat Neurosci 2020, 23(3): 375-385.Qiu YN, Wang GH, Zhou F, Hao JJ, Tian L, Guan LF*, et al.* PM2.5 induces liver fibrosis via triggering ROS-mediated mitophagy. Ecotoxicol Environ Saf 2019, 167: 178-187.Shan H, Li X, Ouyang C, Ke H, Yu X, Tan J*, et al.* Salidroside prevents PM2.5-induced BEAS-2B cell apoptosis via SIRT1-dependent regulation of ROS and mitochondrial function. Ecotoxicol Environ Saf 2022, 231: 113170.Tome D. Admissible daily intake for glutamate. Curr Opin Clin Nutr Metab Care 2020, 23(2): 133-137.Wang Y, Tang M. PM2.5 induces autophagy and apoptosis through endoplasmic reticulum stress in human endothelial cells. Sci Total Environ 2020, 710: 136397.Wang Y, Tang N, Mao M, Zhou Y, Wu Y, Li J*, et al.* Fine particulate matter (PM2.5) promotes IgE-mediated mast cell activation through ROS/Gadd45b/*JNK* axis. J Dermatol Sci 2021, 102(1): 47-57.Zhang J, Liu J, Ren L, Wei J, Duan J, Zhang L*, et al.* PM(2.5) induces male reproductive toxicity via mitochondrial dysfunction, DNA damage and RIPK1 mediated apoptotic signaling pathway. Sci Total Environ 2018, 634: 1435-1444;Zhang Y, Hu H, Shi Y, Yang X, Cao L, Wu J*, et al.* (1)H NMR-based metabolomics study on repeat dose toxicity of fine particulate matter in rats after intratracheal instillation. Sci Total Environ 2017, 589: 212-221;Zhao C, Wang Y, Su Z, Pu W, Niu M, Song S*, et al.* Respiratory exposure to PM2.5 soluble extract disrupts mucosal barrier function and promotes the development of experimental asthma. Sci Total Environ 2020, 730: 139145.

2. Regarding the English written in the manuscript, I found several misspelled words, such as typo errors. Also, the authors need to check a few wrong or missing prepositions, punctuation, and the agreement between subject and verb in some sentences. Furthermore, please check words in the original language.

Thank you for your valuable guidance on the writing of my manuscript. After conducting a meticulous sentence-by-sentence review, I have rectified any erroneous writing in the original manuscript and made necessary amendments.

3. In the manuscript, beginning with the abstract, it would be essential to change the meaning of the inferences obtained from the analysis of the RNAseq experiments. Since this type of in vitro experiment is only exploratory, it then raises hypotheses that must be confirmed a posteriori through other in vivo experiments. In the abstract the current text reads: "we comprehensively analyzed the transcriptional landscape of HTR8/SVneo cells exposed to PM2.5 through RNA-Seq and confirmed that PM2.5 triggered oxidative stress and mitochondrial apoptosis to damage HTR8/SVneo cell biological functions through CYP1A1." The meaning could be changed by using this statement "we comprehensively analyzed the transcriptional landscape of HTR8/SVneo cells exposed to PM2.5 through RNA-Seq and observed that PM2.5 triggered overexpression of pathways involved in oxidative stress and mitochondrial apoptosis to damage HTR8/SVneo cell biological functions through CYP1A1." Or "we comprehensively analyzed the transcriptional landscape of HTR8/SVneo cells exposed to PM2.5 through RNA-Seq and confirmed by validation tests that PM2.5 triggered oxidative stress and mitochondrial apoptosis to damage HTR8/SVneo cell biological functions through CYP1A1."

Thank you for your valuable suggestion on my summary, which I wholeheartedly agree with. I have changed the abstract with:

“we comprehensively analyzed the transcriptional landscape of HTR8/SVneo cells exposed to PM2.5 through RNA-Seq and observed that PM2.5 triggered overexpression of pathways involved in oxidative stress and mitochondrial apoptosis to damage HTR8/SVneo cell biological functions through CYP1A1.” (P3 L57-59).

Your revision has enhanced the rigor and scientificity of our article, greatly assisting us in its development. We sincerely appreciate for your valuable suggestions.

4. The methodology for the bioinformatics analysis of the RNA-seq experiments needs to be better described. The authors should better detail the data pre-processing steps (to get quality control), including the software and the parameters applied until achieved at Differentially Expressed Genes (DEGs), such as read alignments, counts, normalization, and DEGs.Although the analyses of the KEGG pathway enrichment analysis of differentially expressed genes and Gene Set Enrichment Analysis (GSEA) were carried out in a particular company, it is also desirable that the details of the analyses are described in the methodology with their respective parameters.

Thank you for your valuable suggestion on our manuscript. We concur with the reviewer's recommendation and have revised the methodological section of our RNA-seq bioinformatics analysis accordingly. The modifications are outlined below:

“The raw sequencing data was subjected to filtration using SOAPnuke in order to obtain clean reads. The clean reads were mapped to the reference genome using HISAT, and aligned to the assembled unique gene set using Bowtie2. Subsequently, the data analysis and mapping were performed utilizing the sequencing company's proprietary system (https://biosys.bgi.com). The gene expression was quantified utilizing the RNA-Seq by Expectation-Maximization (RSEM) algorithm, and differentially expressed genes (DEGs) were identified by the R-Bioconductor package DESeq2 with pre-set Q value﹤0.05 and |log2[Fold Change]|≥1. KEGG enrichment analysis of annotated DEGs was performed using phyper based on Hypergeometric test, and the significance values of pathways were strictly threshold corrected by Q values (Q values≤0.05). Gene Set Enrichment Analysis (GSEA) was performed on control and treatment groups based on KEGG database data. All parameters were kept at default settings, except for the maximum size of the filtering threshold which was adjusted to 5000. Gene features with FDR Q values≤0.25 were deemed statistically significant.” (P30 L697-708).

5. Some bioinformatics analyses deserve to be better explained. This is because this manuscript reports an interdisciplinary work, and it is expected that non-experts in bioinformatics can understand the findings and correlations between the different results. For example, explain in more detail the analyses and how to interpret each result shown in figure 4, from A to H, particularly curves and cutoff values.

Many thanks for your suggestion. As recommended by the reviewer, our description of biogenic analysis is overly simplistic. Therefore, in order to provide readers with a more comprehensive understanding of our research, we have revised our results and figure legends sections. The following was modified in the results:

“in addition, 3D principal component analysis showed that the clusters of samples with high similarity were consistent with the actual exposure grouping, suggesting that PM2.5 significantly changed the gene expression of the HTR8 cells. By applying the cut-off criteria of Q value﹤0.05 and |log2[Fold Change]|≥1, we identified 32 coding genes that exhibited differential expression between the control and PM2.5 treatment groups, comprising 24 up-regulated genes and 8 down-regulated genes. These findings were visually represented through volcano plots and heat maps. The KEGG pathway enrichment analysis was performed to predict the potential regulatory mechanisms of PM2.5-induced trophoblast damage. Our findings were further supported by GSEA, which revealed significant upregulation of four enriched pathways – Cytochrome P450 pathway, chemical carcinogenesis pathway, ovarian steroidogenesis pathway and steroid biosynthesis pathway – in the PM2.5-exposed group. The results from both analyses suggest that cytochrome P450 is the most significantly enriched pathway”. (P10 L222-234) We also append the following caption to the legend of Figure 4: “The zero-cross line indicates the point in which the difference between expression in the PM2.5-treated and control groups is zero. NES, normalized enrichment score; FDR, false discovery rate.” (P44 L1190)

6. Figures:6.1. I feel that the figures presented accompanying the results are not self-explanatory. There are too many tags that the non-expert reader cannot understand because they have not dominated the meaning. To cite just one example here: Figure 9D: si-NC – si-KLF9#1 – si-KLF9#2. The authors need to put the meaning of the figures' tags in the respective captions. Remembering again that this is an interdisciplinary paper.

Many thanks for your suggestion. I have provided a more detailed description in the figure legends to enhance the reader's comprehension of my manuscript. Such as: “FSC-A: Forward Scatter-Area; SSC-A: Side Scatter-Area” in P44 L1170; “The zero-cross line indicates the point in which the difference between expression in the PM2.5-treated and control groups is zero. NES, normalized enrichment score; FDR, false discovery rate.” in P45 L1192; “si-CYP1A1#1, si-CYP1A1#2: siRNAs to knockdown CYP1A1; si-NC: siRNA Negative Control” in P48 L1264; “si-KLF9#1, si-KLF9#2: siRNA to knockdown KLF9 expression; si-NC: siRNA Negative Control” in P49 L1299; “(I) The histogram indicated the Mito-SOX Red Staining mean intensity in each group. (J) The histogram indicated the Red/Green intensity in each group” in P46 L1214.

6.2. There is a critical error in Figure 4E, where the authors show the Z-score scale in the heat map. While in the manuscript text they describe that 24 genes were up-regulated and eight genes down-regulated in the PM 2.5 samples, in Figure 4E, we see the opposite. In Figure 4E, we see 24 genes in red that represent an expression level lower than the mean, whereas eight green genes that represent an expression level above the mean related to PM 2.5 samples compared to control samples. Please check this figure and identify and correct this critical error.

We are very grateful to the reviewer for helping us find this major mistake. The Row Z-Score legend was mistakenly labelled during the drawing process, resulting in this issue. We have rectified it in the Figure 4B. (As we have removed 4A, 4B, 4C to Supplement 1A, 1B, 1C as advised by the other reviewer, the previous Figure 4E has become the current Figure 4B.)

6.3. Figure 9G is very confusing to me. This figure contains a left and right panel that the reader must interpret as correlating. The left panel shows the CYP1A1 promoter region with several deletions that either resulted in failure of binding by FT KLF9 or not. Therefore, in the right panel, the reader would expect to see which mutations prevent FT KLF9-mediated CY1A1 expression. However, in Figure 9G on the right panel, the authors showed the expression of KLF9 or control. So, in this scenario, the picture gets confused. I suggest that the authors correct this figure to clarify this experiment's results.Also, in the explanation of this figure 9, there is a mistake in the sense of writing, e.g., "These results suggested that a response element of the CYP1A1 promoter located in the -64 bp to -49 bp region, GAAGGAGGCGTGGCC, was required for the transcription of KLF9." A change to the following is recommended: "…, was required for the transcription of CYP1A1 meditated by KLF9."

Many thanks for your suggestion. We apologize for any confusion caused by the image in regards to the reviewer's comprehension. The left schematic illustrates the predicted binding sites of the CYP1A1 promoter through the KLF9 motif and the construction of mutant reporter gene plasmids based on site-specific design. The mutated human CYP1A1 promoter-luciferase reporter gene plasmids were co-transfected with pRL-TK into HTR8/SVneo cells stably expressing Vector and KLF9 for further analysis. The right bar showed the relative luciferase activity of different CYP1A1 promoters in KLF9 overexpressing cells (KLF9) compared to control cells (Vector) using dual-luciferase reporter gene assays. This was illustrated in the legend of the previous manuscript. We have rewritten the legend of this figure to facilitate the understanding of this result by reviewers and readers:

“The bars on the right showed the relative luciferase activity of different CYP1A1 promoters in KLF9 overexpressing cells (KLF9) compared to control cells (Vector) using dual-luciferase reporter gene assays.” (P49 L1290).

Also, we have changed the color of the legend "KLF9 binding sites" on the left side of Figure 9G to orange and the coordinates of the bar on the right side to "Relative luciferase activity". Moreover, we have corrected it in result section as follows:

“These results suggested that a response element of the CYP1A1 promoter located in the -64 bp to -49 bp region, GAAGGAGGCGTGGCC, was required for the transcription of CYP1A1 meditated by KLF9.”

(P15 L340). As we have deleted Figure 9C as advised by the other reviewer, the previous Figure 9G has become the current Figure 9F.

6.4. Figures 9 H and I do not show what the authors claim in the text: "The chromatin was precipitated with specific antibodies for KLF9 or IgG, and PCR analysis with the indicated primers showed that KLF9 was able to bind directly to the CYP1A1 promoter (Figure 9H and I)."The authors need to show the figure that corresponds with this statement. It would be interesting to perform the ChIP analysis experiment using wild-type and mutant constructs of the CY1A1 promoter, confirming the results of the region of this promoter that effectively binds KLF9.

Many thanks for your advice. We apologize for your confusion about our results due to the simplicity of the previous manuscript. In the previous Figure 9H and I, we employed ChIP to investigate the binding of KLF9 to the CYP1A1 promoter. The chromatin was precipitated with antibodies specific for KLF9 or IgG, and PCR was conducted with CYP1A1 promoter primers (spanning the -96/+16 bp region of the CYP1A1 promoter). The result of agarose gel electrophoresis and RT-qPCR analysis confirmed that KLF9 was able to bind directly to the CYP1A1 promoter. To make it easier to understand, we revised the result:

“The chromatin was precipitated with antibodies specific for KLF9 or IgG, and PCR analysis with the primers (spanning the -96/+16 bp region of the CYP1A1 promoter) showed that KLF9 was able to bind directly to the CYP1A1 promoter (Figure 9G and H).”

(P15 L342-344).

Meanwhile, we inserted the following description of the picture in the legend:

“(G) Chromatin from KLF9 over-expression HTR8/SVneo cells was subjected to ChIP assay using KLF9 antibody or control IgG. PCR amplification with primers spanning the -96/+16 bp region of the CYP1A1 promoter was performed. A 2% agarose gel electrophoresis was performed on PCR products. (H) RT-qPCR analysis quantitatively demonstrated that KLF9 overexpression increased its binding to the endogenous CYP1A1 promoter” (P49 L1291-1295).

As we have deleted Figure 9C as advised by the other reviewer, the previous Figure 9H and I have become the current Figure 9G and H.

In our study, in order to demonstrate that KLF9 can bind directly to CYP1A1 promoter, we first identified the region by which KLF9 can promote transcription of CYP1A1 by dual-luciferase assays, then designed primers based on this region and verified by ChIP-qPCR that KLF9 can bind directly to this region. The method we used to demonstrate that KLF9 binds directly to the CYP1A1 promoter region is as same as many authoritative papers (Wang L et al. Cancer Lett. 2020; Lin CC et al. Theranostics 2020). The plasmids that reviewer mentioned that we constructed containing mutated CYP1A1 promoter regions were specifically for luciferase expression. So, we may not able to perform the ChIP analysis experiment using wild-type and mutant constructs of the CY1A1 promoter currently. In the future work, we will use mutant constructs of the CY1A1 promoter to confirming the region of this promoter that effectively binds KLF9 as suggested by the reviewer. Thank you again for your very useful advice.

Reference:

Wang L, Zhang Z, Yu X, Li Q, Wang Q, Chang A*, et al. SOX9*/miR-203a axis drives PI3K/AKT signaling to promote esophageal cancer progression. Cancer Lett 2020, 468: 14-26.Lin CC, Kuo IY, Wu LT, Kuan WH, Liao SY, Jen J*, et al.* Dysregulated Kras/YY1/ZNF322A/Shh transcriptional axis enhances neo-angiogenesis to promote lung cancer progression. Theranostics 2020, 10(22)**:** 10001-10015.

7. Discussion: related to this sentence "To our knowledge, this study is the first to report that PM2.5 caused mitochondrial apoptosis via inducing oxidative stress, which in-turn impaired a series of biological functions such as invasion, migration, and angiogenesis in placental trophoblasts.", Actually, this one paper Front. Endocrinol., 12 March 2020 Sec. Translational Endocrinology Volume 11 – 2020 https://doi.org/10.3389/fendo.2020.00075. Please double check, since they also described oxidative damage in trophoblasts and correct corresponding.

Many thanks for your advice. The purpose of our previous manuscript was to demonstrate that we were the first time to identify that PM2.5 altered cell biological function in trophoblast cells through oxidative stress induced mitochondrial apoptosis. In the paper “Endocrinol., 12 March 2020 Sec. Translational Endocrinology Volume 11 – 2020 | https://doi.org/10.3389/fendo.2020.00075.”, the authors discovered that PM2.5 induced oxidative stress, mitochondrial damage and inflammation in trophoblasts. However, the study did not explore whether oxidative stress was responsible for causing mitochondrial damage and inflammation.

Our previous description failed to effectively convey our perspective, thus we have revised.

"Our study demonstrated that exposure to PM2.5 induced oxidative stress in placental trophoblast, leading to mitochondrial apoptosis and impairment of cell biological functions."(P18 L426-428)

8. Inferences report to metformin:Inferences about the application of the drug metformin were raised from experiments testing a single dose at a high level (20 mM). The authors only concentrated on the target genes highlighted in this work (CYP1A1 and KLF9).I suggest that metformin evaluation should be better explored in another paper. In this scenario, the results are not very robust and complicate the focus of this paper, which turned out to be very large. Metformin could be explored by tests using more treatments and even performing new RNAseq experiments on trophoblast cells exposed to the drug at different doses.

Thanks a lot for your advice. In our study, we have observed a robust correlation between PM2.5 and adverse pregnancy outcomes; however, there is currently no clinical intervention or prophylactic medication available to alleviate this association. To mitigate the incidence of unfavorable pregnancy outcomes caused by atmospheric pollution, we identified metformin by communicating with clinicians. We then combined with the KLF9/CYP1A1, a signalling pathway that we had demonstrated that was involved in PM2.5-induced oxidative stress damage in trophoblast cells, to preliminarily validate the mechanism of metformin. However, our investigation of metformin was insufficient because of time limit. We only utilized a single concentration of metformin and solely examined its mechanistic effects on KLF9 and CYP1A1. Therefore, as suggested by the reviewer and editor, we have removed the metformin-related content. Considering metformin could be a promising avenue for the clinical management of adverse pregnancy outcomes associated with PM2.5 exposure. Therefore, in future studies, we will enhance the concentration gradient of metformin and conduct a more thorough exploration into its role in safeguarding against adverse pregnancies caused by PM2.5 through new RNA-Seq or proteomic sequencing analysis.

Reviewer #2 (Recommendations for the authors):Epidemiological studies have linked an increase in air pollutants, including fine particulate matter (PM2.5) to adverse pregnancy and postnatal outcomes. However, the molecular details of this are unclear, partially because there is no established mouse model in which to investigate the effects of PM2.5 on mammalian pregnancy. Li and Li et al. begin this study with the collection and characterization of PM2.5 from a high-volume sampling system set up in Jinan City, China. Upon collection of this material, the authors began their study using scanning electron microscopy and elemental analysis to define the properties of PM2.5. The authors use this material to establish a mouse model and cellular system to test the molecular effects of PM2.5 on pregnancy, postnatal mouse health, and trophoblast cells. One caveat of these studies is the dosage of PM2.5 given to the mice; the authors calculate dosages based on their estimation of PM2.5 exposure of pregnant women in Jinan in 2020, but include a 100-fold "uncertainty factor," which substantially increases the amount of PM2.5 given to the mice in their model system. This, along with the multiple timepoints at which PM2.5 is administered, suggests that the authors may be dosing mice with up to 400 times higher levels of PM2.5 than humans experience, on average, based on their own calculations. It is unclear if more physiologically relevant doses of PM2.5 (less the uncertainty factor) would have such stark effects on pregnancy outcomes as are shown in this study.

Thank you very much for your suggestion on our manuscript. We apologize for your confusion about our PM2.5 dosage given to the mice due to the over simplicity of our writing in the previous manuscript. Please kindly allow me to provide a detailed explanation of the dosage used in in vivo experiments.

In the in vivo experiments, we determined the PM2.5 exposure dosage for the whole gestation period by combining the PM2.5 exposure of pregnant women in Jinan and the physiological indicators of mice, multiplied by 100-fold uncertainty factor. The 100-fold uncertainty factor is one of the guidelines for the design of dosage for animal experiments in toxicological studies. It was derived from the integration of toxicokinetics and toxicodynamics, resulting in a multiplication of 10-fold interspecies difference with 10-fold interindividual variation. It was initially proposed by Lehman and Fitzhugh (Lehman A.J. et al. Association of the Food Drug Officials Quartely Bulletin 1954) over 60 years ago to convert a no-observed-adverse-effect level (NOAEL) from an animal toxicity study to a safe value for human intake (ADI). Scientists later applied the 100-fold uncertainty factor extensively to the selection of toxic dosage for animal experiments in toxicological studies, such as Lautz LS et al. Toxicol Lett, 2021; Arnot JA et al. J Expo Sci Environ Epidemiol, 2022; Tome D et al. Curr Opin Clin Nutr Metab Care, 2020; Cooper A.B. et al. Regul Toxicol Pharmacol, 2019. Meanwhile, the Unite States Environmental Protection Agency (EPA) also identified the 100-fold uncertainty factor as the criteria for animal study in conducting a human health risk assessment (https://www.epa.gov/risk/conducting-human-health-risk-assessment). In authoritative studies related to PM2.5, the 100-fold uncertainty factor is also considered as one of the criteria for in vivo experiments (Chen Q et al. J Exp Clin Cancer Res 2022; Li Y. et al. Ecotoxicol Environ Saf, 2021; Li J. et al. Sci Total Environ, 2020; Zhang J et al. Sci Total Environ 2018; Zhang Y et al. Sci Total Environ 2017.). In these PM2.5 related studies, the dosage of PM2.5 in the in vivo experiments was determined by combining the ambient PM2.5 level with the physiological indicators of mice, multiplying by 100 times the uncertainty factor, which aligned with our calculations. The only difference of the dosage between these study and our manuscript is the ambient PM2.5 exposure level. They employed ambient PM2.5 level recommended by WHO, while we employed ambient PM2.5 levels of pregnant women in Jinan area ( where we collected PM2.5 particules).

Additionally, the PM2.5 dosage we calculated was the total exposure of pregnant mice to PM2.5 throughout their entire pregnancy. In order to mitigate the impact of intratracheally instilled PM2.5 on pregnant rats, we partitioned the calculated dose of PM2.5 into three equal portions (rather than multiplying it by 4 times which reviewer mentioned) and administered them via intratracheal instillation on 1.5d, 7.5d, and 12.5d of pregnancy separately (the three points of time also corresponded to the First trimester, second trimester, and third trimester of human pregnancy respectively) (Theiler, K et al., Springer. Berlin 1972; Amack JD. et al. Cell Commun Signal 2021; Bunnell TM,et al. Cytoskeleton (Hoboken) 2010). Therefore, although the dosage in our in vivo experiment may appear higher than the PM2.5 dosage absorbed by normal pregnant women, it was determined through a scientific methodology that taked into consideration interspecies difference and interindividual variation. In order for reviewers and readers to better understand the reasons for our choice of PM2.5 dose in the in vivo experiment, we have revised the method section:

“At 8 weeks, the average tidal volume of Kunming mice was about 0.25 mL, and the frequency of per mouse’s respiratory was about 163/min, thus, the total air intake per day was 0.25 x 163 x 60 mins x 24 hrs = 58680 mL ≈ 0.0587 m^3^/day[41]. It has been reported that the PM2.5 exposure of pregnant women in Jinan in 2020 was about 64 μg/m^3^ daily [40]. Therefore, the total PM2.5 intake though out the whole pregnancy of mice was about 0.0587 m^3^/day × 64 μg/m^3^ × 20 days × 100 (uncertainty factor) = 7511 μg. The 100 fold uncertainty factor = 10-fold interspecies difference ×10-fold interindividual variation [95] [60]. We weighed 7511 μg PM2.5 particles and dissolved it in 60 μL PBS buffer to prepare a PM2.5 suspension, which was subsequently divided into three equal portions. Pregnant mice were anaesthetized by intraperitoneal injection of 0.5% pentobarbital sodium (50 mg/kg) on 1.5 d, 7.5 d, and 12.5 d of pregnancy (corresponding to first, second and third trimester of human), followed by intratracheal instillation of 20 μL PM2.5 suspension, and the control group was intratracheally instilled with the same volume of PBS (n = 8 for per group).” (P24 L563-572).

We have also added the text about 100 uncertainty factor to the Discussion section:

“The selection of appropriate PM2.5 exposure dosage in mice was critical for our experiments. We combined the PM2.5 exposure level of pregnant women from the Jinan [40] (where we collected PM2.5 particles) with physiological indicators of mice [41], then multiplied by 100-fold uncertainty factor to obtain the corresponding PM2.5 exposure dosage during mice pregnancy. The 100-fold uncertainty factor ( = 10-fold interspecies difference × 10-fold interindividual variation) is used to convert a no-observed-adverse-effect level (NOAEL) from an animal toxicity study to a safe value for human intake (ADI), which is the criteria for determining experimental dosages in toxicological studies involving animals. It was originally proposed, over 60 years ago, by Lehman and Fitzhugh[57], and still forms the basis of the uncertainty factors which are in use today. Also, the 100-uncertainty factor is considered one of the criteria for in vivo experiments in authoritative studies related to PM2.5 [58-60].” (P16 L383-392)

Reference:

Amack JD. Cellular dynamics of EMT: lessons from live in vivo imaging of embryonic development. Cell Commun Signal 2021, 19(1): 79;Bunnell TM, Ervasti JM. Delayed embryonic development and impaired cell growth and survival in Actg1 null mice. Cytoskeleton (Hoboken) 2010, 67(9): 564-572.Chen Q, Wang Y, Yang L, Sun L, Wen Y, Huang Y*, et al.* PM2.5 promotes NSCLC carcinogenesis through translationally and transcriptionally activating DLAT-mediated glycolysis reprograming. J Exp Clin Cancer Res 2022, 41(1): 229;Cooper AB, Aggarwal M, Bartels MJ, Morriss A, Terry C, Lord GA*, et al.* PBTK model for assessment of operator exposure to haloxyfop using human biomonitoring and toxicokinetic data. Regul Toxicol Pharmacol 2019, 102: 1-12.Lautz LS, Jeddi MZ, Girolami F, Nebbia C, Dorne J. Metabolism and pharmacokinetics of pharmaceuticals in cats (Felix sylvestris catus) and implications for the risk assessment of feed additives and contaminants. Toxicol Lett 2021, 338: 114-127.Lehman A.J., O.G. Fitzhugh, 100-fold margin of safety, Association of the Food Drug Officials Quartely Bulletin (1954).Li Y, Batibawa JW, Du Z, Liang S, Duan J, Sun Z. Acute exposure to PM(2.5) triggers lung inflammatory response and apoptosis in rat. Ecotoxicol Environ Saf 2021, 222: 112526.Li J, Hu Y, Liu L, Wang Q, Zeng J, Chen C. PM2.5 exposure perturbs lung microbiome and its metabolic profile in mice. Sci Total Environ 2020, 721: 137432.Theiler K, JQRoB. The house mouse : development and normal stages from fertilization to 4 weeks of age. 1972, 17(3): 133-145Tome D. Admissible daily intake for glutamate. Curr Opin Clin Nutr Metab Care 2020, 23(2): 133-137.Zhang J, Liu J, Ren L, Wei J, Duan J, Zhang L*, et al.* PM(2.5) induces male reproductive toxicity via mitochondrial dysfunction, DNA damage and RIPK1 mediated apoptotic signaling pathway. Sci Total Environ 2018, 634: 1435-1444;Zhang Y, Hu H, Shi Y, Yang X, Cao L, Wu J*, et al.* (1)H NMR-based metabolomics study on repeat dose toxicity of fine particulate matter in rats after intratracheal instillation. Sci Total Environ 2017, 589**:** 212-221;

Nonetheless, the authors show that administration of PM2.5 through intratracheal inhalation leads to increased fetal mortality, decreased fetal weight and number, damaged placental structure, and increased trophoblast apoptosis in mice and in isolated trophoblast cells. To investigate the molecular mechanisms underlying these phenotypes, the authors performed RNAseq on PM2.5-treated trophoblasts and identify two genes, CYP1A1 and KLF9, that are transcriptionally upregulated upon PM2.5 exposure. The authors find that genetic ablation of CYP1A1 diminishes oxidative stress and intrinsic cell death in PM2.5-treated cells, suggesting upregulation of this cytochrome P450 family member at least partially drives toxicity in this model. The authors then link KLF9, a transcription factor also upregulated in PM2.5-treated trophoblasts, to CYP1A1 expression, mapping the KLF9 responsive element in the CYP1A1 promoter. These data suggest that PM2.5 exposure induces KLF9-mediated transcription of CYP1A1 to promote oxidative stress and cell death in trophoblasts and in the placenta of pregnant mice. In a final set of experiments, the authors find that a relatively high dose of 20 mM metformin can ameliorate some toxic effects of PM2.5 in trophoblast cells, and that this correlates with a decrease in PM2.5-induced CYP1A1 and KLF9 expression.Collectively, the data presented by the authors convincingly demonstrate that: 1. The chosen dose of PM2.5 administered to mice causes significant increases in fetal mortality and morbidity; 2. Treatment of trophoblasts with PM2.5 causes oxidative stress, increases in apoptosis, and results in the transcriptional upregulation of CYP1A1 and KLF9; 3. KLF9 binds to the CYP1A1 promoter, driving its expression upon PM2.5 exposure, and that 4. CYP1A1 upregulation is necessary for PM2.5-induced oxidative stress. The authors also claim that "metformin could eliminate the toxicity induced by PM2.5 via the KLF9/CYP1A1 transcriptional regulatory axis" in their model trophoblast cell line, although this link is correlative as presented within the current study. While the data strongly implicate KLF9 and CYP1A1 to PM2.5-mediated toxicity, the mechanisms by which KLF9 senses PM2.5 were not established in this study, nor were the effects of KLF9 on gene products other than CYP1A1, which may also contribute to the phenotypes seen within this mouse model. Finally, though the reasoning for testing metformin in this disease model is not fully clear, there does seem to be therapeutic benefit at high doses of metformin to ameliorate phenotypes associated with PM2.5-mediated toxicity.

Many thanks for your comments on our manuscript. In our study, we employed a combination of RNA-Seq and validation assays to identify CYP1A1 as a crucial gene involved in PM2.5-induced oxidative stress in trophoblast cells. Subsequently, bioinformatic predictions were utilized to determine its transcription factor KLF9, which was then confirmed through Chip and Dual-luciferase assays to regulate the transcriptional activity of CYP1A1. As the reviewer suggested, we lacked research on the regulation of KLF9 by PM2.5 and whether its impact on genes beyond CYP1A1, which are the work we are currently engaged in. On one hand, our work exploring issues would take a lot of time, and on the other hand, I think adding PM2.5 regulation of KLF9 to the current article might complicate our research focus, so we will explore how PM2.5 regulates KLF9 and whether KLF9 affects trophoblastic biological function through other signaling pathways in future study. Meanwhile, I have incorporated the limitations of the current KLF9 study into the Discussion section:

“Furthermore, our current article lacks investigations on the regulatory of PM2.5 on KLF9. These limitations should be further investigated in future”. (P22 L525-527)

I would like to express my gratitude for your valuable advice once again.

This is an important study on the effects of PM2.5 on pregnancy outcomes in mice. Overall, the authors use multiple independent systems to test the effects of PM2.5 in both mice and in trophoblast cells, adding rigor to the approach. The authors put forth a number of claims in the abstract, most of which are well justified:1. "…PM2.5 induced adverse gestational outcomes such as increased fetal mortality rates, decreased fetal number and weight, damaged placental structure, and increased apoptosis of trophoblasts."2. "…PM2.5 induced dysfunction of the trophoblast cell line HTR8/SVneo, including its proliferation, apoptosis, invasion, migration, and angiogenesis."3. "…PM2.5 triggered oxidative stress and mitochondrial apoptosis to damage HTR8/SVneo cell biological functions through CYP1A1."4. "…PM2.5 stimulated KLF9, a transcription factor identified as binding to CYP1A1 promoter region."5. "…metformin could eliminate the toxicity induced by PM2.5 via the KLF9/CYP1A1 transcriptional regulatory axis in HTR8/SVneo."The first four of these claims are well justified. The data presented may support the last claim on metformin treatment, but the data supporting this are correlative (see revision suggestions below). While the data collected support the hypothesis that PM2.5 leads to oxidative stress via the upregulation of KLF9 and CYP1A1, I have four main concerns that the authors could address to strengthen this work, which I outline below.1. I am concerned with the dosages of particulate matter administered to the mice and cell models, and that these exceed physiologically relevant doses based on the calculations the authors provide in the methods section. The authors calculate a single dose of administration based on the average daily PM2.5 inhalation rate, and multiply it by 20 to reflect the full-term pregnancy in a mouse. However, the authors then use a 100-fold 'uncertainty factor' and administer this dose four times, despite the fact that the full pregnancy duration had already been factored into their calculations. Thus, one dose is ~100x the full exposure of PM2.5, but the authors administering this four times increases the potential exposure to ~400x what a pregnant woman in Jinan may experience. It is not hard to imagine that an excess dose may cause more severe pathologies than what women actually experience through PM2.5 inhalation. The authors should explain and justify why they used such high doses based on their own calculations and should repeat some key experiments with lower doses of PM2.5 to reflect physiologically relevant exposures.

Thank you very much for your suggestion on our manuscript. We apologize for your confusion about our results due to the over simplicity of our writing in the previous manuscript. Please kindly allow me to provide a detailed explanation of the dosage used in in vivo experiments.

In the in vivo experiments, we determined the PM2.5 exposure dosage for the whole gestation period by combining the PM2.5 exposure of pregnant women in Jinan and the physiological indicators of mice, multiplied by 100-fold uncertainty factor. 100-fold uncertainty factor was derived from the integration of toxicokinetics and toxicodynamics, resulting in a multiplication of 10-fold interspecies difference with 10-fold interindividual variation. It was initially proposed by Lehman and Fitzhugh (Lehman A.J. et al. Association of the Food Drug Officials Quartely Bulletin 1954) over 60 years ago to convert a no-observed-adverse-effect level (NOAEL) from an animal toxicity study to a safe value for human intake (ADI). Scientists later applied the 100-fold uncertainty factor extensively to the selection of toxic dosage for animal experiments in toxicological studies, such as Lautz LS et al. Toxicol Lett, 2021; Arnot JA et al. J Expo Sci Environ Epidemiol, 2022; Tome D et al. Curr Opin Clin Nutr Metab Care, 2020; Cooper A.B. et al. Regul Toxicol Pharmacol, 2019. Meanwhile, the Unite States Environmental Protection Agency (EPA) also identified the 100-fold uncertainty factor as the criteria for animal study in conducting a human health risk assessment (https://www.epa.gov/risk/conducting-human-health-risk-assessment). In authoritative studies related to PM2.5, the 100-fold uncertainty factor is also considered as one of the criteria for in vivo experiments (Chen Q et al. J Exp Clin Cancer Res 2022; Li Y. et al. Ecotoxicol Environ Saf, 2021; Li J. et al. Sci Total Environ, 2020; Zhang J et al. Sci Total Environ 2018; Zhang Y et al. Sci Total Environ 2017.). In these PM2.5 related studies, the dosage of PM2.5 in the in vivo experiments was determined by combining the ambient PM2.5 level with the physiological indicators of mice, multiplying by 100 times the uncertainty factor, which aligned with our calculations. The only difference of the dosage between these study and our manuscript is the ambient PM2.5 exposure level. They employed ambient PM2.5 level recommended by WHO, and we employed ambient PM2.5 levels of pregnant women in Jinan area ( where we collected PM2.5 particules).

Additionally, the PM2.5 dosage we calculated was the total exposure of pregnant mice to PM2.5 throughout their entire pregnancy. In order to mitigate the impact of intratracheally instilled PM2.5 on pregnant rats, we partitioned the calculated dose of PM2.5 into three equal portions (rather than multiplying it by 4 times which reviewer mentioned) and administered them via intratracheal instillation on 1.5d, 7.5d, and 12.5d of pregnancy separately (the three points of time also corresponded to the First trimester, second trimester, and third trimester of human pregnancy respectively) (Theiler, K et al., Springer. Berlin 1972; Amack JD. et al. Cell Commun Signal 2021; Bunnell TM,et al. Cytoskeleton (Hoboken) 2010). Therefore, although the dosage in our in vivo experiment may appear higher than the PM2.5 dosage absorbed by normal pregnant women, it was determined through a scientific methodology that taked into consideration interspecies difference and interindividual variation. In order for reviewers and readers to better understand the reasons for our choice of PM2.5 dose in the in vivo experiment, we have revised the method section:

“At 8 weeks, the average tidal volume of Kunming mice was about 0.25 mL, and the frequency of per mouse’s respiratory was about 163/min, thus, the total air intake per day was 0.25 x 163 x 60 mins x 24 hrs = 58680 mL ≈ 0.0587 m^3^/day[41]. It has been reported that the PM2.5 exposure of pregnant women in Jinan in 2020 was about 64 μg/m^3^ daily [40]. Therefore, the total PM2.5 intake though out the whole pregnancy of mice was about 0.0587 m^3^/day × 64 μg/m^3^ × 20 days × 100 (uncertainty factor) = 7511 μg. The 100 fold uncertainty factor = 10-fold interspecies difference ×10-fold interindividual variation [95] [60]. We weighed 7511 μg PM2.5 particles and dissolved it in 60 μL PBS buffer to prepare a PM2.5 suspension, which was subsequently divided into three equal portions. Pregnant mice were anaesthetized by intraperitoneal injection of 0.5% pentobarbital sodium (50 mg/kg) on 1.5 d, 7.5 d, and 12.5 d of pregnancy (corresponding to first, second and third trimester of human), followed by intratracheal instillation of 20 μL PM2.5 suspension, and the control group was intratracheally instilled with the same volume of PBS (n = 8 for per group).” (P24 L563-572).

We have also added the text about 100 uncertainty factor to the Discussion section:

“The selection of appropriate PM2.5 exposure dosage in mice was critical for our experiments. We combined the PM2.5 exposure level of pregnant women from the Jinan [40] (where we collected PM2.5 particles) with physiological indicators of mice [41], then multiplied by 100-fold uncertainty factor to obtain the corresponding PM2.5 exposure dosage during mice pregnancy. The 100-fold uncertainty factor ( = 10-fold interspecies difference × 10-fold interindividual variation) is used to convert a no-observed-adverse-effect level (NOAEL) from an animal toxicity study to a safe value for human intake (ADI), which is the criteria for determining experimental dosages in toxicological studies involving animals. It was originally proposed, over 60 years ago, by Lehman and Fitzhugh[57], and still forms the basis of the uncertainty factors which are in use today. Also, the 100-uncertainty factor is considered one of the criteria for in vivo experiments in authoritative studies related to PM2.5 [58-60].” (P16 L383-392)

Reference:

Amack JD. Cellular dynamics of EMT: lessons from live in vivo imaging of embryonic development. Cell Commun Signal 2021, 19(1): 79;Bunnell TM, Ervasti JM. Delayed embryonic development and impaired cell growth and survival in Actg1 null mice. Cytoskeleton (Hoboken) 2010, 67(9): 564-572Chen Q, Wang Y, Yang L, Sun L, Wen Y, Huang Y*, et al.* PM2.5 promotes NSCLC carcinogenesis through translationally and transcriptionally activating DLAT-mediated glycolysis reprograming. J Exp Clin Cancer Res 2022, 41(1): 229;Cooper AB, Aggarwal M, Bartels MJ, Morriss A, Terry C, Lord GA*, et al.* PBTK model for assessment of operator exposure to haloxyfop using human biomonitoring and toxicokinetic data. Regul Toxicol Pharmacol 2019, 102: 1-12;Lautz LS, Jeddi MZ, Girolami F, Nebbia C, Dorne J. Metabolism and pharmacokinetics of pharmaceuticals in cats (Felix sylvestris catus) and implications for the risk assessment of feed additives and contaminants. Toxicol Lett 2021, 338: 114-127.Lehman A.J. O.G. Fitzhugh, 100-fold margin of safety, Association of the Food Drug Officials Quartely Bulletin (1954).Li Y, Batibawa JW, Du Z, Liang S, Duan J, Sun Z. Acute exposure to PM(2.5) triggers lung inflammatory response and apoptosis in rat. Ecotoxicol Environ Saf 2021, 222: 112526.Li J, Hu Y, Liu L, Wang Q, Zeng J, Chen C. PM2.5 exposure perturbs lung microbiome and its metabolic profile in mice. Sci Total Environ 2020, 721: 137432.Theiler K, JQRoB. The house mouse : development and normal stages from fertilization to 4 weeks of age. 1972, 17(3): 133-145Tome D. Admissible daily intake for glutamate. Curr Opin Clin Nutr Metab Care 2020, 23(2): 133-137.Zhang J, Liu J, Ren L, Wei J, Duan J, Zhang L*, et al.* PM(2.5) induces male reproductive toxicity via mitochondrial dysfunction, DNA damage and RIPK1 mediated apoptotic signaling pathway. Sci Total Environ 2018, 634: 1435-1444;Zhang Y, Hu H, Shi Y, Yang X, Cao L, Wu J*, et al.* (1)H NMR-based metabolomics study on repeat dose toxicity of fine particulate matter in rats after intratracheal instillation. Sci Total Environ 2017, 589: 212-221;

2. The metformin data at the end of the manuscript are unnecessary, and I believe they should be removed from the manuscript. While the authors demonstrate that a single high dose (20 mM) metformin can alleviate select phenotypes induced by PM2.5, these data are not rigorous enough to justify the use of metformin to alleviate PM2.5 toxicity in mice. Collection of such data would be beyond the scope of this manuscript, as it is already 11 figures worth of data. Furthermore, the authors suggest that metformin acts through the CYP1A1/KLF9 signaling axis, but they only show correlative data that these two proteins change at one dose of metformin administered for one timepoint. While these two key PM2.5 targets do change in response to metformin, significantly more data would need to be collected to demonstrate that these targets are required for this response. Finally, the justification for looking at metformin is obscure, and it is unclear why the authors chose this compound other than their statement that metformin is a 'miracle' drug. Collectively, the lack of rationale, effects only at a single high dose, and correlative nature of involvement with the KLF9/CYP1A1 pathway leave this line of investigation with insufficient rigor. As a substantial amount of work would be required to alleviate this, it would be advisable for the authors to remove this portion of the manuscript so as to be able to more fully characterize these phenotypes in a future study.

Thank you very much for your valuable advice for our manuscript. In our study, we have observed a robust correlation between PM2.5 and adverse pregnancy outcomes; however, there is currently no clinical intervention or prophylactic medication available to alleviate this association. To mitigate the incidence of unfavorable pregnancy outcomes in the presence of atmospheric pollution, we had performed some research of metformin. However, our investigation of metformin was insufficient because of time limit. We only utilized a single concentration of metformin and solely examined its mechanistic effects on KLF9 and CYP1A1. Therefore, as suggested by the reviewer, we have removed the metformin-related content. Considering metformin could be a promising avenue for the clinical management of adverse pregnancy outcomes associated with PM2.5 exposure. So, we will explore the effects of different concentrations of metformin and investigate the mechanism of metformin through cell function experiments combined with new RNA sequencing in the future work.

3. Some data would be more powerful if shown in a more quantitative manner. This includes the MitoSOX microscopy (Figures 5G, 7I, 10G) and the JC-1 microscopy (Figures 5H, 7J, 10F). There are also portions of the manuscript in which the authors claim "significant" changes in their data, but these experiments are not accompanied by statistical analysis (Figure 7B… "The results showed that, compared to the control group, the expression of CYP1A1 was significantly increased by PM2.5 exposure.") The authors should change the verbiage (e.g., 'substantially') or perform the relevant statistical analysis (which would be preferred).

Many thanks for your suggestion. We have added the relevant statistical analysis for the MitoSOX staining results in Figure 5H, 7J, 10I and for the JC-1 staining results in Figure 5J, 7L, 10J. The figure legends have also been added in line with the new figures.

4. The paper is too long, particularly in the discussion and conclusions (8 pages). The manuscript would be better if it were shortened.

Many thanks for your suggestion. We have eliminated the metformin-related content and streamlined the discussion to 2200 words from previous 2612 words. Additionally, we have condensed the conclusion to 111 words from previous 163 words.

Reviewer #3 (Recommendations for the authors):This article investigated the PM2.5-induced toxicity on pregnancy outcomes and trophoblasts and revealed a novel mechanism by which PM2.5 caused trophoblast mitochondrial damage through KLF9/CYP1A1 transcriptional axis. Furthermore, this was the first time to identify KLF9 acted as the transcription factor positively modulating CYP1A1 in humans. This article also suggested a potential therapeutic effect of metformin. Overall, the work is logical and well supported by the data organized, and sufficiently innovative.

Thank you very much for your recognition on our study.

I have a few suggestions to improve the manuscript, please see below.1. The authors constructed a PM2.5-exposed pregnant mice model by intratracheal instillation. The concentrations setting of PM2.5 were described in detail, but the selection of PM2.5 treatment time points (1.5 d, 7.5 d, and 12.5 d of pregnancy) was not explained. The authors need to describe the reason why these time points were chosen. In addition, the intratracheal instillation did not simulate PM2.5 environmental exposure properly, could the authors construct the animal model of PM2.5 exposure using a meteorological and environmental animal exposure system? (Ran Z, An Y, Zhou J, Yang J, Zhang Y, Yang J, et al. Subchronic exposure to concentrated ambient PM2.5 perturbs gut and lung microbiota as well as metabolic profiles in mice. Environ Pollut 2021, 272: 115987)

Many thanks for your advice. In our in vivo experiments, we opted to perform intratracheal instillation on pregnant rats at 1.5d, 7.5d and 12.5d of pregnancy to mitigate potential harm caused by the procedure. These time points corresponded to First, second and third trimesters respectively, given that the gestation period is approximately 20 days for rats and 280 days for humans (Theiler, K et al. Springer. Berlin 1972; Amack JD. et al. Cell Commun Signal 2021; Bunnell TM,et al. Cytoskeleton (Hoboken) 2010). So, we performed PM2.5 intratracheal instillation at these three time points to investigate the effect of PM2.5 on the entire gestation period. In many toxicological studies on embryonic development, same or similar time point was also chosen for in vivo experiments (Koren O et al. Cell 2012; Tata B et al. Nat Med 2018).

As recommended by the reviewer, intratracheal instillation is not a faithful representation of PM2.5 environmental exposure. However, due to our limited laboratory resources, we are currently restricted to conducting in vivo experiments via intratracheal instillation – a commonly employed method for in vivo studies related to PM2.5 such as Chen Q et al. J Exp Clin Cancer Res 2022; Zhang J et al. Sci Total Environ 2018; Zhang Y et al. Sci Total Environ 2017. In future research, we will collaborate with other advanced experimental platforms to conduct more scientifically rigorous studies on PM2.5 using meteorological and environmental animal exposure system. We appreciate your valuable suggestion on our testing methodologies.

Reference:

(1) Amack JD. Cellular dynamics of EMT: lessons from live in vivo imaging of embryonic development. Cell Commun Signal 2021, 19(1): 79.

(2) Bunnell TM, Ervasti JM. Delayed embryonic development and impaired cell growth and survival in Actg1 null mice. Cytoskeleton (Hoboken) 2010, 67(9): 564-572

(3) Chen Q, Wang Y, Yang L, Sun L, Wen Y, Huang Y, et al. PM2.5 promotes NSCLC carcinogenesis through translationally and transcriptionally activating DLAT-mediated glycolysis reprograming. J Exp Clin Cancer Res 2022, 41(1): 229.

(4) Koren O, Goodrich JK, Cullender TC, Spor A, Laitinen K, Backhed HK, et al. Host remodeling of the gut microbiome and metabolic changes during pregnancy. Cell 2012, 150(3): 470-480.

(5) Tata B, Mimouni NEH, Barbotin AL, Malone SA, Loyens A, Pigny P, et al. Elevated prenatal anti-Mullerian hormone reprograms the fetus and induces polycystic ovary syndrome in adulthood. Nat Med 2018, 24(6): 834-846.

(6) Theiler K, JQRoB. The house mouse: development and normal stages from fertilization to 4 weeks of age. 1972, 17(3): 133-145

(7) Zhang J, Liu J, Ren L, Wei J, Duan J, Zhang L, et al. PM(2.5) induces male reproductive toxicity via mitochondrial dysfunction, DNA damage and RIPK1 mediated apoptotic signaling pathway. Sci Total Environ 2018, 634: 1435-1444.

(8) Zhang Y, Hu H, Shi Y, Yang X, Cao L, Wu J, et al. (1)H NMR-based metabolomics study on repeat dose toxicity of fine particulate matter in rats after intratracheal instillation. Sci Total Environ 2017, 589: 212-221.

2. In result 3.9, the authors concluded that metformin reversed oxidative stress damage caused by PM2.5 through the KLF9/CYP1A1 transcriptional axis, but there were no results on the impact of metformin on the transcriptional level of CYP1A1. The authors should provide additional experiments to illustrate this issue and also add pertinent content to the discussion.

Thank you very much for your advice. In the previous manuscript, we only confirmed the reversal of PM2.5-induced increase in CYP1A1 protein expression levels by metformin through Western Blot. As suggested by the reviewer, we assessed whether metformin modulated CYP1A1 mRNA levels. Total RNA was extracted from control, PM2.5 and PM2.5+metformin groups followed by RT-PCR analysis for evaluating CYP1A1 mRNA expression levels. The results showed that metformin reduced the increase in CYP1A1 mRNA expression caused by PM2.5, as shown in Author response image 2.

The other two reviewers and the editor expressed concerns regarding the depth of our study on metformin, citing that we only examined one concentration of the metformin and investigated its impact on a limited number of genes (KLF9 and CYP1A1). As such, they suggested us to remove the section of metformin from this article and instead devote another study to exploring its effects in greater detail. So we have deleted the metformin section from our current manuscript and conduct a more in-depth analysis of its role in another study. Your suggestions regarding the regulation of CYP1A1 mRNA expression levels by metformin will be incorporated into this future study. Thank you once again for your valuable suggestions regarding our metformin section.

3. As the authors described in the paper, trophoblast biological functions such as invasion, migration, and tube formation were essential for placental development. The authors did not conduct relevant studies exploring the toxic effects of metformin in protecting trophoblast cells from PM2.5. The authors should perform further experiments to optimize the effects of metformin.

Many thanks for your advice. We have conducted experiments to investigate the potential effect of metformin in reversing PM2.5-induced functional impairment of trophoblasts as per your recommendation. Our findings demonstrated that metformin can partially restore the invasion, migration and tube-forming capacity of trophoblasts. The results were delineated as in Author response image 3.

**Author response image 3. sa2fig3:** 

These findings reinforced the protective role of metformin against PM2.5-induced damage in trophoblast cells. Although we have excluded metformin from the current manuscript based on suggestions from the other two reviewers and editors, your valuable advice will guide our future research on metformin, and we plan to incorporate these results into the forthcoming article. Thank you again for your insightful advice.

4. KLF9 is the focus of this manuscript to explore the toxic effects of PM2.5, but there are not many studies describing the role of KLF9 in toxicology in the Discussion section. For example, YUE GU et al. found that Klf9 is involved in BLM-induced pulmonary toxicity in human lung fibroblasts (Gu Y, Wu YB, Wang LH, Yin JN. Involvement of Kruppel-like factor 9 in bleomycin-induced pulmonary toxicity. Mol Med Rep 2015, 12(4): 5262-5266.) and Daqian Yang et al. identified that KLF9 was essential in allicin resisting against arsenic trioxide-Induced hepatotoxicity (Yang D, Lv Z, Zhang H, Liu B, Jiang H, Tan X, et al. Activation of the Nrf2 Signaling Pathway Involving KLF9 Plays a Critical Role in Allicin Resisting Against Arsenic Trioxide-Induced Hepatotoxicity in Rats. Biol Trace Elem Res 2017, 176(1): 192-200). Previously published evidence of KLF9-dependent toxicological responses needs to be cited more clearly in the manuscript.

Many thanks for your advice. As per the reviewer's suggestion, we have incorporated a section in our discussion elucidating the role of KLF9 in toxicological studies:

“There are also numerous studies indicating the critical role of KLF9 in toxicological research. For example, Yue Gu et al. found that Klf9 is involved in BLM-induced pulmonary toxicity in human lung fibroblasts, Daqian Yang et al. identified that KLF9 was essential in allicin resisting against arsenic trioxide-Induced hepatotoxicity, but little is known regarding its role in the occurrence of PM2.5-induced toxicological processes.” (P22 L514-518)

5. Is it possible to evaluate the role of KLF9 in placental toxicity caused by PM2.5 using knockout mice models?

Many thanks for your advice. As suggested by the reviewer, the conduction of KLF9 knockout mice would facilitate a more comprehensive investigation into the role of KLF9 in PM2.5-induced placental toxicity. However, due to technological constraints within our laboratory, we are currently unable to generate KLF9 knockout mice. In future studies, we plan to collaborate with external laboratories to establish this model and further elucidate the role of KLF9.

6. The authors used only trophoblast cell line HTR8/SVneo for in vitro experiments. Could it be possible to add trophoblast primary cells or other primary cells such as HUVEC? The authors should mention this and refer to this point in the manuscript.

Many thanks for the helpful advice. HTR-8/SVneo trophoblast cell line is the representative and commonly used extravillous trophoblasts (EVTs) model for research on trophoblast function. HTR-8/SVneo has been utilized as the subject of study in numerous authoritative articles on trophoblast toxicology (Li T et al. Environ Res 2022; Hu J et al. Environ Pollut 2022; Liao Y et al. Ecotoxicol Environ Saf 2021; Liu W et al. Ecotoxicol Environ Saf 2022). Therefore, the HTR8 trophoblast cell line was selected for in vitro experiments in the current study to investigate the impact of PM2.5 on trophoblast cells. As the reviewer suggested, together by using trophoblasts primary cells can better illustrate the effect of PM2.5 on the impairment of trophoblasts. However, the extraction of primary trophoblast cells remains a challenge for us due to current limitations in laboratory technology. We have also included our limitations for this point in the Discussion section (“We also lack the utilization of primary cells in our in vitro experiments. These limitations should be further investigated in future” ).(P22 L525-526) At present, we are cultivating trophoblast stem cells, and then inducing them to differentiate into extravillous trophoblasts with the characteristics of primary trophoblast. We will definitely use primary trophoblast cells in our future research.

Reference:

Hu J, Zhu Y, Zhang J, Xu Y, Wu J, Zeng W*, et al.* The potential toxicity of polystyrene nanoplastics to human trophoblasts in vitro. Environ Pollut 2022, 311: 119924.Li T, Li Z, Fu J, Tang C, Liu L, Xu J*, et al.* Nickel nanoparticles exert cytotoxic effects on trophoblast HTR-8/SVneo cells possibly via Nrf2/MAPK/caspase 3 pathway. Environ Res 2022, 215(Pt 2): 114336.Liao Y, Peng S, He L, Wang Y, Li Y, Ma D*, et al.* Methylmercury cytotoxicity and possible mechanisms in human trophoblastic HTR-8/SVneo cells. Ecotoxicol Environ Saf 2021, 207: 111520.Liu W, Li S, Zhou Q, Fu Z, Liu P, Cao X*, et al.* 2, 2', 4, 4'-tetrabromodiphenyl ether induces placental toxicity via activation of p38 MAPK signaling pathway in vivo and in vitro. Ecotoxicol Environ Saf 2022, 244: 114034.

7. In result 3.8, the authors described "Cellular immunofluorescence showed that the increase in KLF9 expression was primarily observed in the nucleus (Figure 10D)". The authors assumed that KLF9 functioned as a transcription factor through nuclear translocation. To better corroborate this conclusion, I suggest that the authors should extract nuclear protein and detect the expression of KLF9 quantitatively.

Many thanks for your suggestion. As suggested by the reviewer, extractng nuclear protein and assessing KLF9 expression would provide greater clarity regarding the role of KLF9 as a transcription factor through nuclear translocation. Therefore, we isolated nuclear proteins from PM2.5-treated cells and assessed the expression of KLF9. The results revealed that the expression level of KLF9 in the nuclei was increased in a PM2.5 concentration-dependent manner. The results were delineated in Figure 10—figure supplement 1A.

This result reinforced the prvotal role of PM2.5 in promoting KLF9 as a transcription factor via nuclear translocation. We have added this to the Figure 10—figure supplement 1A and Results section (“The protein expression level of KLF9 in the nucleus was increased with the concentration of PM2.5 by western blotting (Figure 10—figure supplement 1A)”). (P15 L357)

8. AHR has been recognized as a receptor of environmental pollutants and a mediator of chemical toxicity including PM2.5. Meanwhile, AHR is an essential ligand-activated transcription factor of CYP1A1, which is also mentioned in the discussion of the manuscript. I am interested in the expression of AHR in trophoblast cells under PM2.5 exposure, does PM2.5 also increase AHR expression? Is it possible for the authors to conduct research on the role of PM2.5 on AHR?

Many thanks for your advice. The primary concern of our study is to identify the molecular mechanism of PM2.5 induced cytotoxicity on trophoblasts mediated by a novel transcription factor KLF9 and its regulation on CYP1A1. So we didn’t consider AHR. As the reviewer has pointed out, AHR is a crucial transcription factor that plays a pivotal role in regulating CYP1A1 expression, which has also been substantiated by numerous authoritative studies. We followed the reviewer's suggestion and conducted tests to investigate whether PM2.5 affects AHR expression. Western Blot results demonstrated that increasing concentrations of PM2.5 led to elevated levels of AHR expression as follows, suggesting that AHR may also play a critical role in the impact of PM2.5 on trophoblast cells.

**Author response image 4. sa2fig4:** 

In our present article, we have omitted AHR-related studies to clarify the primary focus of our research. However, in future investigations, we intend to concentrate on AHR and explore its role in PM2.5-induced adverse pregnancy outcomes. Thank you for providing valuable guidance on the future direction of our research.[Editors’ note: what follows is the authors’ response to the second round of review.]

The manuscript has been improved, but there are some remaining issues that need to be addressed, as outlined below:To ensure transparency and reproducibility, we kindly request the supplemental material containing the complete dataset of the RNA-Seq analysis, including the total of 17,795 genes identified and the corresponding statistical parameters.Reviewer #1 (Recommendations for the authors):This study used PM2.5 collected from the urban region of Jinan, China, to establish pollutant exposure experiments by in vivo animal models and trophoblast cells. This relevant study applies several in vitro and in silico techniques from these models. The authors identify that PM2.5 activates the KLF9/CYP1A1 signaling pathway causing oxidative stress damage with mitochondrial apoptosis, which can be correlated to poor pregnancy outcomes observed in mice.In general, the authors have adequately addressed the primary issues I raised in this revised version.However, the importance of the authors providing a supplementary Table containing the complete dataset of the RNA-Seq analysis is necessary. In the current scientific landscape, transparency and reproducibility of experiments are vital principles that promote the advancement of knowledge and enable the scientific community to build upon previous findings.By providing the entire dataset, including the total of 17,795 genes identified in the sequencing, along with the corresponding statistical parameters of the differential expression analysis (DEG) such as P-value, FDR, and Log2FC, you will contribute significantly to the transparency and reproducibility of your study. This will enable other researchers to validate and replicate your findings, facilitating scientific progress.

Firstly, we would like to express our gratitude for your recognition on our study. Secondly, we wholeheartedly concur with your advice that providing comprehensive RNA-Seq analysis data is necessary. Because our RNA-Seq sequencing data file was too large, containing both the raw data and processed data, we have uploaded the sequencing data into GEO database (accession number: GSE237795). The aforementioned information has been incorporated into the manuscripts:

“The RNA sequencing data were deposited into the Gene Expression Omnibus (GEO) database (accession number: GSE237795).”(P30 L709).

We have also indicated in the *eLife* submission system that we have deposited the RNA-Seq sequencing data into GEO database. As our article is yet to be officially published, we have designated August 1, 2023 as the date for the GEO database system to release RNA-Seq sequencing data to the public. We encourage further utilization of our RNA-Seq sequencing data by other researchers to investigate the impact of PM2.5 on pregnancy.

Thank you for your all valuable suggestion on our manuscript, which help making our study more comprehensive and in-depth.

Reviewer #2 (Recommendations for the authors):This is an important study in which Zhang, Wang, and colleagues collect PM2.5 samples from city air and use these to establish a mouse model of air pollutant exposure and its effects on pregnancy. Using a wide breadth of techniques, the authors use this model to show that PM2.5 induces a KLF9/CYP1A1 signaling axis that leads to placental dysfunction, oxidative stress, mitochondrial dysfunction, and poor gestational outcomes.In the first draft of this paper, I had four concerns. The first involved the dosages of the particulate matter administered in mouse and cell models, as these seemingly were too high to model physiologically relevant doses. The authors have provided further context into their rationale in choosing the doses and administration time frames, which I believe are necessary for the general and broad audience of eLife's readership. My second concern was the lack of depth in the original metformin analysis, which the authors have removed. My third concern involved the lack of quantitative representation of select figures, including MitoSOX microscopy (Figures 5G, 7I, 10G) and the JC-1 microscopy (Figures 5H, 7J, 10F). These analyses have been added. Finally, I suggested that the paper could be shortened, which the authors addressed through elimination of their discussion of the metformin data. Overall, the authors have addressed my concerns in this revision.

Thank you very much for your recognition on our study and our revison of first draft. We would like to express our gratitude for your valuable suggestions on our study. Your suggestions and advice for our manuscript are very important and the revised manuscript has become more comprehensive and in-depth. Thank you again for your altruistic help!